# The development and evolution of inhibitory neurons in primate cerebrum

Matthew T. Schmitz[1,2], Kadellyn Sandoval[2,3], Christopher P. Chen[1,2], Mohammed A. Mostajo-Radji[1,2], William W. Seeley[2], Tomasz J. Nowakowski[3,4,5,6,7], Chun Jimmie Ye[7,8,9,10], Mercedes F. Paredes[2,3] & Alex A. Pollen[1,2,3,7 ✉]

Neuroanatomists have long speculated that expanded primate brains contain an increased morphological diversity of inhibitory neurons (INs)[1], and recent studies have identified primate-specific neuronal populations at the molecular level[2]. However, we know little about the developmental mechanisms that specify evolutionarily novel cell types in the brain. Here, we reconstruct gene expression trajectories specifying INs generated throughout the neurogenic period in macaques and mice by analysing the transcriptomes of 250,181 cells. We find that the initial classes of INs generated prenatally are largely conserved among mammals. Nonetheless, we identify two contrasting developmental mechanisms for specifying evolutionarily novel cell types during prenatal development. First, we show that recently identified primate-specific *TAC3* striatal INs are specified by a unique transcriptional programme in progenitors followed by induction of a distinct suite of neuropeptides and neurotransmitter receptors in new-born neurons. Second, we find that multiple classes of transcriptionally conserved olfactory bulb (OB)-bound precursors are redirected to expanded primate white matter and striatum. These classes include a novel peristriatal class of striatum laureatum neurons that resemble dopaminergic periglomerular cells of the OB. We propose an evolutionary model in which conserved initial classes of neurons supplying the smaller primate OB are reused in the enlarged striatum and cortex. Together, our results provide a unified developmental taxonomy of initial classes of mammalian INs and reveal multiple developmental mechanisms for neural cell type evolution.

To examine the diversity of primate inhibitory neurons (INs) during development, we dissected progenitor zones in the ventral telencephalon and migratory destinations in the cortex and basal nuclei of prenatal rhesus macaque brains. We focused on the lateral, medial and caudal ganglionic eminences (LGE, MGE and CGE, respectively) and ventromedial forebrain (VMF) regions including the septum, preoptic area and preoptic hypothalamus, where distinct configurations of transcription factors specify initial IN classes[3–6]. We also sampled migratory destinations in the cortex and basal nuclei, where local signals further influence the maturation and postmitotic refinement of neuron classes[7]. In total, we collected 71 samples across 9 specimens, spanning the onset of cortical neurogenesis, postconception day 40 (PCD40), to the conclusion, PCD100 (ref. [8]; Fig. 1a and Extended Data Fig. 1). We performed single-cell RNA sequencing using the 10x Chromium Controller, incorporating recent data from PCD110 (ref. [9]) and mouse studies from PCD13 to PCD21 (refs [10,11]) as well as adult olfactory bulb (OB)[12] (Supplementary Table 1). We applied stringent quality control,

batch correction, dimensionality reduction, Leiden clustering and RNA velocity trajectory analysis to identify transcriptionally similar classes of progenitors and postmitotic INs among 109,112 macaque cells and 141,065 mouse cells, which were identified by expression of *DLX* and *GAD* genes (Methods).

Macaque and mouse IN progenitors clustered mainly by cell cycle phase rather than spatial origin (Fig. 1b–d and Extended Data Figs. 1–3). Similarly, the most immature new-born neurons clustered by both class and differentiation stage, according to the RNA velocity latent time (Fig. 1b–d and Extended Data Figs. 1–3). From the Leiden clusters, we delineated 11 discrete initial classes of macaque postmitotic neurons, which resolved to 17 initial classes in mouse (Fig. 1c, d). We found that canonical marker genes for established progenitor territories exhibited significant correlations among the progenitors, suggesting that core transcriptional regulatory programmes that are present at or before the last cell division predict the identity of postmitotic neurons in the initial classes. For example, signatures reflecting spatial origin and subtypes

[1]Eli and Edythe Broad Center for Regeneration Medicine and Stem Cell Research, University of California, San Francisco, San Francisco, CA, USA. [2]Department of Neurology, University of California, San Francisco, San Francisco, CA, USA. [3]Weill Institute for Neurosciences, University of California, San Francisco, San Francisco, CA, USA. [4]Department of Anatomy, University of California, San Francisco, San Francisco, CA, USA. [5]Department of Psychiatry and Behavioral Sciences, University of California, San Francisco, San Francisco, CA, USA. [6]Department of Neurological Surgery, University of California, San Francisco, San Francisco, CA, USA. [7]Chan Zuckerberg Biohub, San Francisco, CA, USA. [8]Institute for Human Genetics, University of California, San Francisco, San Francisco, CA, USA. [9]Department of Epidemiology and Biostatistics, University of California, San Francisco, San Francisco, CA, USA. [10]Parker Institute for Cancer Immunotherapy, San Francisco, CA, USA. ✉e-mail: alex.pollen@ucsf.edu

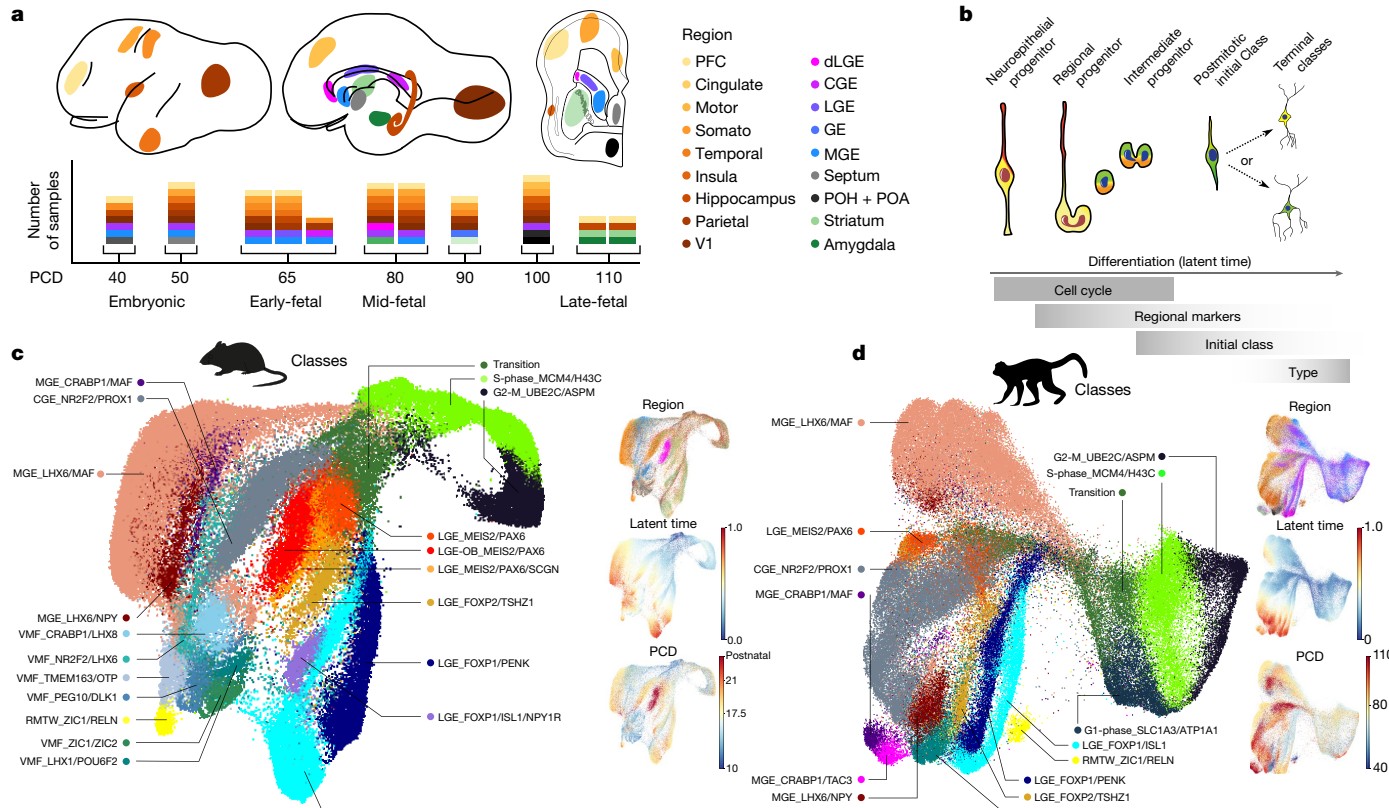

**Fig. 1 | Transcriptional diversity of IN precursors in developing macaque and mouse telencephalon. a**, Regions dissected for single-cell RNA sequencing, labelled on PCD80 macaque lateral, medial and frontal coronal section traces. Columns of stacked boxes represent samples from each individual, with each box representing a region dissected and sampled. PCD110 samples from Zhu et al.[9]. PFC, prefrontal cortex; POH, preoptic hypothalamus; POA, preoptic area. **b**, Model of inhibitory neurogenesis. **c**, **d**, UMAP projections coloured by progenitor state and initial class for mice (**c**) and macaques (**d**). Insets in **c**, **d** show the dissection region from **a**, the scVelo dynamical RNA velocity shared latent time and age.

in the CGE (*NR2F1* and *NR2F2*), LGE (*MEIS2*, *EBF1* and *ISL1*), MGE (*MAF*, *LHX6* and *CRABP1*) and VMF (*ZIC1* and *ZIC2*) were visible in G1 progenitors, and signatures related to four distinct LGE initial classes further emerged in dividing cells as correlations of the pan-LGE marker *MEIS2* to *PAX6*, *FOXP2*, *ISL1* or *PENK* (Extended Data Fig. 1d). These observations support a model in which we refer to differentiating postmitotic neuron clusters that are largely defined by germinal zone regional transcription factors as initial classes of new-born neurons. This terminology reflects the idea that a small number of discrete transcriptional classes are initially produced following a neuron's final cell division and that each of these immature transcriptional states is later partitioned into one or many different mature classes by nurture and circumstance as neurons migrate and integrate into the circuitry (Fig. 1b)[7]. Although many studies have demonstrated that these transcriptionally defined classes result from shared lineage relationships[13], we refrain from referring to classes as lineages in the absence of direct lineage tracing data.

## Conserved and divergent initial classes

To construct a taxonomy of IN development, we sought to identify evolutionarily conserved cell classes and to link them to candidate adult populations[14]. We found most well-known initial class markers to be conserved between species (Fig. 2a and Extended Data Fig. 4). Gene expression signatures for each progenitor and postmitotic initial class in macaques correlated strongly with at least one comparable class in mice (Extended Data Fig. 4). Most classes had one-to-one relationships, although subclasses such as LGE_FOXP1/ISL1/NPY1R and LGE_MEIS2/PAX6/SCGN as well as a number of VMF classes were apparent in the mouse data but were undersampled in macaques. Because cell type

correlation methods depend on clustering resolution in each species, we further examined homology at the level of individual cells. Mutual nearest-neighbour analysis showed that all telencephalic initial classes present in mice were also present in macaques (Fig. 2b). To infer the putative fates of the initial classes in the absence of lineage tracing, we compiled the most complete available data of adult mouse brains (Extended Data Fig. 5). We then computed the terminal class absorption probabilities for prenatal neurons using nearest-neighbour relationships and RNA velocity with equal weight in CellRank's Markov chain model[15]. Our predicted mapping of postmitotic differentiation and partitioning of each initial class using transcriptional similarities recapitulated known lineage relationships and made a number of unexpected predictions that support unresolved linkages in the literature, as summarized in Fig. 2c, such as an *NKX2-1*[+] MGE-derived *LAMP5*[+] cortical chandelier population[16–18] and a shared origin of amygdala intercalated cells (ITCs) and striatal eccentric spiny neurons[19–21]. The widespread distribution and diversity of derivatives from some initial classes such as MGE_LHX6/NPY also highlights the shared genetic programmes underlying the initial specification of populations that later diversify according to regional destinations, where terminal classes are commonly subdivided into many transcription types and morphotypes. Although our results suggest that the initial classes are largely uniform, minor axes of variation may already exist within classes, which could trigger downstream cascades that bias terminal fate partitioning. However, such variation cannot be identified here without knowing a cell's fate a priori.

Recent comparative studies of adult primate, rodent and ferret telencephalon showed a primate-specific population of striatal INs that express the neuropeptide TAC3 (ref. [2]), although the developmental

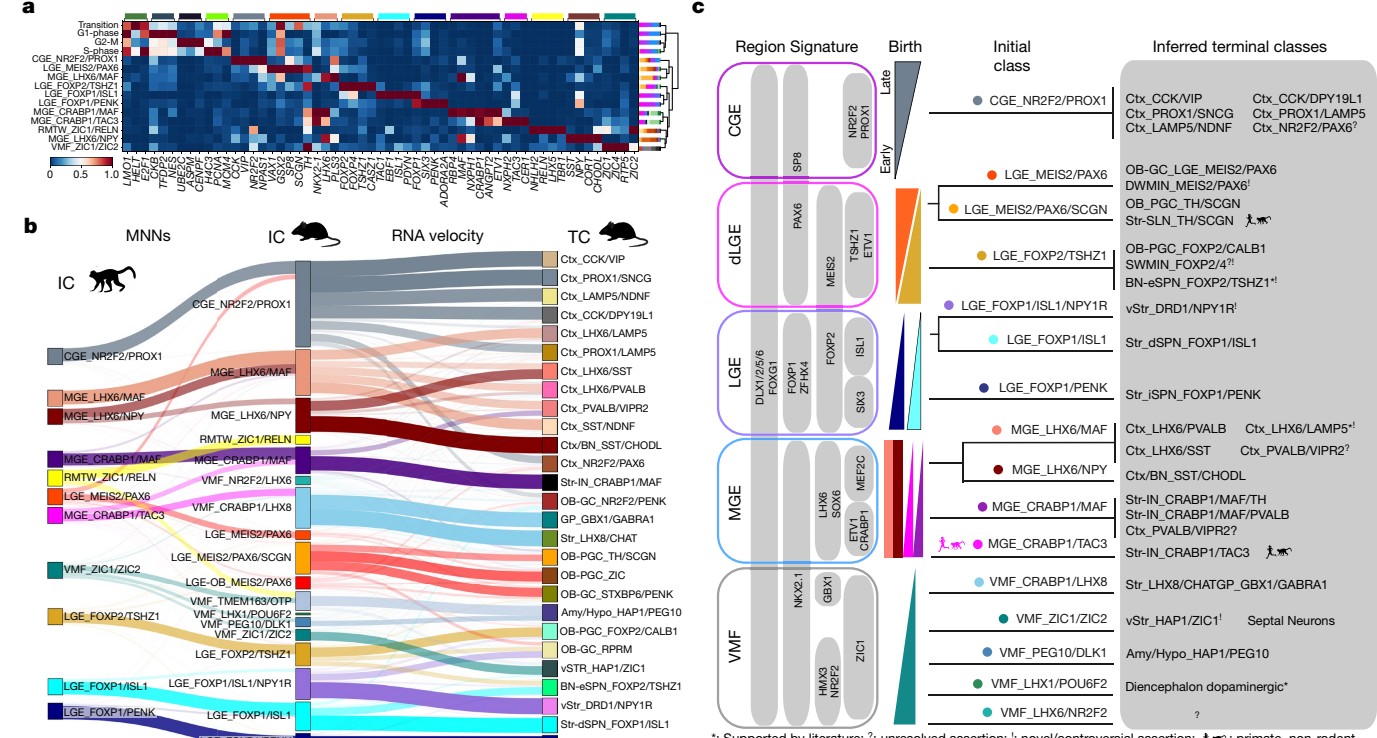

**Fig. 2 | Unified taxonomy of Euarchontogliran telencephalic IN specification. a**, Heatmap of macaque initial class marker expression, scaled by column. Stacked barplots correspond to region of origin for cells in each class. The dendrogram represents complete linkage of the Pearson correlation distance of mean expression values. Stacked bar plots show the regional distribution of each class. **b**, Sankey diagram in which the thickness of the lines between the left and middle columns represents the number of mutual nearest-neighbour (MNN) cells shared between each class and that between the middle and right columns reflects the initial class (IC) identity of the 100 cells with the highest (CellRank) probabilities of being absorbed in each terminal

class (TC). **c**, Summarized taxonomy of initial and terminal classes observed in macaques and mice. Forked lines represent subclasses that become apparent postmitotically. Initial classes of INs are organized by the presumptive birthplace based on the expression of regional marker genes and putative birthdates, presented in the manner of Lim et al.[5]. Inferred terminal fates are based on our gene expression and histology analysis and the literature, as denoted and discussed in detail in Supplementary Table 4. S/DWMIN, superficial/deep white matter inhibitory neuron; BN, basal nuclei. RMTW_ZIC1/RELN and VMF_TMEM163/OTP were not included because they are excitatory cortical and hypothalamic classes, respectively.

origin of this evolutionarily novel population remains unclear. These striatal INs are important exceptions to the one-to-one conservation of initial classes between primates and rodents (Fig. 2b). Instead, mice have a single ancestral class of MGE_CRABP1/MAF neurons that shows strong homology to the MGE_CRABP1/MAF and MGE_CRABP1/TAC3 clusters in macaques (Extended Data Fig. 4c). We further examined the gene networks that define this primate-specific population (Fig. 3a, b and Extended Data Fig. 6). Using RNAscope, we quantified the co-expression of the dividing cell marker *MKI67* and the initial class markers *CRABP1*, *TAC3*, *MAF* and *LHX8* across the rostrocaudal expanse of the MGE and striatum at PCD65. Our results showed a bias of MGE_CRABP1/MAF neurons rostrally and MGE_CRABP1/TAC3 neurons caudally in the MGE progenitor zone (Fig. 3c, d). In addition, we detected a low fraction of cells co-expressing *CRABP1*, *TAC3* and *MKI67* that were displaced from the ventricle, suggesting that subventricular zone (SVZ) progenitors upregulate the programme for this novel initial class at or before their final cell division (Extended Data Fig. 7). In the striatum, both classes showed uniform distributions, which was also confirmed by RNAscope for *STXBP6*, *ANGPT2* and *RBP4* in two additional individuals (Extended Data Figs. 7 and 8). *LHX8* expression was restricted to a subset of *CRABP1*⁺*TAC3*⁺ cells outside the MGE (Fig. 3d), highlighting early postmitotic specification of a *TAC3/LHX8* subclass observed in adult marmosets[2]. Interestingly, although they were clearly distinct from primate MGE_CRABP1/TAC3 neurons, mouse cholinergic and pallidal neurons (VMF_CRABP1/LHX8) also expressed *Zic1* and *Lhx8* (Fig. 3a), hinting that a combination of transcriptional programmes

used by neighbouring initial classes may define the novel *TAC3* population. Differential expression and regulon analysis showed that the earliest molecular programmes that distinguish *TAC3* INs involve distinct neuropeptides, acetylcholine receptors and immediate early gene networks (Extended Data Fig. 6 and Supplementary Table 6), suggesting that *TAC3* neurons may receive signals from nearby cholinergic neurons. Notably, the primate-specific *TAC3* population emerged as a distinct class as cells became postmitotic by PCD65. This occurred far earlier in development than the conserved *PTHLH*⁺, *PVALB*⁺ and *TH*⁺ terminal fates that ultimately arise from the related MGE_CRABP1/MAF class[2,22]. Lastly, we found that MGE_CRABP1 classes emerged in vitro as rare populations in human pluripotent stem cell-derived telencephalon organoids (Extended Data Fig. 6).

## Reuse of OB neurons in primate cerebrum

We next analysed the initial classes of neurons detected within and probably derived from the LGE[6]. Two classes, LGE_FOXP2/TSHZ1 and LGE_MEIS2/PAX6, showed unexpected enrichment in the cortical frontal lobe in addition to the ventral telencephalon (Fig. 4a, b and Extended Data Fig. 9). LGE_MEIS2/PAX6 neurons express *ETV1*, *SP8*, *MEIS2*, *SALL3*, *TSHZ1* and *PAX6* during differentiation, all of which are markers of and are required for proper production of OB granule cells and dopaminergic *TH*⁺ periglomerular cells (PGCs)[23,24]. Indeed, the transcriptomes of cells in this class showed strong correlations to mouse adult-born granule cells (OB-GC_MEIS2/PAX6; Extended Data Fig. 4c).

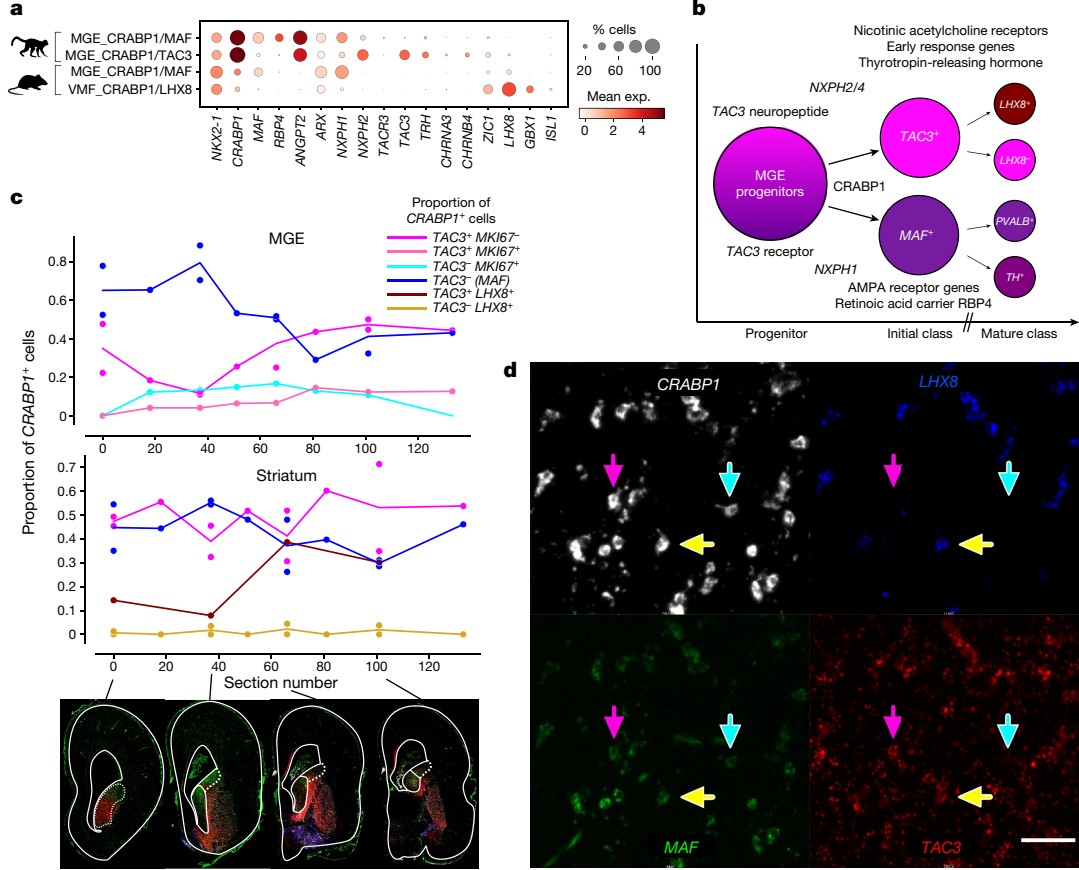

**Fig. 3 | Emergence of primate-specific MGE_CRABP1/TAC3 striatal INs.**
**a**, Dot plot of expression of striatal IN marker genes. **b**, Schematic summarizing properties distinguishing new-born MGE *CRABP1 TAC3*[+] and *CRABP1*[+]*MAF*[+] neurons, from markers given in Supplementary Table 6. **c**, Line plots showing the Rostro–Caudal distribution of classes of *CRABP1*[+] cells. *TAC3*[–] (*MAF*) denotes that the MAF class is inferred by the lack of *TAC3*, as distinct positive markers for this class are not apparent until later in differentiation. Each point is the sum of all cells in at least five random fields of view in each section/

region. Cells were counted from whole-section scans of RNAscope in situ hybridization on representative sections (full size shown in Extended Data Fig. 8). The solid (dotted) outlines of the GE region in the images represents MGE (LGE). One individual was used with four pairs of tandem sections interspersed with four single sections. **d**, Representative image of MGE_CRABP1/MAF (blue arrows), MGE_CRABP1/TAC3 (pink arrows), MGE_CRABP1/TAC3/LHX8 (yellow arrows) and VMF_CRABP1/LHX8 INs (*LHX8*[+]*TAC3*[–] cells) in the putamen from section 66 (Extended Data Fig. 8c). Scale bar 50 μm.

Similarly, trajectory analysis linked the mouse LGE_FOXP2/TSHZ1 class to OB-PGC_FOXP2/CALB1 PCGs of the OB, connecting each LGE class to distinct olfactory populations (Fig. 2b).

We performed immunofluorescence microscopy to visualize the spatial distribution of the LGE_FOXP2/TSHZ1 and LGE_MEIS2/PAX6 classes, using combinations of MEIS2 together with FOXP2/FOXP4 and SCGN/SP8/PAX6, respectively. Both populations appeared to emanate from the dorsal LGE (dLGE) but showed complementary distributions. LGE_FOXP2/TSHZ1 cells immunoreactive for FOXP2, FOXP4 and SCGN were found mainly in the dorsolateral dLGE (DL-dLGE; Extended Data Figs. 10 and 11) but were not detected in the anterior dLGE (A-dLGE) or rostral migratory stream (RMS). Instead, cells of this class migrated directly into the striatum, via the lateral migratory stream (LMS) to the outer OB and ventromedially to cortical superficial white matter (Extended Data Figs. 10–12). Consistent with trajectory analysis, markers of dLGE origin (*ETV1*, *SCGN* and *SP8*) were downregulated, whereas *FOXP2*, *CASZ1*, *OPRM1* and projection neuron markers were upregulated, as the cells differentiated and migrated into the striatum (Extended Data Figs. 9 and 10). The expression of *TSHZ1*, *LYPD1*, *PCDH8* and *CASZ1*, the absence of expression for the canonical medium spiny projection neuron markers *NPY* and *FOXP1*, and the results of RNA velocity analysis all imply that the LGE_FOXP2/TSHZ1 initial class also explains the previously unknown developmental origin of recently described striatal projection neurons in adult mice, eccentric spiny projection

neurons (eSPNs)[21] and amygdala ITCs (Fig. 2 and Extended Data Fig. 9). This linkage is consistent with reports that cells in mouse dLGE initially express SP8 and maintain TSHZ1 expression as they migrate via the LMS to become amygdala ITCs[19,25]. This developmental perspective suggests that these cells are not eccentric deviations from canonical spiny projection neuron development; instead, the LGE_FOXP2/TSHZ1 class converges on a similar striatal and amygdaloid projection neuron transcription profile despite its distinct origin.

By contrast, we observed *MEIS2*[+]*PAX6*[+]*SP8*[+]*SCGN*[+] cells representing the LGE_MEIS2/PAX6 class continuously from the anterior end of the dLGE along the RMS to the OB granule cell layer (Fig. 4c and Extended Data Figs. 11 and 12). Notably, we observed dense parenchymal chains[26] of these cells radiating from the dLGE at PCD80 (*n* = 3 hemispheres; Fig. 4c and Extended Data Fig. 12h, i). At PCD120, we found large numbers of LGE_MEIS2/PAX6 precursors that express SCGN extending dorsomedially and caudally in the Arc migratory stream[27] in addition to the RMS (Extended Data Fig. 13). These cells were densest in chains running along the entire striatum in the primary tier of the Arc with fewer cells radially[27]. Unexpectedly, we also observed a robust stream diverted from the Arc that stretched from the A-dLGE into the anterior cingulate cortex (ACC; Fig. 4d, e). This stream, referred to as the Arc–ACC, appeared to be bounded by TH[+] fibres in superficial white matter (Fig. 4d). Cells from the Arc and Arc–ACC were common in dorsomedial cortex deep white matter but were rarely found lateral or ventral to the

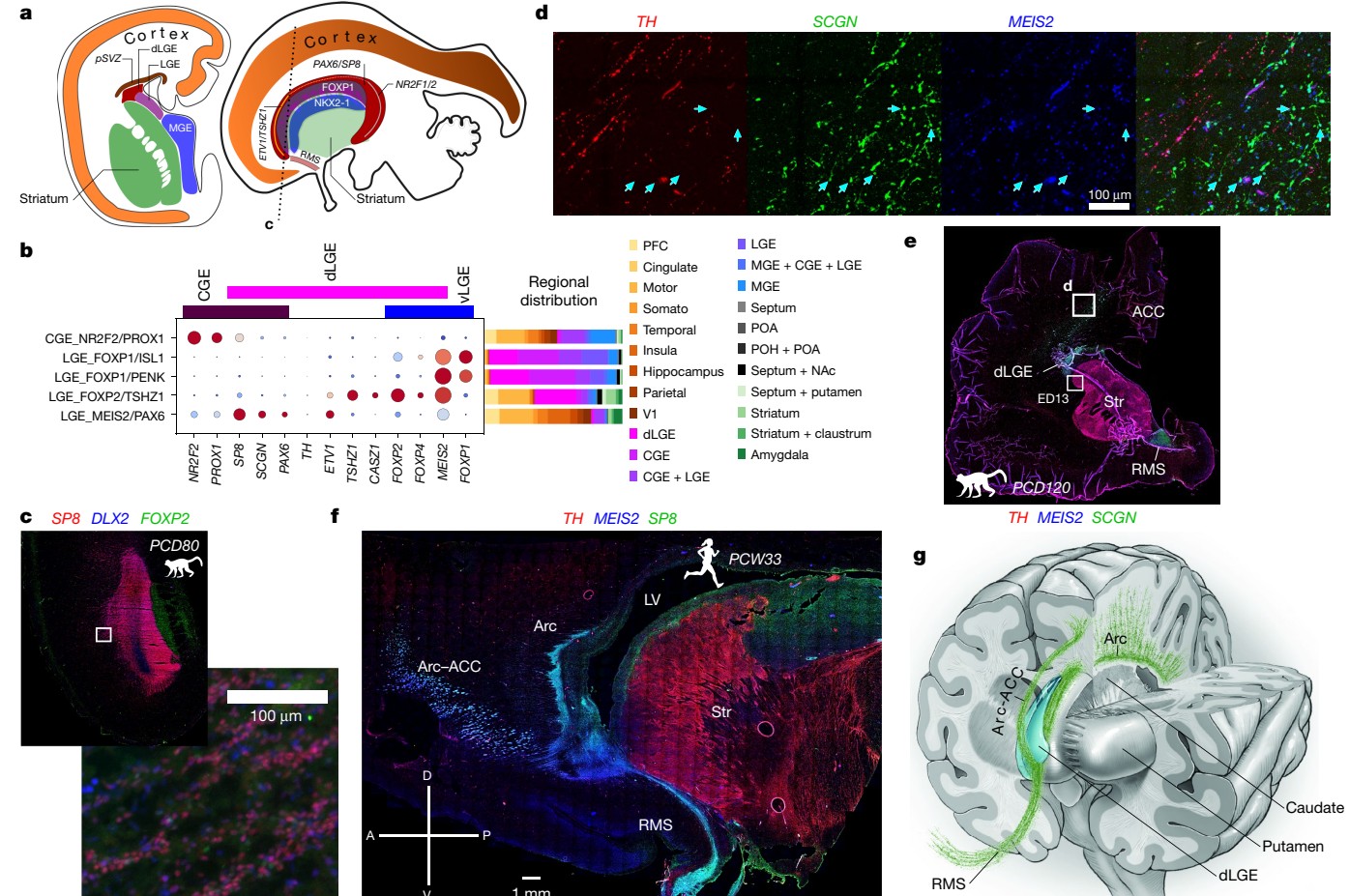

**Fig. 4 | Redistribution of LGE_MEIS2/PAX6 granule cells. a**, Approximate ganglionic eminence transcription factor territories in PCD80 macaque brain, showing estimated section planes. **b**, Dot plot of expression of markers of LGE- and CGE-derived classes showing overlap of transcription factor domains. The expression values are scaled from 0–1 for each gene. The dot size represents the percentage of cells that express each gene. Stacked bar plots show the regional distribution of each class. NAc, nucleus accumbens. **c**–**f**, Immunohistochemistry for LGE class markers in macaque and human. **c**, Coronal–axial section of the A-dLGE and PFC at PCD80 with the inset highlighting DLX2+SP8+FOXP2− parenchymal chains. **d**, Arc–ACC SCGN+MEIS2+

cells (teal arrows) shown migrating within the boundary of dense TH+ axons at PCD120. **e**, Whole PCD120 coronal section from **d** showing the Arc–ACC and RMS wrapping around the striatum from the A-dLGE. Box marked ED13 corresponds to Extended Data Fig. 13f. **f**, Low-magnification sagittal image of a human postconception week 33 (PCW33) specimen showing large streams of *MEIS2+SP8+* neurons originating from the A-dLGE; these neurons contribute to the RMS and the Arc. LV, left ventricle; Str, striatum. **g**, Schematic of a macaque brain. Multiple streams extend from the anterior pole of the dLGE, or the SVZ at later stages, to the RMS leading to the OB, the Arc extending dorsomedially and the Arc–ACC subsidiary stream extending to the ACC.

striatum; however, many CGE-derived MEIS2−SP8+NR2F2+ neurons were observed throughout the white matter (Extended Data Fig. 13), highlighting regional heterogeneity in the composition of white matter INs.

We confirmed that LGE_MEIS2/PAX6 neurons from the A-dLGE also contribute to the RMS, Arc and Arc–ACC in perinatal humans (Fig. 4f) and postnatal macaques (Extended Data Fig. 13). We further found that these neurons persist postnatally in the deep white matter of the cingulate cortices and the superior corona radiata (Extended Data Fig. 13l, m). By contrast, in postnatal day 2 (P2) mice, we identified only rare instances of LGE_MEIS2/PAX6 cells in deep white matter (Extended Data Fig. 14a–c), consistent with recent reports that sparse MEIS2+HTR3A+ neurons in the white matter integrate into cortical circuitry perinatally[28]. Instead, the vast majority of these cells appeared in the anterior SVZ and RMS in mice (Extended Data Fig. 14). Overall, we found that neurons derived from the dLGE are more widely distributed than previously recognized in primates, representing a major source of neurons in the primate Arc migratory streams and persisting in the deep white matter.

Our analysis identified a third presumed dLGE-derived class in and around the striatum, insula and claustrum, which we refer to as striatum laureatum neurons (SLNs or Str-SLN_TH/SCGN). Likely derived

from the LGE_MEIS2/PAX6 initial class, SLNs are named for the wreath shape they form around the striatum. At both PCD120 and 7 months postnatally, SLNs were immunoreactive for PAX6, MEIS2, SP8, TH and SCGN but not for FOXP2, NKX2-1 or NR2F2, which is also characteristic of TH+ PCGs (OB-PGC_TH/SCGN) of the OB (Fig. 5a–d and Extended Data Figs. 13f and 15). This distribution matches observations of TH+ cells circumscribing the primate striatum and their reported absence in rodents and illuminates their molecular identity and origin[29]. Indeed, we did not identify MEIS2+PAX6+SCGN+TH+ cells along the mouse striatum border or the claustrum (Extended Data Fig. 14e–g). Instead, these cells in mice were restricted to the OB, olfactory tract or olfactory tubercle, matching the macaque olfactory peduncle domain (Fig. 5e, f). We found that SLNs form a reticule at the white matter boundaries of the caudate and putamen of macaques, exist in humans and persist throughout life (Fig. 5a–h).

## Discussion

By identifying transcriptional regulatory programmes distinguishing the earliest specification of initial classes, our study provides

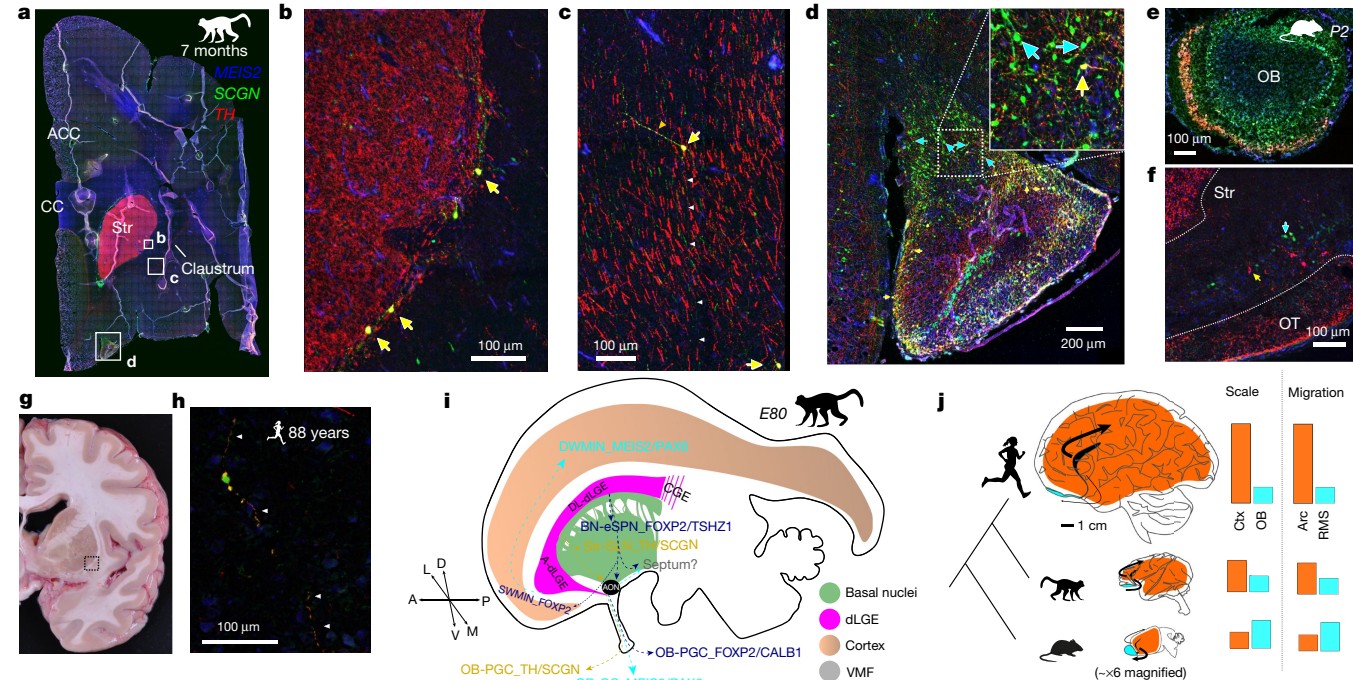

**Fig. 5 | Primate *TH*⁺ SLNs and ancestral olfactory populations.** All immunofluorescence images in this figure are of MEIS2, SCGN and TH. **a**, Seven-month-postnatal macaque coronal section including the remnant RMS. **b**, TH⁺ SLNs at the border of the striatum. MEIS2⁺SCGN⁺TH⁺ peristriatal SLNs are indicated by yellow arrows. **c**, SLNs in the claustrum with long, straight processes among dense TH⁺ midbrain–cortical fibre synapses including one TH⁺ process (orange arrowheads) and one TH⁻ process (white arrowheads). **d**, Anterior olfactory nucleus at the olfactory peduncle, with MEIS2⁺SCGN⁺TH⁻ cells (blue arrows) including SCGN⁺TH⁻ fibres entering the ventral cortex (inset) and triple-positive cells (yellow arrows). **e**, Coronal section of mouse P2 OB showing SLN-analogous TH⁺SCGN⁺ cells. **f**, Mouse P2 coronal section showing the olfactory tubercle (OT) and striatum (Str) outlined by dotted lines, with only MEIS2⁺TH⁺ or MEIS2⁺SCGN⁺ cells. **g**, Photograph of a brain coronal slab from an 88-year-old human. The box shows the approximate location of the inset section block. **h**, TH⁺SCGN⁺ human SLN, with double-positive processes highlighted with white arrowheads. Only two cells were observed across this section. **i**, Schematic summarizing the unequal scaling of cortical and olfactory structures. **j**, Schematic summarizing migration of dLGE neurons and unequal evolutionary scaling of their destinations. Relative values in the bar plots are arbitrary.

a resource for identifying conserved molecular mechanisms that specify cell type diversity. This resource can support rational in vitro derivation of these populations from pluripotent stem cells and interpretation of the cellular substrates of genetic disorders of neural development. *TAC3*-expressing striatal INs represent an exceptional case in which an evolutionarily novel initial class of neurons emerges in differentiating progenitors. A limited number of gene networks distinguish the *TAC3* initial class from the related MGE_CRABP1/MAF class, consistent with a recent model of cell type evolution. Under this model, an ancestral cell type is partitioned into distinct subtypes by changes in transcription factor expression that enable genomic individuation of sister cell classes that still share many regulatory complexes and developmental trajectories[30]. However, the conservation of nearly all other initial classes of INs between macaques and mice suggests that evolutionary diversification of primate INs arises mainly by radiation of conserved initial classes of new-born neurons and may be shaped by the expanded diversity of primate regional destinations[7].

The neurons of the dLGE appeared to be particularly affected by primate brain reorganization. In both macaques and mice, the LGE_MEIS2/PAX6 class is among the latest-born INs and migrates to olfactory structures and deep white matter. However, the absolute migration distance of late-born A-dLGE neurons to the OB is more than two orders of magnitude longer in new-born macaques than in mice and increases further as the brain expands after birth[27,31]. Similarly, the volume of white matter is more than three and five orders of magnitude larger in macaques and humans than in mice, respectively[32], whereas the relative size of the primate OB is markedly smaller (Fig. 5i, j)[33]. Thus, in mice, the birthplace is only several cell lengths

from any point in the adjacent deep white matter. In macaques, however, these homologous cells traverse histologically distinct dorsal migratory streams and apparently reuse the chain migration strategy. OB granule cells derived from this class contribute to adult plasticity[34], and myelination is delayed for up to two decades in human frontal lobe white matter[35], potentially linking these cells to white matter plasticity. Notably, abnormal accumulations of frontal lobe white matter neurons have been reproducibly associated with schizophrenia and autism[36]. With their prolonged migration to far-flung and ever-changing destinations, the A-dLGE neurons we identified here may be particularly vulnerable to environmental influences, and the markers we identified will be useful for assessing the molecular heterogeneity of disease-associated populations.

Finally, we identified SLNs, another likely OB sister type, which are redistributed to peristriatal regions and show a molecular resemblance to dopaminergic OB TH⁺ PGCs. Future studies can examine whether this primate striatal population partly explains the human-specific increase in TH-expressing striatal neurons[37,38] and whether these neurons produce dopamine themselves or have an auxiliary role to compensate for increased demands on midbrain dopaminergic neurons[39,40]. Crick and Koch[41] speculated that, in the claustrum, hitherto undiscovered sparse INs resembling intraglomerular OB cells with dendrodendritic synapses could contribute to binding information. Molecular access to SLNs will enable future circuit-level studies of this rare claustrum population. Together, our results highlight contrasting models for diversification of primate INs by specification of an entirely novel initial class and by redistribution of conserved initial classes that supply the OB into primate white matter migratory streams and peristriatal locations.

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

## Methods

### Samples

The Primate Center at the University of California, Davis, provided nine specimens of cortical tissue from PCD40, PCD50, PCD65 (*n* = 3), PCD80 (*n* = 2), PCD90 and PCD100 macaques. All animal procedures conformed to the requirements of the Animal Welfare Act, and protocols were approved before implementation by the Institutional Animal Care and Use Committee at the University of California, Davis. PCD40 represents embryonic Carnegie stage 20 and marks the approximate beginning of neurogenesis of both excitatory neurons and INs, whereas PCD100 is the approximate end of excitatory neurogenesis in the cortex[42]. Macaque data from PCD110 (*n* = 2) were taken from ref. [9]. In addition, we used public mouse datasets, including for embryonic day 13.5 (E13.5) and E14.5 ganglionic eminences, which were enriched for DLX6+ cells[10], and three samples from the 10x Genomics E18 mouse cortex example dataset with 1.3 million cells (GSE93421; samples 1, 3 and 4). We also used mouse public datasets for E14, including neonatal cortex, subcortex[11] and whole-brain developmental structures as well as adult structures[43,44]; P9 striatum[45]; and adult OB[12]. In total, we analysed single-cell transcriptomes from 109,112 cells from developing macaque, 76,828 cells from developing mouse and 141,065 total mouse cells. De-identified human tissue samples were collected with previous patient consent in strict observance of legal and institutional ethical regulations in accordance with the Declaration of Helsinki. Protocols were approved by the Human Gamete, Embryo, and Stem Cell Research Committee and the Committee on Human Research (institutional review board) at the University of California, San Francisco.

### Single-cell RNA sequencing tissue processing

For the PCD40 to PCD100 macaques, dissections were performed in PBS under a stereo dissection microscope (Olympus SZ61). A number of regions were difficult to distinguish at earlier time points because key anatomical landmarks were still forming. Accordingly, presumptive regions were dissected such as motor versus somatosensory cortex before the appearance of the central sulcus or the anterior ends of the MGE and LGE. For single-cell dissociation, samples were cut into small pieces and incubated with a prewarmed solution of papain (Worthington Biochemical Corporation) prepared according to the manufacturer's instructions for 10 min at 37 °C. After 30–60 min of incubation, samples were gently triturated with glass pipette tips, and the PCD100 macaque samples were further spun through an ovomucoid gradient to remove debris. Cells were then pelleted at 300*g* and resuspended in PBS supplemented with 0.1% BSA (Sigma). Samples for MULTIseq were prepared in strip tubes and were maintained at 4 °C for the labelling protocol, as described in McGinnis et al.[46]. Single-cell RNA sequencing was completed using the 10x Genomics Chromium controller and version 2 or 3 3-prime RNA capture kits. Most samples were loaded at approximately 10,000 cells per well; up to 25,000 cells were loaded per lane for multiplexed samples. Transcriptome library preparation was completed using the associated 10x Genomics RNA library preparation kit. Multiseq barcode library preparation was completed as described in McGinnis et al.[46]. Following library preparation, libraries were sequenced on Illumina HiSeq and NovaSeq platforms.

### Alignments and gene models

Fastq files were generated from Illumina BCL files using bcl2fastq2. Genes were quantified using Kallisto release 0.46 (ref. [47]) and the RheMac10 genome assembly, newly annotated using the comparative annotation toolkit[48], as well as the transcript annotations of *Mus musculus* ENSEMBL release 100. A custom Kallisto reference for each species was created for the quantification of exons and introns together, in which introns were defined as the complement of exonic and intergenic space. The Kallisto index used *k*-mers of length 31. Public data were downloaded as raw fastq files or as BAM files that were converted back to fastq files. All data were processed from raw reads using the same Kallisto pipeline to minimize annotation and alignment artefacts.

### Quality control

Kallisto–Bus output matrix files (including both introns and exons) were input to Cellbender (release 0.2.0; https://github.com/broadinstitute/CellBender), which was used to remove probable ambient RNA only. Only droplets with a greater than 0.99 probability of being cells (not ambient RNA), as calculated by the Cellbender model, were included in further analysis. Droplets with fewer than 800 genes detected, or greater than 40% ribosomal or 15% mitochondrial reads, were filtered from the dataset. Doublets were then detected and removed from the dataset using Scrublet (release 0.2.2; using threshold parameter 0.5).

### Clustering and determination of homologous cell types

Much of the analysis pipeline was based on scanpy infrastructure and AnnData data structures[49]. Counts in cells were normalized by read depth, log transformed and then scaled for each gene across all cells. Principal-component analysis was then performed using the top 12,000 most variable genes by applying the original Seurat variable gene selection method implemented in the scanpy package, with the 100 most variance-encompassing principal components used for the following steps. Batch correction was limited to the requirement that highly variable genes be variable in more than one sequencing sample and by application of batch-balanced *k*-nearest neighbours (BBKNN)[50] using the Euclidean distance of principal components to find 3 neighbours per batch in the developmental data and 12 neighbours per dataset in the developmental and adult merged mouse data. Leiden clustering using BBKNN-derived *k*-nearest-neighbour graphs was then applied according to the KNN graph with the scanpy resolution parameter set to 10 (or 7 in the developmental mouse dataset). Glia, along with excitatory progenitor and neuron clusters, were removed from the dataset in non-ganglionic eminence batches if their expression value was below the mean for two or more of the following genes: *GAD1*, *GAD2*, *DLX1*, *DLX2*, *DLX5* and *DLX6*. Cajal–Retzius cells (RMTW_ZIC1/RELN) met this threshold and served as a useful outgroup. These cells were considered to be derived from the rostromedial telencephalic wall (RMTW) on the basis of *ZIC1* and *RELN* expression even though they are known to have multiple origins[51]. After non-INs were removed, scaling, principal-component analysis and the following steps were repeated with this final IN dataset.

High-resolution Leiden clusters partitioned continuous differentiation trajectories of postmitotic initial classes into subclusters based on maturation stage. These high-resolution clusters were then manually merged to initial classes by using hierarchical clustering of cluster gene expression averages and distinctness of individual Leiden cluster markers as a guide. The nomenclature for merged clusters incorporates the presumptive spatial origin of the initial classes and specific marker genes. Spatial origin for each class was inferred according to the expression of canonical marker genes for the RMTW, MGE, LGE, CGE and VMF, such as *LHX5*, *NKX2.1*, *MEIS2*, *NR2F2* and *ZIC1* and was supported by immunostaining and the enrichment of these genes in cells from region-specific dissections. For merged species analysis, genes were normalized and scaled within species and were then merged for downstream analysis using BBKNN with 25 neighbours across and within species, the mutual nearest neighbours of which were used for Sankey plot comparison of developing macaques and mice. After clustering, the mean expression in each class was calculated for each gene that was among the original 12,000 most variable one-to-one orthologues from each dataset showing variability in both species (6,227 genes). These classes were then compared across species by Pearson correlation of their gene expression vectors.

### Trajectory analysis of activating and inactivating macaque genes

We applied scVelo's dynamical model (release 0.2.3)[52] to derive a shared latent time based on RNA velocity using spliced and unspliced counts

from Kallisto. Next, we used the related CellRank (release 1.3.1) package[15] to derive absorption probabilities for immature cells in the transition cluster to likely initial classes. This step was necessary because new-born neurons, like children, are more similar to each other than to their mature state. By using adjacency along the paths of differentiation, it is possible to infer which mature state is likely to absorb a given immature cell. We then classified the new-born neurons as cells below the 0.5 quantile of latent time for that class. Recent studies have indicated that these transcriptionally immature neurons correspond to new-born neurons as labelled by classical nucleoside-based methods[53]. To identify genes that were activated or inactivated along trajectories, we used linear regression implemented in SciPy based on latent time values ($x$) versus gene expression values ($y$). This yielded linear regression coefficients and two-tailed $P$ values for each gene, which were corrected for multiple-hypothesis testing using the Holm–Sidak method implemented in the statsmodels (release 0.12.2) package to derive $q$ values. The gene sets were compared by calculating the Jaccard indices of set intersections, defined as the number of intersecting elements between two sets divided by the number of elements in the union of the two sets.

### Linking developmental and adult data

Similarly to the reassignment of macaque transition cells, we also used CellRank-derived absorption probabilities, with equally weighted KNN and RNA velocity kernels, to estimate the precursor states of adult cells. Because absorption probabilities are biased by cell numbers in terminal states, and the goal this time was not to assign each developmental cell to a terminal state, we subsampled a maximum of 1,000 cells per class, with the rarest class having 707 cells from MGE_CRABP1/MAF, and we report the class identity of the 100 developing cells with the highest probability of being absorbed into each terminal class. This enabled us to provide an estimate of which developing class was the likely origin of the terminal classes, which is reflected in the weights of the edges in the Sankey diagram in Fig. 2. We also calculated the mean absorption probability for cells in each initial class to each terminal state to alleviate compositional effects, which we present as a heatmap. Notably, RMTW_ZIC1/RELN and VMF_TMEM163/OTP were not included because they are excitatory cortical and hypothalamic classes, respectively.

### Immunohistochemistry tissue processing and imaging

Mouse, macaque and human tissues for histology were fixed in 4% paraformaldehyde in PBS overnight at 4 °C with constant agitation. The paraformaldehyde was then replaced with fresh PBS (pH 7.4) and samples were cryopreserved by incubation for 24–48 h in 30% sucrose diluted in PBS (pH 7.4) before being embedded in a mixture of OCT (Tissue-Tek, VWR) and 30% sucrose. Tissue was then frozen at −80 °C and was cryosectioned at 16–20 µm. For RNAscope RNA in situ hybridization, fixed cryosections were stained according to the protocol for the Advanced Cell Diagnostics RNAscope Multiplex Fluorescent Reagent Kit V2 Assay (ACD, 323120). For immunostaining, antigen retrieval was performed by placing tissue slides in 95 °C citrate buffer and then allowing them to cool at room temperature. Antibodies were diluted in blocking buffer (0.1% Triton X-100, 5% donkey serum and 0.2% gelatine in PBS). Sections were incubated with primary antibodies overnight at room temperature under bright light to photobleach autofluorescence in a light box[54]. The primary antibodies and dilutions used are recorded in Supplementary Table 7.

Alexa dye-conjugated donkey secondary antibodies were incubated in the dark at room temperature for 1 h. All tiled scans were acquired using an Evos M7000 microscope. All images were stitched using a custom Python script and ImageJ's max correlation grid/collection stitching (release 1.2). They were then processed using ImageJ (release 1.53c) Rolling Ball background subtraction and manual brightness/contrast adjustment within an ImageJ macro. Image quantification of CRABP1+ cells was conducted using a custom ImageJ macro with the CRABP1+ area automatically thresholded using maximum entropy. Positivity for

other gene products was classified manually for every cell in at least five random areas in the striatum or MGE and was defined as >1 puncta within CRABP1+ areas not clearly belonging to another cell.

### Reporting summary

Further information on research design is available in the Nature Research Reporting Summary linked to this paper.

### Data availability

The sequencing data have been deposited in the Gene Expression Omnibus under accession number GSE169122; the data are browsable at https://dev-inhibitory-neurons.cells.ucsc.edu/. Scripts and annotation files for the study have been deposited on github at https://github.com/mtvector/dev-and-evo-of-primate-inhibitory-neurons.

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

**Acknowledgements** We thank A. Bhaduri, A. Alvarez-Buylla, J. Rubenstein, C. Harwell, M. Turrero Garcia, M. Andrews, S. Nolbrant, J. Wallace, S. Salama, I. Fiddes and members of the Pollen and Ye laboratories for helpful discussion and A. Li, S. Gaus, S. Wang, B. Alvarado, A. Tarantal and A. Fox for providing samples for genomic and histological analyses, and K. Probst for drawing the artwork in Fig. 4g. This study was supported by NIH awards F31NS124333 (M.T.S.), DP2MH122400-01 (A.A.P.), U01MH114825 (A.A.P., T.J.N.), R01AI136972 (C.J.Y.) and DP2NS122550-01 (M.F.P.); the Roberta and Oscar Gregory Endowment (M.F.P.); the Schmidt Futures Foundation (A.A.P., T.J.N.); the Chan Zuckerberg Biohub (C.J.Y., A.A.P., T.J.N.); and a grant from the Shurl and Kay Curci Foundation to the Innovative Genomics Institute (A.A.P.). A.A.P is a New York Stem Cell Foundation Robertson Investigator and member of the UCSF Kavli Institute for Fundamental Neuroscience.

**Author contributions** M.T.S., T.J.N., M.F.P. and A.A.P. designed the methodology. M.T.S. K.S., C.P.C., M.A.M., T.J.N. and A.A.P. performed the experiments. W.W.S., M.F.P., T.J.N. and A.A.P. acquired the analysis resources. M.T.S., C.P.C. and K.S. conducted the microscopic analysis. M.T.S., M.A.M., T.J.N. and A.A.P. performed the single-cell RNA sequencing processing. M.T.S. implemented the software and analysed the study data. M.T.S. and A.A.P. wrote the manuscript with input from all authors. M.T.S., T.J.N., M.F.P., C.J.Y. and A.A.P. acquired the funding. M.T.S., M.F.P. and A.A.P. conceptualized the study. T.J.N., C.J.Y., M.F.P. and A.A.P. supervised the project.

**Competing interests** C.J.Y. is a scientific advisory board member for and holds equity in Related Sciences and ImmunAI, is a consultant for and holds equity in Maze Therapeutics, and is a consultant for TReX Bio. C.J.Y. has received research support from the Chan Zuckerberg Initiative, the Chan Zuckerberg Biohub and Genentech.

**Additional information**

**Correspondence and requests for materials** should be addressed to Alex A. Pollen.

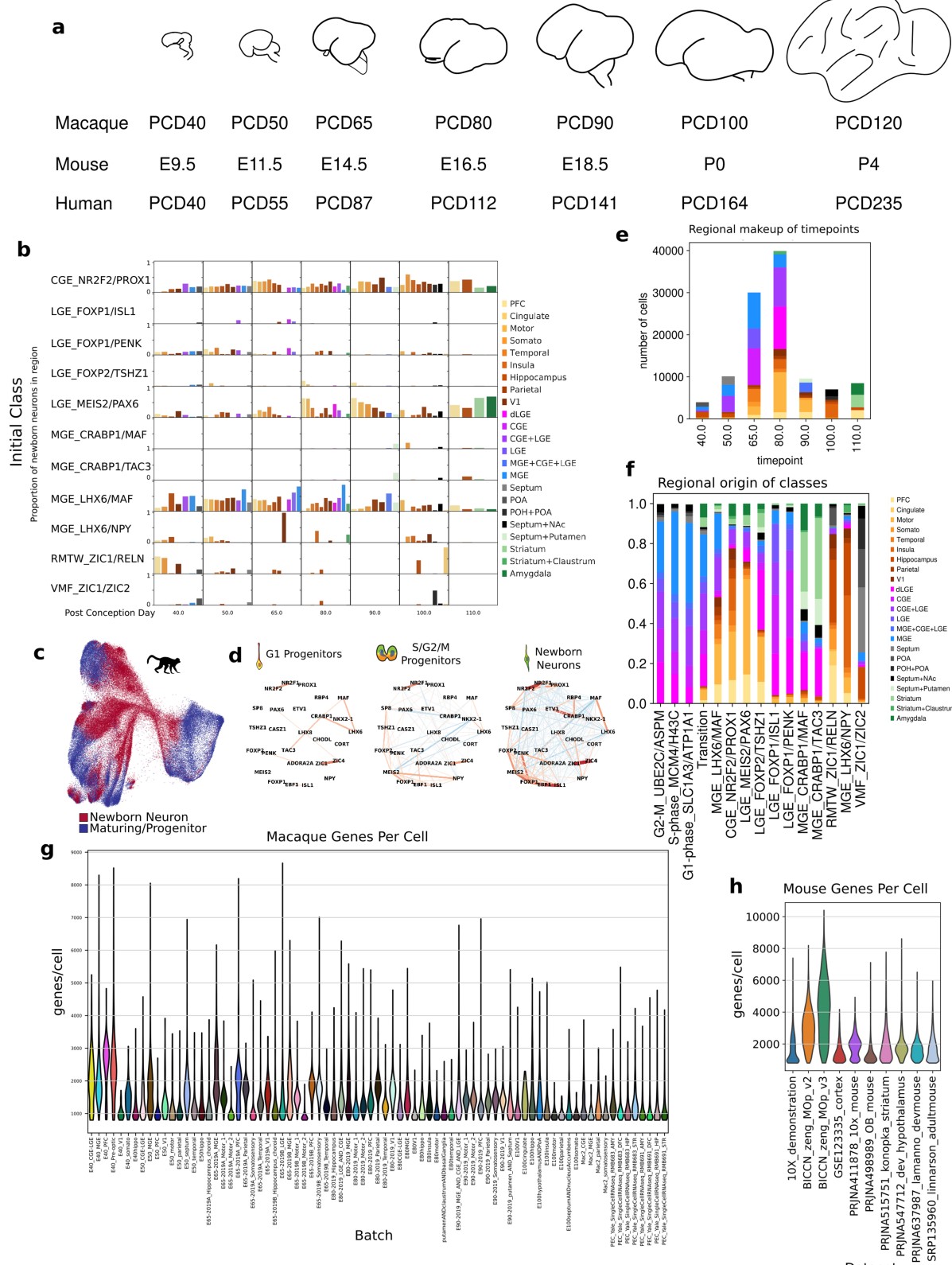

**Extended Data Fig. 1** | See next page for caption.

**Extended Data Fig. 1 | Birthdates of initial classes of INs in macaque.**
**a**. Drawings of the lateral view of developing macaque brains across stages surveyed in this study and estimated comparable human and mouse stages based on the translating time model[42] of cortical neurogenesis. **b**. Spatiotemporal distribution of new-born neurons, as determined by RNA velocity latent time, from each class as a proxy for birthdate. Bars represent the proportion of cells from each class in each region, at each timepoint (columns sum to 1). Cajal-Retzius neurons, MGE-derived cortical interneurons, and LGE-derived projection neurons first appeared early in development, starting at PCD40, followed by the later appearance of immature CGE-derived cortical interneurons and LGE_MEIS2/PAX6 neurons, consistent with broad patterns of temporal ordering in mouse[51,55,56]. **c**. UMAP showing which cells are new-born in red (latent time < 0.5 quantile of latent time for each class) or maturing (>0.5 quantile) in blue. Clearly cycling progenitors are not included. **d**. Kamada-Kawai graph visualization of Pearson correlations between a gene pair's expression highlight the emergence of initial class gene co-expression patterns during macaque neuronal differentiation. Edges shown are Holm-Šídák corrected q value < 0.05 calculated by bootstrap, with thickness and color representing correlation. **e**. Stacked bar chart showing number of macaque cells collected at each timepoint, colored by region from which the cells are derived. Note undersampling of VMF structures between PCD50 and PCD100. **f**. Normalized stacked bar chart showing regions from which each macaque initial class is derived across the whole dataset. **g**. Violin plot showing the distribution of genes detected per cell for each macaque batch. **h**. Violin plot showing the distribution genes per cell detected for each mouse dataset.

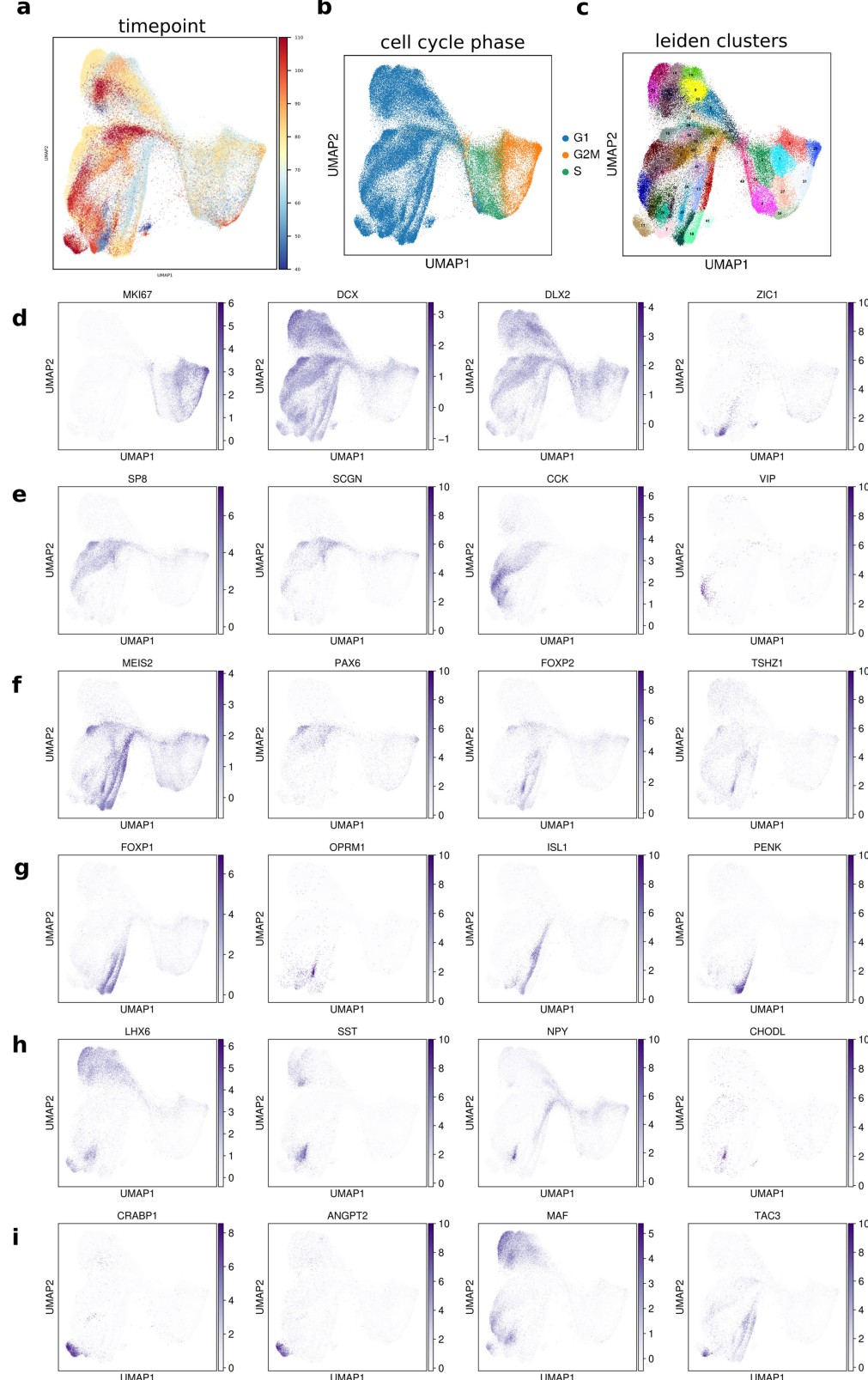

**Extended Data Fig. 2 | Macaque single cell RNAseq gene expression landscape. a**. Macaque scRNAseq UMAP colored by post conception day from which cells are sampled. **b**. UMAP projection colored by cell-cycle phase as classified by scanpy score_genes_cell_cycle function. **c**. UMAP projection colored by Leiden clusters. Although cell intrinsic differences within initial classes may predict their further subclass partitioning, the fine-grained Leiden clusters did not yield groups appearing to match terminal classes and mainly varied by neuronal differentiation trajectories. **d**–**i**. Scaled and normalized expression of **d**, dividing and new-born neuron marker genes **e**, CGE-derived neuron markers **f**, dLGE-derived neuron markers **g**, LGE-derived projection neuron markers **h**, MGE-derived cortical neuron markers **i**, MGE-derived striatal interneuron markers. An interactive browser for exploring the transcriptional features of inhibitory neuron development is available (https://dev-inhibitory-neurons.cells.ucsc.edu/)[57].

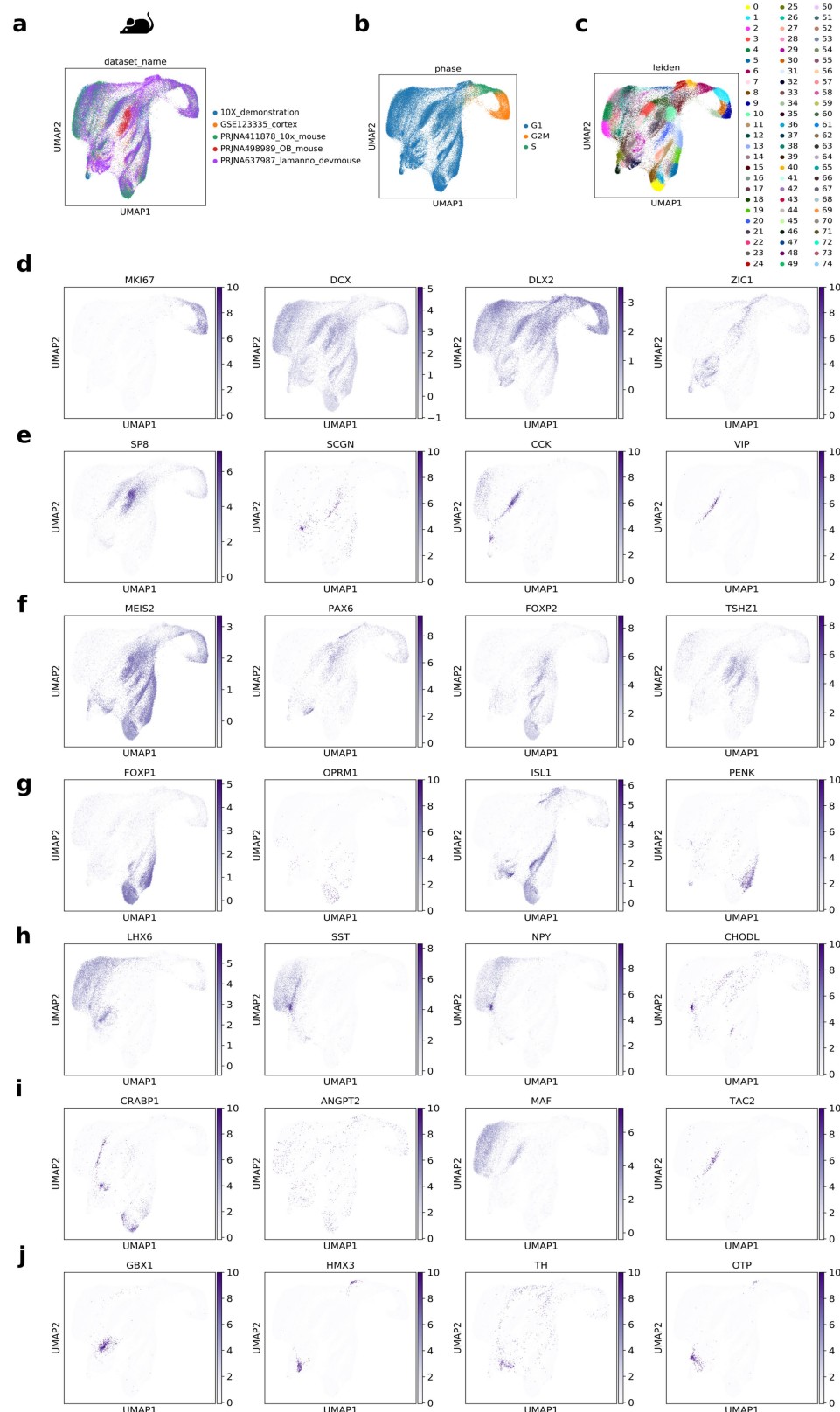

**Extended Data Fig. 3 | Mouse single cell RNAseq gene expression landscape. a**. Mouse scRNAseq UMAP colored by public dataset from which the cells are derived. **b**. UMAP projection colored by cell-cycle phase as classified by scanpy score_genes_cell_cycle function. **c**. UMAP projection colored by Leiden clusters. **d–i** Scaled and normalized expression of **d**, dividing and new-born neuron marker genes **e**, CGE-derived neuron markers **f**, dLGE-derived neuron markers **g**, LGE-derived projection neuron markers **h**, MGE-derived cortical neuron markers **i**, MGE-derived striatal interneuron markers **i**, VMF-derived markers.

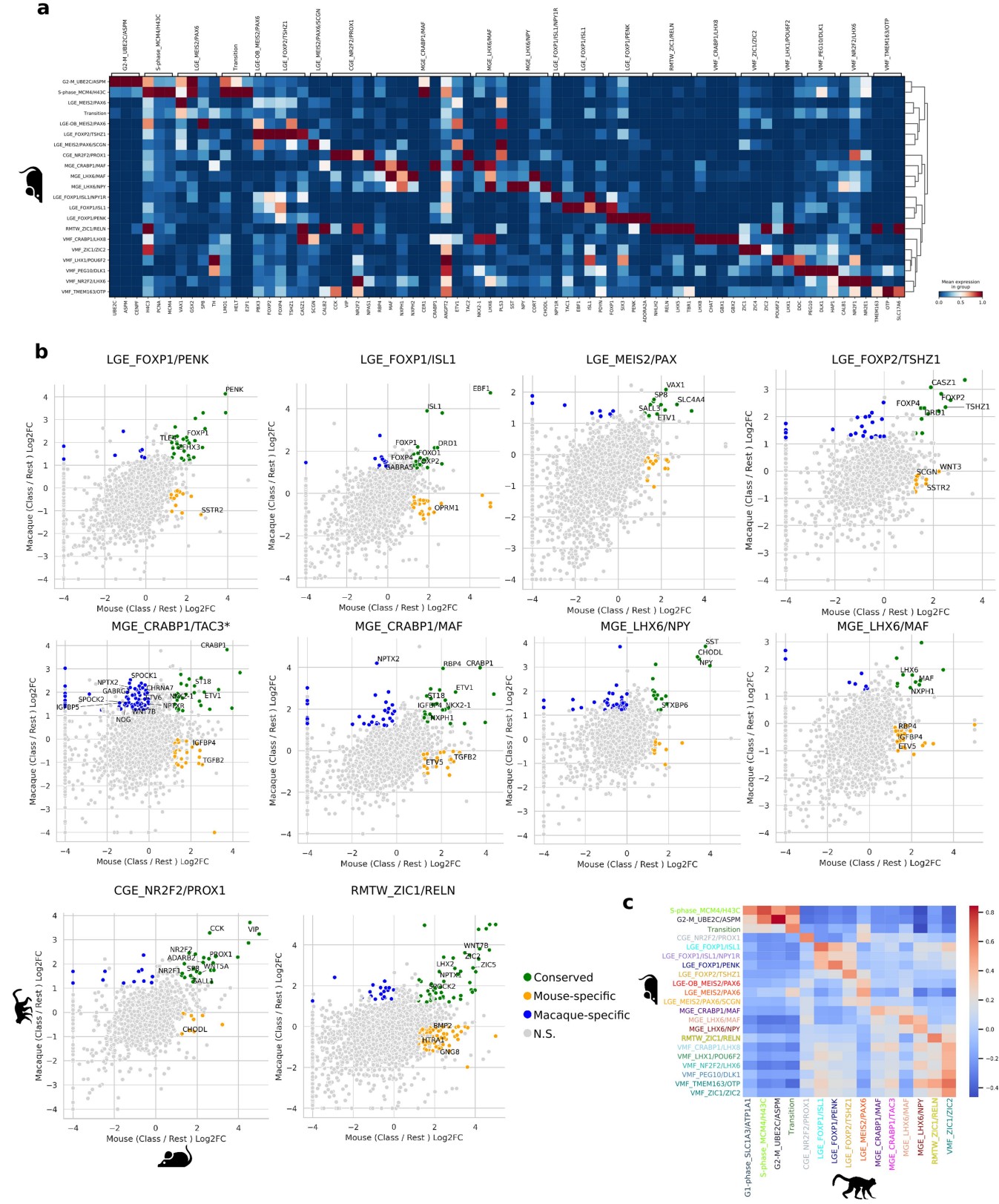

**Extended Data Fig. 4 | Markers of mouse and macaque initial classes.**
**a.** Heatmap of mouse initial class marker genes selected from the top markers of each, scaled by column. **b.** Scatterplots of log 2 fold changes of each initial class vs the rest of the dataset for mouse vs macaque, with selected gene families of interest labeled. Conserved represents genes significantly upregulated in class vs rest in both species, Mouse-specific in mouse but not macaque, etc. N.S.= Not Significant in either species. Significance defined as |log2fc| > 1.2 and adjusted p value < .01, with significance marked "signif" in Supplementary Table 3. MGE_CRABP1/TAC3* is the comparison of macaque MGE_CRABP1/TAC3 vs mouse MGE_CRABP1/MAF, as this is the ancestral class comparison. Note that more genes show specific correlations to the macaque MGE_CRABP1/TAC3 class versus the macaque MGE_CRABP1/MAF class in the comparison to the single mouse MGE_CRABP1/MAF class. **c.** Pairwise Pearson correlations of mean gene expression in classes across species.

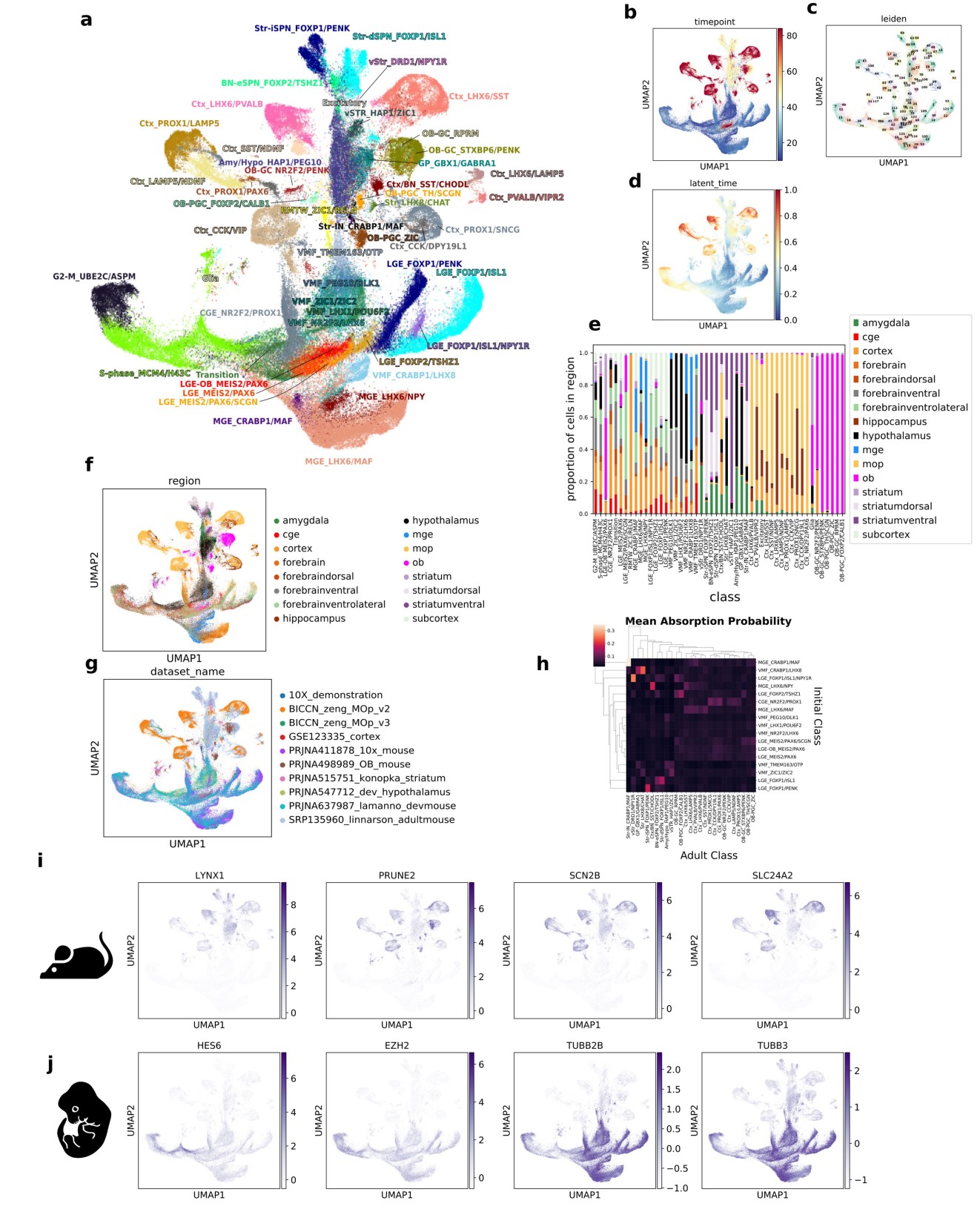

**Extended Data Fig. 5 | Inhibitory neurons of the developing and adult mouse forebrain. a**. UMAP projection of developing and adult mouse single cell RNAseq data, with initial and terminal classes labeled. **b**. UMAP of all mouse data, labeled by the post conception timepoint from which the cells are derived. **c**. UMAP of all mouse data, labeled by Leiden clusters used to determine terminal classes. **d**. UMAP of all mouse data, labeled by the scVelo dynamical shared latent time of each cell. **e**. Normalized stacked barplot, showing proportion of total cells of each class from each region. **f**. UMAP of all mouse data, labeled by the region from which the cells are derived. **g**. UMAP of all mouse data, labeled by the public dataset from which cells are derived. **h**. Heatmap representing the mean absorption probabilities of cells in each initial class to each terminal class. **i**. Selected genes differentially expressed in all terminal classes over all initial classes. **j**. Selected genes differentially expressed in initial classes over terminal classes.

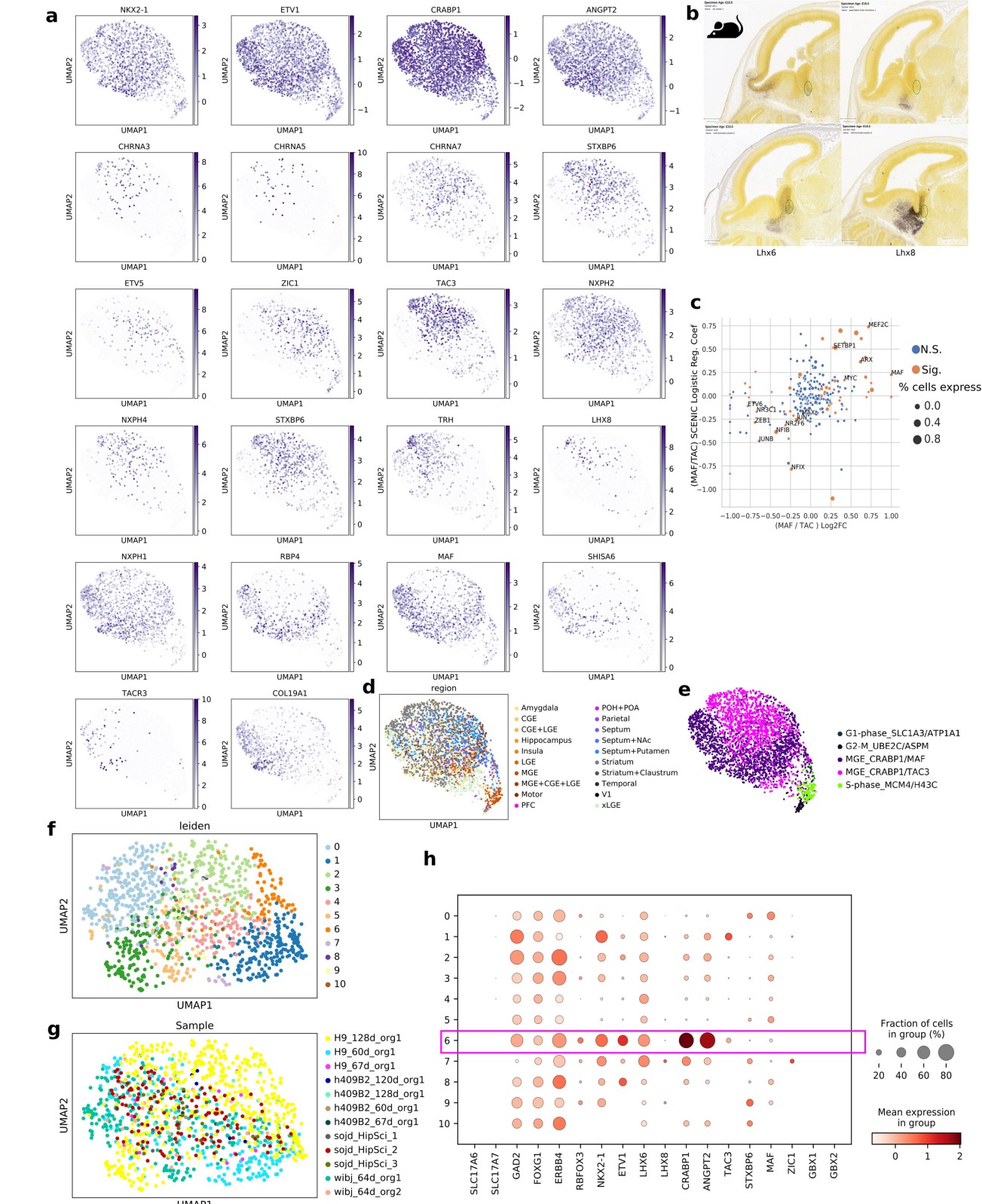

**Extended Data Fig. 6** | See next page for caption.

**Extended Data Fig. 6 | Expression of CRABP1+/TAC3+ and MAF+ striatal interneuron markers in developing macaque. a**. UMAP projection of NKX2-1/ CRABP1+/ETV1+ cells only colored by scaled and normalized expression of TAC3 or MAF class markers. **b**. Allen Institute E15.5 mouse brain in situ hybridization showing expression of CRABP1+ neuron-related regional transcription factors. Green circle denotes boundary between *Lhx8*+ MGE and rostroventral MGE/septum known to produce cholinergic neurons and the *Etv1*+ MGE thought to produce CRABP1+ striatal interneurons, indicating partitioning of *Etv1* and *Lhx8* domains in mouse MGE. **c**. SCENIC module scores (Y-axis) vs log2 fold change of hub transcription factor predicted to regulate the module (X-axis). Significance represents multiple testing corrected q-value < 0.1 for both diffxpy differential expression in macaque and also q-value < 0.1 SCENIC logistic regression coefficient q-value calculated by shuffling class labels. Size represents the proportion of all CRABP1+ cells which also express the gene. **d**. UMAP projection showing the region from which macaque cells are derived. **e**. UMAP projection showing classes in cells expressing 2 or more of (CRABP1, ETV1, ANGPT2). **f**. Subclustering of rare NKX2-1+ cells from organoid dataset[58], labeled by Leiden subclusters. **g**. NKX2-1+ cells from organoid dataset[58], labeled by the experimental conditions of the differentiation. **h**. Dotplot of expression of MGE_CRABP1 related markers in Leiden subclusters showing cluster 6 likely contains MGE_CRABP1/MAF and MGE_CRABP1/TAC3 cells.

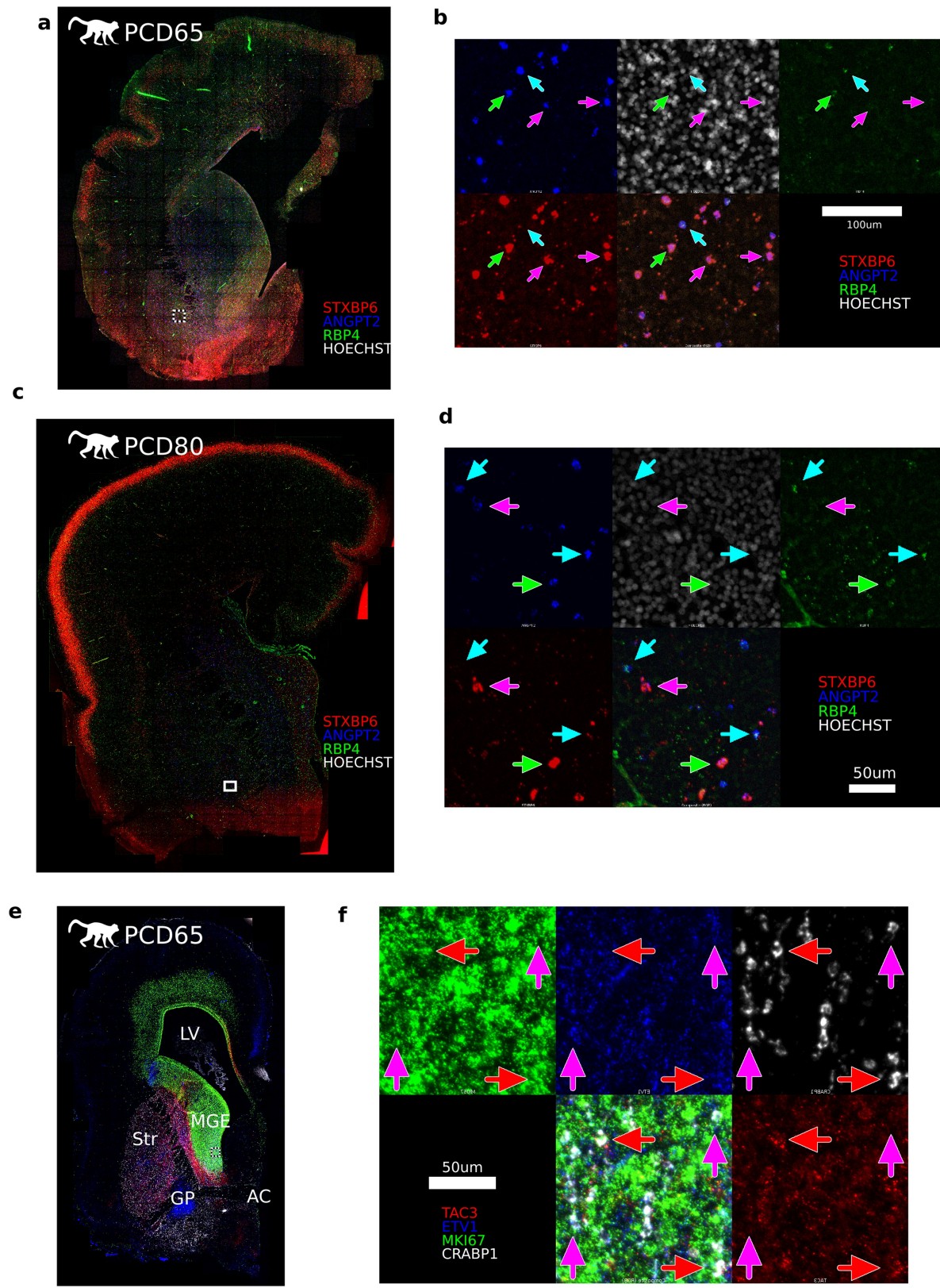

**Extended Data Fig. 7** | See next page for caption.

**Extended Data Fig. 7 | Spatial distribution of CRABP1+/TAC3+ and MAF+ striatal interneuron marker expression. a**. PCD65 macaque brain section, from different individual than main text quantifications showing RNA expression of alternative markers of both CRABP1 classes (*ANGPT2*), the MGE_CRABP1/MAF class (*RBP4*) and the MGE_CRABP1/TAC3 class (*STXBP6*). Note that *RBP4* expression is rare at PCD65, but much more common at PCD80 (see **c, d**). **b**. Montage from **a** showing MGE_CRABP1/MAF (cyan arrows), MGE_CRABP1/TAC3 (magenta arrows) and STXBP6+/RBP4+ cells (green arrows). **c**. PCD80 macaque brain section showing RNA expression of alternative markers of both CRABP1 classes (*ANGPT2*), the MGE_CRABP1/MAF class (*RBP4*) and MGE_CRABP1/TAC3 (*STXBP6*). **d**. Montage from **c** showing MGE_CRABP1/MAF (cyan arrows), MGE_CRABP1/TAC3 (magenta arrows) and STXBP6+/RBP4+ cells (green arrows). **e**. PCD65 macaque brain section showing RNA expression of *CRABP1*, marking both CRABP1 classes, *ETV1*, marking both CRABP1 classes, the dLGE, and the GP, *TAC3* marking the MGE_CRABP1/TAC3 class and MKI67 marking dividing cells. Labeled regions are abbreviated LV: Lateral Ventricle, MGE: Medial Ganglionic Eminence, Str: Striatum, GP: Globus Pallidus, AC: Anterior Commissure **f**. Montage from **e** showing MGE_CRABP1/TAC3 (magenta arrows) and TAC3+/MKI67+ cells (red arrows).

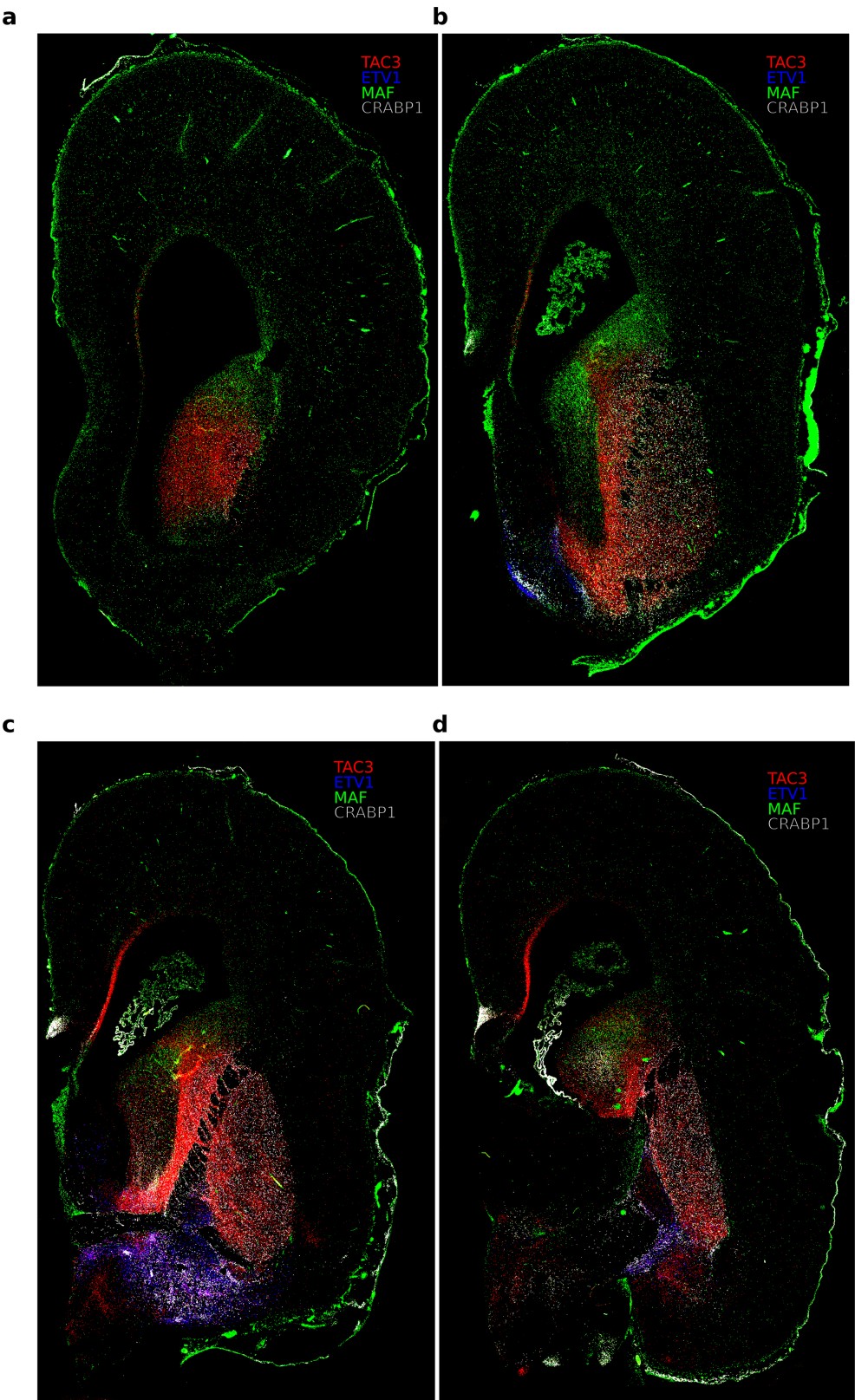

**Extended Data Fig. 8 | CRABP1+/TAC3+ and MAF+ striatal interneuron marker expression.** Full size tile scanned representative images from Fig. 3c. **a**–**d**. Sections 0, 37, 66, and 101 four color in situ hybridization for CRABP1, LHX8, MAF, TAC3.

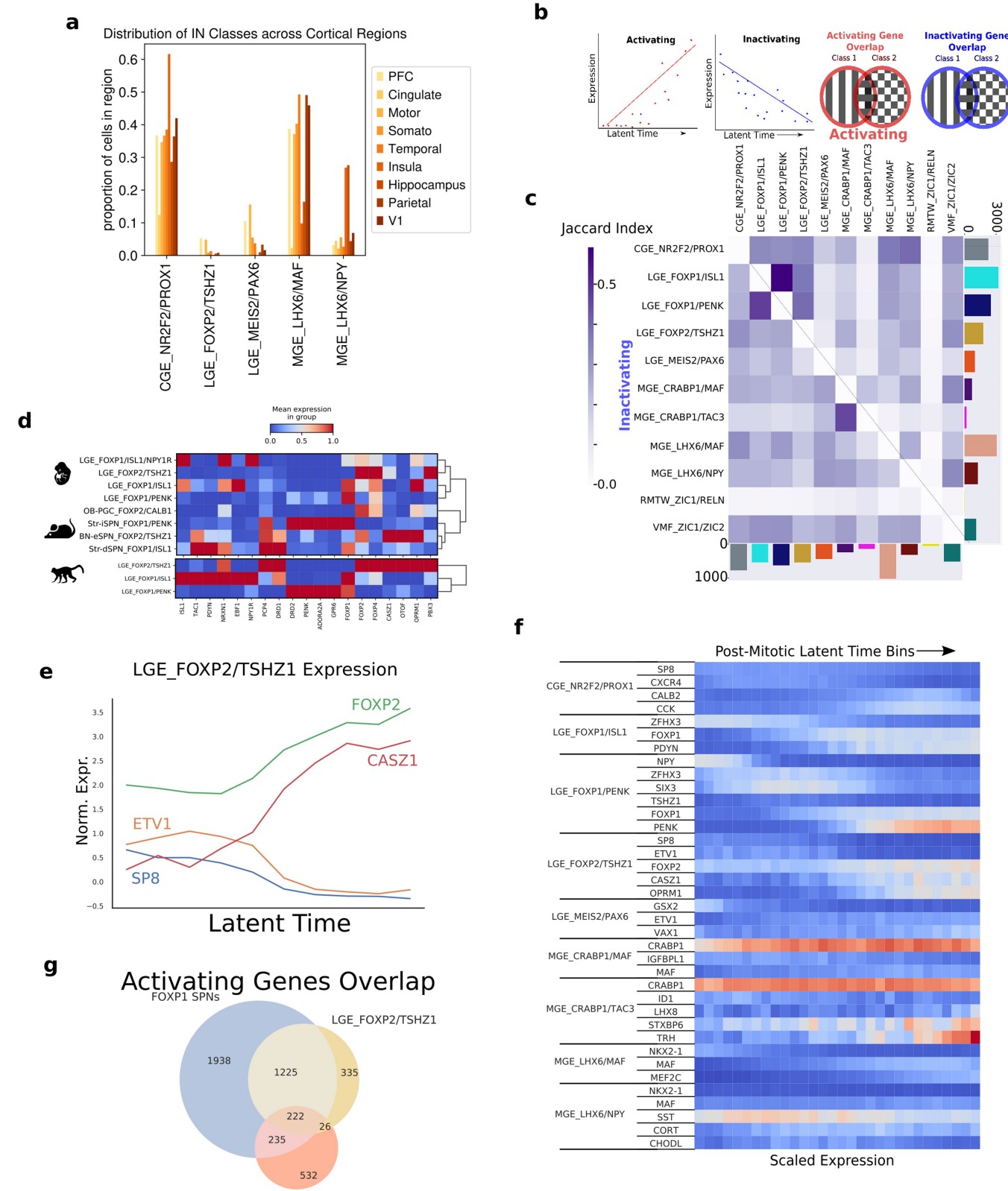

**Extended Data Fig. 9 |** See next page for caption.

**Extended Data Fig. 9 | Spatial, temporal, and molecular distinctions among initial LGE-derived neurons. a**. Bar plot of the proportion of cells from each initial class across cortical regions for CGE-, LGE- and MGE-derived classes highlights frontal lobe enrichment of LGE classes during sampling. **b**. Schematic for **c**, showing cartoon scatter plots for the expression of two genes in individual cells regressed against latent time (left) to describe genes that activate or inactivate during neuronal differentiation, followed by Venn diagrams showing the overlap of dynamically regulated genes between pairs of initial classes used to calculate Jaccard indices. **c**. Heatmap of Jaccard indices of significantly overlapping lists of dynamic genes between cell classes of Holm-Šídák corrected linear regression q value < 0.05 on scaled and normalized gene expression to address the extent of shared versus cell type-specific trajectories during post-mitotic differentiation of initial classes. Bar graphs on axes represent the total number of significant genes activating/inactivating. Direct and indirect medium spiny neurons (LGE_FOXP1/ISL1 and LGE_FOXP1/PENK) show strong overlap of both activating and inactivating genes, despite early partitioning as distinct initial classes, while presumed eccentric spiny neurons from LGE_FOXP2/TSHZ1 show strong overlap of activating genes (see also **g**) but not inactivating genes. Conversely, MGE_CRABP1/MAF and MGE_CRABP1/TAC3 classes show strong overlap of inactivating genes among a smaller overall set of dynamically regulated genes, but little overlap in activating genes. **d**. Heatmap of selected marker expression of LGE-derived striatal initial and terminal classes, scaled by gene. **e**. Gene expression of dynamic dLGE marker genes in macaque LGE_FOXP2/TSHZ1 cells across shared latent time, ordered by latent time value and divided into 10 equally sized bins to provide stable mean expression values. **f**. Gene expression of dynamic marker genes in macaque cells across shared latent time, grouped by initial class, ordered by latent time value and divided into 30 equally sized bins to provide stable mean expression values. **g**. Venn diagram of intersections of significantly activating gene sets in LGE initial classes. In contrast to LGE_MEIS2/PAX6, the LGE_FOXP2/TSHZ1 class activates a large set of shared SPN genes during neuronal differentiation as inferred by latent time trajectories.

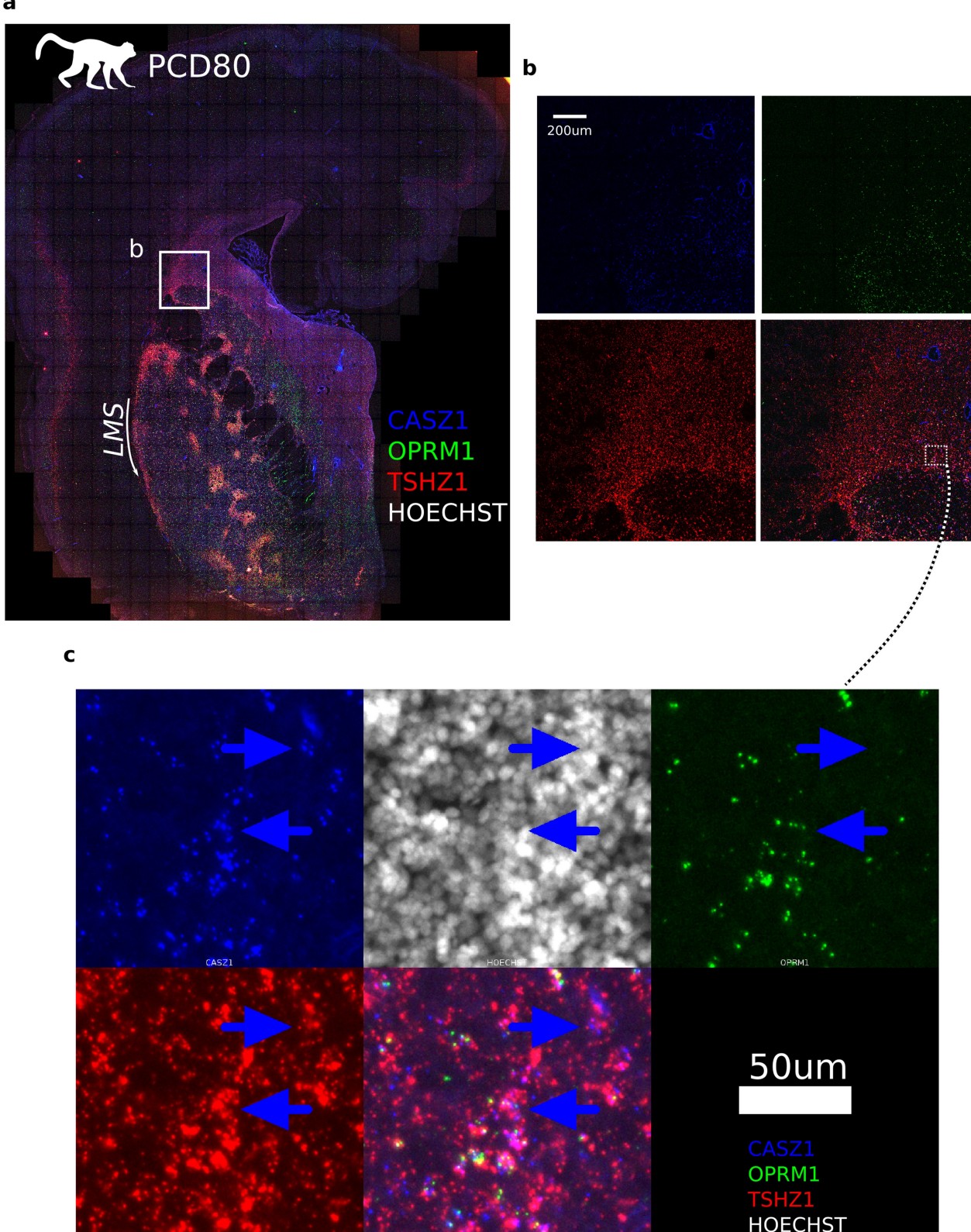

**Extended Data Fig. 10 | Emergence of Str-eSPN_FOXP2/TSHZ1 from LGE_
FOXP2/TSHZ1 in the dLGE. a**. PCD80 macaque coronal section showing RNA
expression of eSPN markers. The Lateral Migratory Stream (LMS) is noted as is
shown in Kuerbitz et al[19]. **b**. Montage of magnified dLGE at the striatum-GE
boundary. **c**. Montage from the box in **b** showing TSHZ1/CASZ1/OPRM1- (top
arrow) and TSHZ1/CASZ1/OPRM1+ (bottom arrow) Str-eSPN_FOXP2/TSHZ1
cells (blue arrows) within the GE.

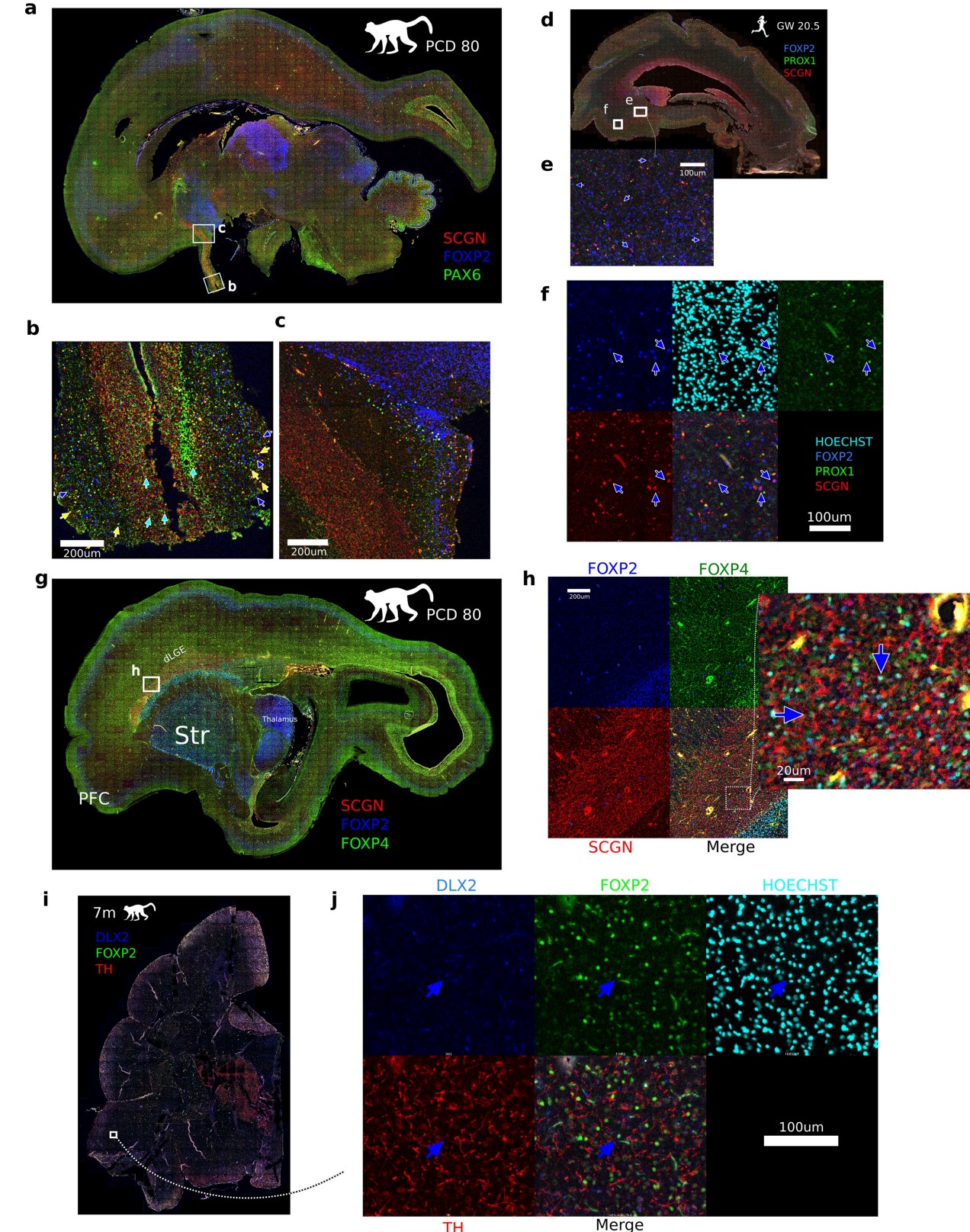

**Extended Data Fig. 11** | See next page for caption.

**Extended Data Fig. 11 | Distribution of dLGE-derived LGE_FOXP2/TSHZ1 precursors in the superficial white matter. a**. Medial sagittal section of PCD80 macaque brain. The SCGN+ RMS originating at the anterior pole of the dLGE is seen extending from the olfactory ventricle to the OB. **b**. FOXP2+/PAX6+ cells from lateral migratory streams converge with RMS and enter periglomerular layers of OB (see also Extended Data Fig. 1). Note that FOXP2+ OB-PGC_FOXP2/ CALB1 cells are largely absent from the RMS but are found ventral of the nucleus accumbens (NAc), anterior olfactory nucleus (AON) and in outer olfactory tract sheath. **c**. SCGN+/PAX6+ granule cells (OB-GC_MEIS2/PAX6) (cyan arrows), TH+ PGCs (OB-PGC_TH/SCGN) (yellow arrows) and FOXP2+ PGCs (OB-PGC_FOXP2/CALB1) (blue arrows) in OB. **d–f**. Human gestation week 20.5 sagittal cortex section shows new-born FOXP2+/SCGN+/PROX1- neurons (blue arrows) migrating into the ventral cortex superficial white matter. SCGN+ expression decreases as cells mature. **g**. Lateral sagittal section of PCD80 macaque brain. **h**. Immunofluorescence FOXP2+/FOXP4+/SCGN+ dLGE-derived projection class neurons (blue arrows) are seen in large numbers in the dLGE portion dorsal of the caudate, and in adjacent cortical white matter and striatum. **i**. 7 month old macaque coronal section. **j**. Montage from box in **i** with rare DLX2/FOXP2+ superficial white matter IN (SWMIN) marked with a blue arrow.

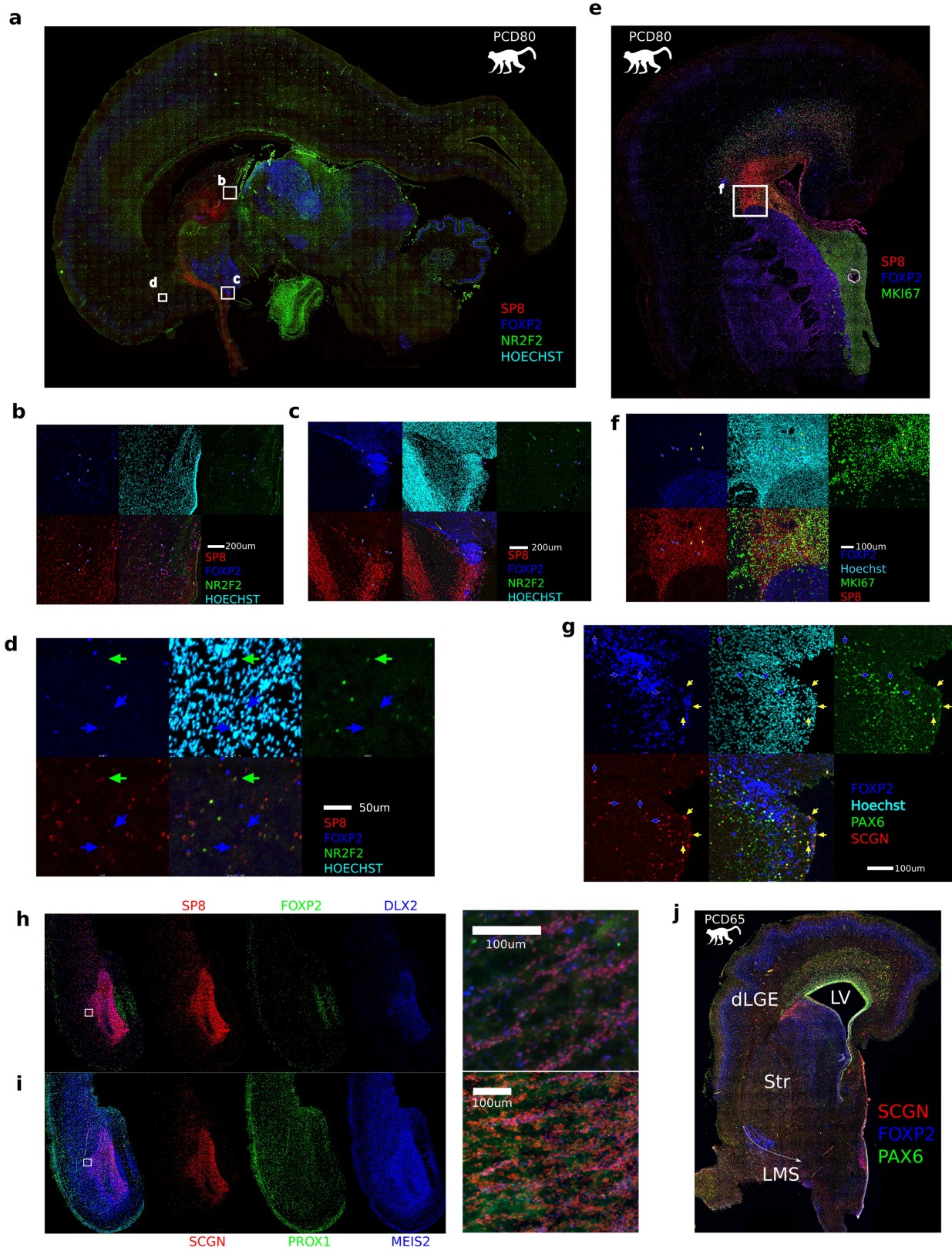

**Extended Data Fig. 12 | dLGE migration streams. a–d**. PCD 80 macaque brain sagittal section. Blue arrows represent SP8+/FOXP2+ cells in (**b**) septum and fornix (it is unclear whether these cells are born here, or arrive via an RMS dorsal extension or via a cortex-indusium griseum-fornix route, possibly seen in[59]) (**c**) anterior olfactory nucleus (these cells appear to be near the point where the LMS is converging with the RMS, suggesting a lateral source of FOXP2+ PGCs) and (**d**) vmPFC. **e**, **f**. Coronal section of macaque PCD80 brain shows large numbers of SP8+/FOXP2+ cells and SP8+/MKI67+ cells in DL-dLGE.

**g**. Sagittal section of anterior olfactory nucleus. New-born dLGE-derived PAX6+/SCGN+/FOXP2- labeled with yellow arrows. **h**. Oblique coronal-axial section of dLGE and PFC with insets highlighting DLX2+/SP8+/FOXP2- parenchymal chains. **i**. Oblique coronal-horizontal section of dLGE and PFC with insets highlighting MEIS2+/SCGN+/PROX1- parenchymal chains. **j**. Coronal section of PCD65 macaque brain with lateral ventricle (LV), striatum (Str), dLGE and approximate lateral migratory stream (LMS) labeled.

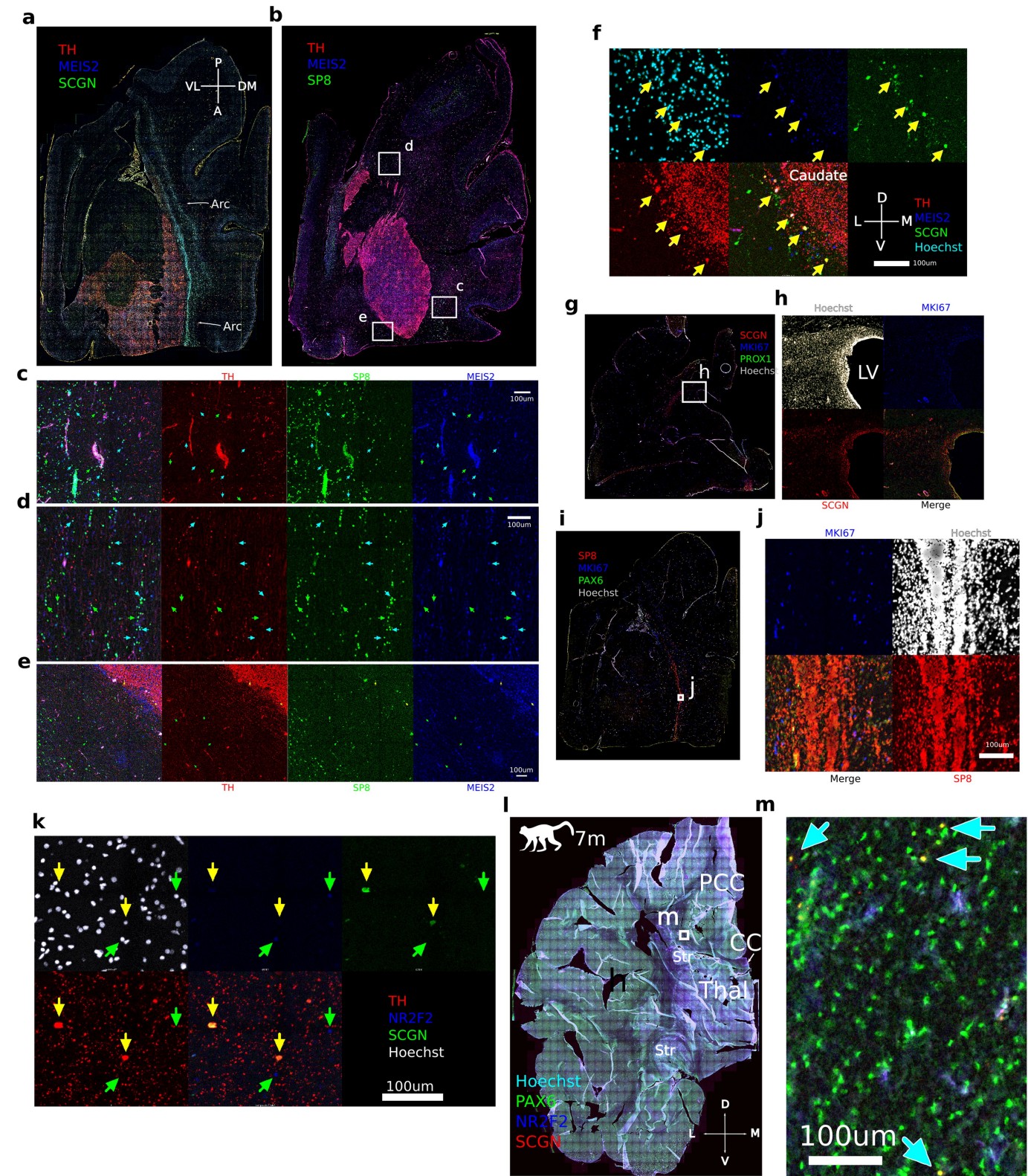

**Extended Data Fig. 13** | See next page for caption.

**Extended Data Fig. 13 | A-dLGE cells in the Arc. a**. Low magnification stitching of oblique horizontal section showing large stream of MEIS2+/SCGN+ chains in the Arc along the dorsomedial edge of the TH+ striatum. **b**. Further dorsal oblique horizontal section showing dorsal and lateral migratory streams. **c**. Enlargement of Arc-ACC. Cyan arrows denote MEIS2+/SCGN+ dLGE cells. Note that the stream is continually bounded by the increased density of TH+ fibers. **d**. Chains caudal of the striatum are of mixed classes. **e**. White matter neurons lateral of the striatum are nearly all MEIS2-negative (green arrows). Also note peristriatal SP8+/MEIS2+/TH+ *striatum laureatum* neurons (SLNs) at lateral border of striatum (yellow arrows). **f**. Coronal section of PCD120 macaque striatum showing an array of TH+ peristriatal SLNs(yellow arrows) at the edge of the caudate nucleus, from Fig. 3b, at the same location as 7 month postnatal, but not yet having developed SCGN+/TH+ processes tangential to the external capsule. **g**, **h**. Coronal section of PCD120. Sparse MKI67+ cells at ventricle, with SCGN+ cells away from the ventricle being MKI67-. **i**, **j**. Chains in Arc do not appear to be MKI67+. **k**. PCD120 SCGN/TH+ peristriatal SLNs (yellow arrows) are NR2F2 negative (NR2F2+ DWMINs labeled with green arrows). Note that the LGE_MEIS2/PAX6 and LGE_FOXP2/TSHZ1 classes very sparsely expressed the transcript encoding tyrosine hydroxylase (TH), a rate limiting enzyme in dopamine production, and were the only cortical IN classes to do so at developmental stages (see Fig. 4b). **l**. Coronal section of macaque cortex at 7 months. **m**. PAX6+/SCGN+/NR2F2- deep white matter neurons in postnatal macaque cingulate cortex (cyan arrows). These DWMINs were found in the cingulate white matter and corona radiata, though not in the corpus callosum itself and rarely near the deep layers of the cortical plate and external capsule white matter.

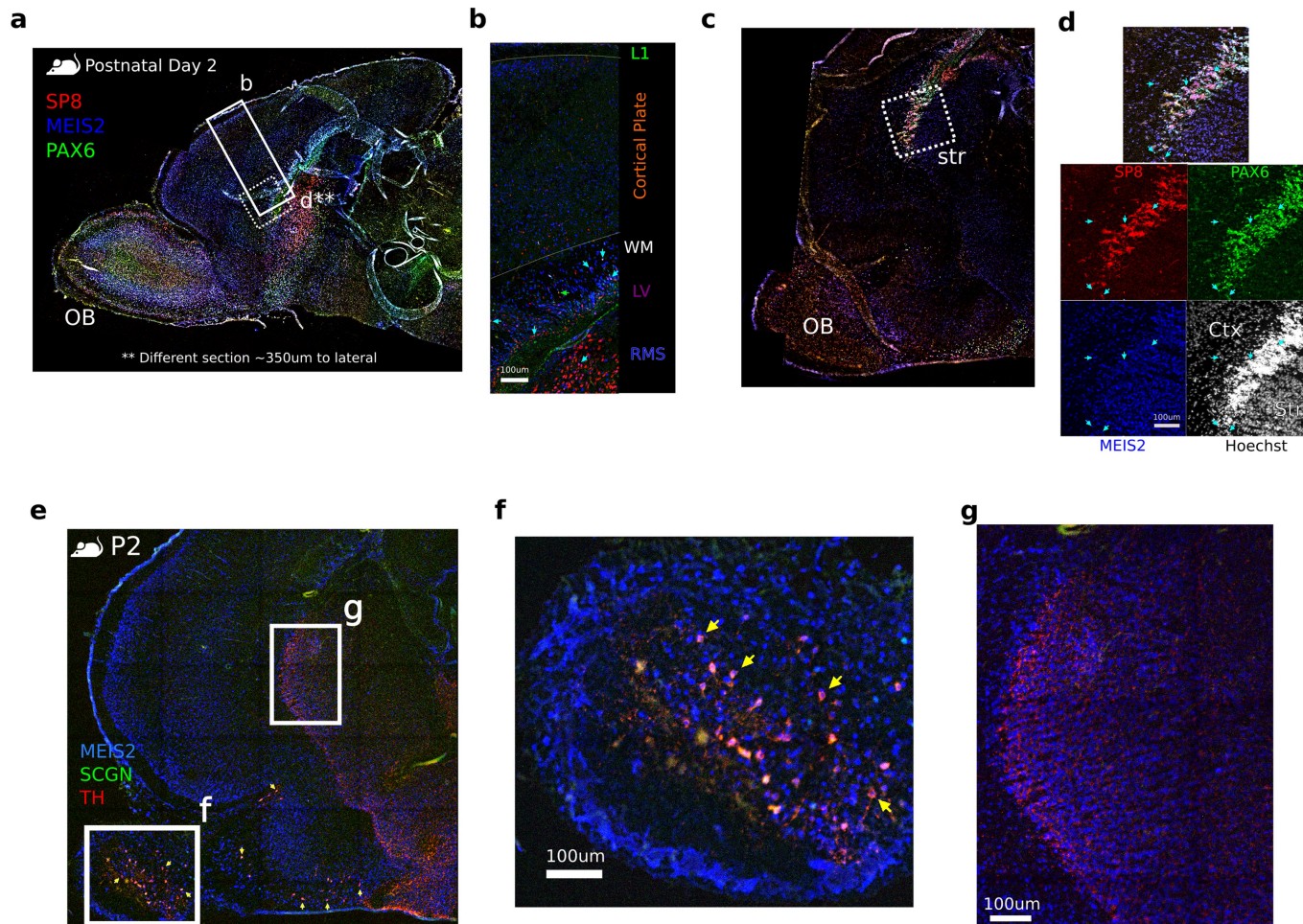

**Extended Data Fig. 14 | Distribution of LGE_MEIS2/PAX6-derived cells in postnatal mouse. a.** Mouse sagittal section showing **b** and approximate **c** magnification locations. **b.** LGE_MEIS2/PAX6 cells in deep white matter (cyan arrows) **c.** Lateral sagittal section **d.** Panel showing SP8+/MEIS2+/PAX6+ cells in remainder of dLGE, likely homologous to dLGE chains in Arc. **e.** Lateral sagittal section of mouse postnatal day 2. **f.** MEIS2+/SCGN+/TH+ periglomerular cells in lateral OB. **g.** Striatum shown with dense TH+ projection fibers and synapses, but no MEIS2+/SCGN+/TH+ cell bodies.

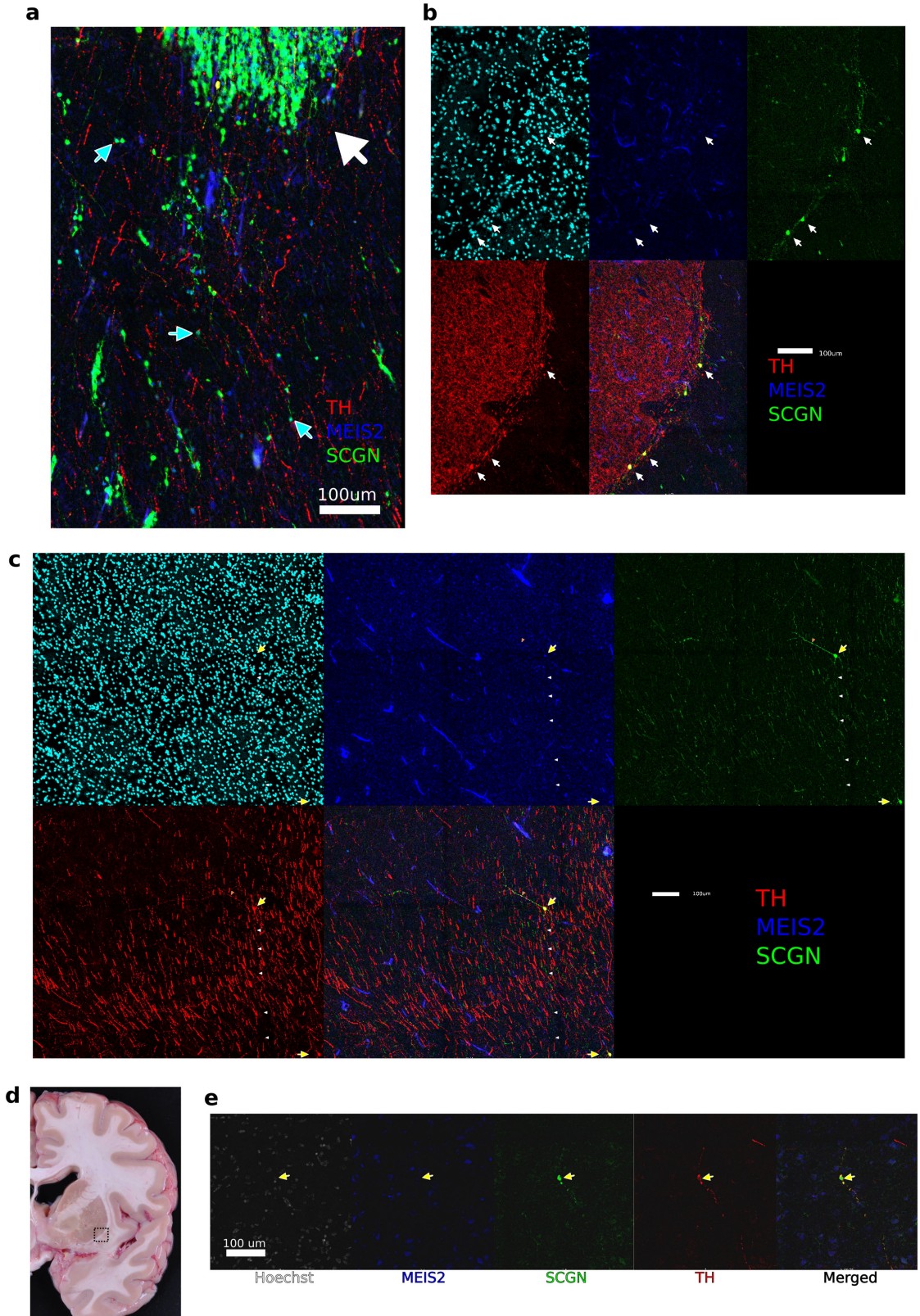

**Extended Data Fig. 15 | Full montages of Fig. 5 peristriatal *striatum laureatum* neurons. a**. Remnant immature neurons of the RMS from 7m macaque (large white arrow) with migrating morphology MEIS2+/SCGN+ cells (teal arrows). **b**. Montage of edge of striatum with arrows pointing to MEIS2+/SCGN+/TH+ SLNs. **c**. Montage of claustrum with yellow arrows pointing to MEIS2+/SCGN+/TH+ peristriatal SLNs. The TH+/SCGN+ process is labeled with orange arrowheads while TH-/SCGN+ process is labeled with white arrowheads. **d**. Photograph of 88 year old brain with approximate region of **e** in box. **e**. Montage of separate channels of the SLN shown in Fig. 5h.

# Reporting Summary

Nature Research wishes to improve the reproducibility of the work that we publish. This form provides structure for consistency and transparency in reporting. For further information on Nature Research policies, see our Editorial Policies and the Editorial Policy Checklist.

## Statistics

For all statistical analyses, confirm that the following items are present in the figure legend, table legend, main text, or Methods section.

| n/a | Confirmed | |
|---|---|---|
| ☐ | ☒ | The exact sample size ($n$) for each experimental group/condition, given as a discrete number and unit of measurement |
| ☐ | ☒ | A statement on whether measurements were taken from distinct samples or whether the same sample was measured repeatedly |
| ☐ | ☒ | The statistical test(s) used AND whether they are one- or two-sided<br>*Only common tests should be described solely by name; describe more complex techniques in the Methods section.* |
| ☐ | ☒ | A description of all covariates tested |
| ☐ | ☒ | A description of any assumptions or corrections, such as tests of normality and adjustment for multiple comparisons |
| ☐ | ☒ | A full description of the statistical parameters including central tendency (e.g. means) or other basic estimates (e.g. regression coefficient) AND variation (e.g. standard deviation) or associated estimates of uncertainty (e.g. confidence intervals) |
| ☐ | ☒ | For null hypothesis testing, the test statistic (e.g. $F$, $t$, $r$) with confidence intervals, effect sizes, degrees of freedom and $P$ value noted<br>*Give P values as exact values whenever suitable.* |
| ☒ | ☐ | For Bayesian analysis, information on the choice of priors and Markov chain Monte Carlo settings |
| ☒ | ☐ | For hierarchical and complex designs, identification of the appropriate level for tests and full reporting of outcomes |
| ☐ | ☒ | Estimates of effect sizes (e.g. Cohen's $d$, Pearson's $r$), indicating how they were calculated |

*Our web collection on statistics for biologists contains articles on many of the points above.*

## Software and code

Policy information about availability of computer code

| | |
|---|---|
| Data collection | In generating data, software used included Thermo Fisher Evos M7000 microscope control software, Illumina sequencer control software and bcl2fastq (bcl2fastq2) software. |
| Data analysis | Single cell RNA-seq analysis was carried out in custom python and bash scripts, implementing published algorithms as described and cited in methods using published algorithms.<br><br>Quality control: Kallisto-Bus output matrix files (including both introns and exons together) were input to Cellbender (release 0.2.0, https://github.com/broadinstitute/CellBender), which was used to remove likely ambient RNA only. Only droplets with greater than 0.99 probability of being cells (not ambient RNA), calculated by the Cellbender model, were included in further analysis. Droplets with fewer than 800 genes detected, or greater than 40% ribosomal or 15% mitochondrial reads were filtered from the dataset. Doublets were then detected and removed from the dataset using scrublet (release 0.2.2, using threshold parameter 0.5).<br><br>Clustering and determining homologous cell types: Much of the analysis pipeline is based on scanpy infrastructure and anndata data structures48. Counts in cells were normalized by read depth, log transformed and then scaled for each gene across all cells. Principal component analysis was then performed using the top 12,000 most variable genes (using the original Seurat variable genes selection method, implemented in the scanpy package), with the 100 most variance-encompassing principal components being used for the following steps. Batch correction was limited to the requirement that highly variable genes be variable in more than one sequencing sample and by application of batch-balanced k-nearest neighbors (BBKNN)49, using Euclidean distance of principal components to find 3 neighbors per batch in the developing data, and 12 neighbors per dataset in the developing and adult merged mouse data. Using BBKNN-derived k-nearest neighbors graphs, Leiden clustering was then applied to cluster based upon the KNN graph with scanpy's resolution parameter set to 10 (or 7 in the developing mouse dataset). Glia, along with excitatory progenitor and neuron clusters were removed from the dataset in non-ganglionic eminence batches if they had below mean expression value for two or more GAD1/2 and DLX1/2/5/6 genes, with Cajal-Retzius cells (RMTW_ZIC1/RELN) meeting this threshold and serving as a useful out-group (these cells were called RMTW-derived based on the ZIC1/2 and RELN expression, though they are known to have multiple origins50). Following removal of non-INs, scaling, PCA and the following steps were |

repeated with this final IN dataset.

High-resolution Leiden clusters partitioned continuous differentiation trajectories of post-mitotic initial classes into subclusters based on maturation stage. These high resolution clusters were then merged to initial classes manually, using hierarchical clustering of cluster gene expression averages and distinctness of individual Leiden cluster markers as a guide. The nomenclature for merged clusters incorporates the presumptive spatial origin of initial classes and specific marker genes. Spatial origin for each class was inferred based on the expression of canonical marker genes for RMTW, MGE, LGE, CGE, and VMF (e.g., LHX5, NKX2.1, MEIS2, NR2F2, ZIC1) and supported by immunostaining and by the enrichment of these genes in cells from region-specific dissections. For merged species analysis, genes were normalized and scaled within species, then merged for downstream analysis using BBKNN (with 25 neighbors across and within species, with the mutual nearest neighbors used for the Sankey plot comparison of developing macaque and developing mouse). Following clustering, mean expression in each class was calculated for each gene which was among the original 12,000 most variable 1-to-1 orthologs from each dataset that were variable in both species (6,227 genes). These classes were then compared across species by Pearson correlation of their gene expression vectors.

Trajectory analysis of activating and inactivating macaque genes: We applied scVelo's dynamical model (release 0.2.3) 51 to derive a shared latent time based on RNA velocity using spliced and unspliced counts from kallisto. Next, we used the related cellrank (release 1.3.1) package15 to derive absorption probabilities of immature cells in the "Transition" cluster to likely initial classes.

Image processing: Stitching was carried out using python scripts implementing ImageJ's max correlation Grid/Collection stitching (V1.2). Downstream image background subtraction, brightness/contrast adjustment, merging and montaging was also performed using ImageJ (1.53c).

For manuscripts utilizing custom algorithms or software that are central to the research but not yet described in published literature, software must be made available to editors and reviewers. We strongly encourage code deposition in a community repository (e.g. GitHub). See the Nature Research guidelines for submitting code & software for further information.

## Data

Policy information about availability of data

All manuscripts must include a data availability statement. This statement should provide the following information, where applicable:

- Accession codes, unique identifiers, or web links for publicly available datasets
- A list of figures that have associated raw data
- A description of any restrictions on data availability

The sequencing data have been deposited in GEO under the accession GSE169122, and the data is browsable at https://cells-test.gi.ucsc.edu/?ds=macaque-dev-inhibitory-neurons. Python scripts to analyze data and generate figures are saved on github and are available upon request. Raw imaging data are available on request, though we are working to make all files available. The images are currently too large, numerous and complex

# Field-specific reporting

Please select the one below that is the best fit for your research. If you are not sure, read the appropriate sections before making your selection.

☒ Life sciences ☐ Behavioural & social sciences ☐ Ecological, evolutionary & environmental sciences

For a reference copy of the document with all sections, see nature.com/documents/nr-reporting-summary-flat.pdf

# Life sciences study design

All studies must disclose on these points even when the disclosure is negative.

| | |
|---|---|
| Sample size | Data was collected from as many individuals as were available, with 9 embryos being collected and an additional 2 from public data being utilized. No sample size calculation was performed, as we do not attempt to test individual-level covariates (individual, sex, timepoint). |
| Data exclusions | Excitatory neurons and their progenitors, as well as glia were filtered from the dataset bioinformatically. This has been done in previous Nature studies, for example Krienen et al., 2020. |
| Replication | In order to improve replicability, samples were collected across 3 years from 9 different individuals. Findings were further verified against data from public datasets, as well as in another species (mouse). Furthermore, immunofluorescence microscopy was used to corroborate findings in at least one and often multiple different brains. This is above the standard of similar studies of this type published in major journals at this time. |
| Randomization | Randomization was not relevant to this study. |
| Blinding | Blinding was not relevant to this study. |

# Reporting for specific materials, systems and methods

We require information from authors about some types of materials, experimental systems and methods used in many studies. Here, indicate whether each material, system or method listed is relevant to your study. If you are not sure if a list item applies to your research, read the appropriate section before selecting a response.

## Materials & experimental systems

| n/a | Involved in the study |
|-----|----------------------|
| ☐ | ☒ Antibodies |
| ☒ | ☐ Eukaryotic cell lines |
| ☒ | ☐ Palaeontology and archaeology |
| ☐ | ☒ Animals and other organisms |
| ☐ | ☒ Human research participants |
| ☒ | ☐ Clinical data |
| ☒ | ☐ Dual use research of concern |

## Methods

| n/a | Involved in the study |
|-----|----------------------|
| ☒ | ☐ ChIP-seq |
| ☒ | ☐ Flow cytometry |
| ☒ | ☐ MRI-based neuroimaging |

## Antibodies

Antibodies used

PAX6
Rabbit
BioLegend
901301
1:750
lot: B277104

PAX6
Sheep
R&D
AF8150
1:200
lot:CDJL0420021

NKX2-1
Rabbit
Millipore
MILL-07-601
1:750
lot: I3448599

FOXP2
Mouse
Millipore
MABE415
1:500
lot: 3278590
FOXP2
Sheep
R&D
AF5647
1:2000

FOXP4
Rabbit
Millipore
ABE74
1:500
lot: 3050680
NR2F2
Mouse
Perseus Proteomics
PP-H7147-00
1:750
lot: A-2

SCGN
Goat
Fisher
AF4878SP
1:500
Lot CASX0119111

MEIS2
Mouse
Sigma
WH0004212M1

1:500
lot: J1071-1H4

SP8
Rabbit
Millipore
HPA054006
1:500
lot: 000003237

TH
Sheep
Millipore
AB1542
1:1000
lot: 3403416

TH
Rabbit
Millipore
AB152
1:1000
lot: 3574360

MKI67
Mouse
Dako
MIB-1
1:250
20057589

Validation | Antibodies are all common and commercially available and have been widely used. Antibodies were used on large tissues, and for each one, the entire section. Each antibody is thus internally controlled by expected anatomical distribution and subcellular localization.

# Animals and other organisms

Policy information about studies involving animals; ARRIVE guidelines recommended for reporting animal research

Laboratory animals | Postmortem macaque embryonic brain tissue (between embryonic day 40 to 100) of unknown sex was obtained from the California National Primate Research Center at UC-Davis.

Wild animals | *Provide details on animals observed in or captured in the field; report species, sex and age where possible. Describe how animals were caught and transported and what happened to captive animals after the study (if killed, explain why and describe method; if released, say where and when) OR state that the study did not involve wild animals.*

Field-collected samples | *For laboratory work with field-collected samples, describe all relevant parameters such as housing, maintenance, temperature, photoperiod and end-of-experiment protocol OR state that the study did not involve samples collected from the field.*

Ethics oversight | Macaque tissue was generously provided by the UC Davis Primate Center. All animal procedures conformed to the requirements of the Animal Welfare Act and protocols were approved prior to implementation by the Institutional Animal Care and Use Committee (IACUC) at the University of California, Davis. De-identified tissue samples were collected with previous patient consent in strict observance of the legal and institutional ethical regulations. Protocols and samples were approved by UCSF GESCR (Gamete, Embryo, and Stem Cell Research) Committee.

Note that full information on the approval of the study protocol must also be provided in the manuscript.

# Human research participants

Policy information about studies involving human research participants

Population characteristics | N/A

Recruitment | N/A

Ethics oversight | Protocols were approved by the Human Gamete, Embryo and Stem Cell Research Committee (institutional review board) at the University of California, San Francisco

Note that full information on the approval of the study protocol must also be provided in the manuscript.

