## [Peer Review File · Nature]

Manuscript Title: Atlas of Initial Inhibitory Neuron Classes During Primate Cerebral Development

Reviewer Comments & Author Rebuttals

Reviewer Reports on the Initial Version:

Referee #1 (Remarks to the Author):

The paper by Schmitz et al. examines the origins of inhibitory neurons in primates compared to that seen in mice and provides a developmental extension of recent work examining the adult fates of these populations in primate by the McCarroll laboratory. The study confirms recent work in mice suggesting that cell type signatures in mitotic cells are weak. Nonetheless, based on transcriptional signatures resembling those seen postmitotically, they argue that there is some indication of differentiated character in inhibitory projection progenitors, but less so in the similarly derived interneuron populations. Even so the identities of these provisionally committed progenitors reflect weak TF signatures defined by broadly expressed factors such as NR2F1/F2, Mies2, Ebf1 and Isl1. Comparison to that seen in mouse showed strong similarities and interestingly better gene conservation in progenitors than seen in more mature populations, suggesting that developmental programs are more conserved than terminal fates. The study then focuses on the TAC3 striatal population that was recently identified as giving rise to a novel striatal interneuron population in primates and tentatively identifies the progenitor zone origins as a CRABP1+ MGE population, which they speculate diverges from the related CRABP1-MAF class. While the speculation here is intriguing, the over-reliance on maintained gene expression to infer relatedness places their conclusions on a weak footing. The narrative of the manuscript is also somewhat out of synch with the actual figures. For instance, Figure 2 extensively explores genes that are extinguished versus initiated across development as latent time extends, which falls short of making a compelling argument for the purported gene trajectories of specific subtypes. Figure 2d and e while making an argument for the change in the developmental expression of class1 and 2 genes using regression across time and Jaccard indices present a confusing picture of dynamic genes in cell types whose meaning is poorly explained. In the end these trajectories while arguing for the trajectory of genes within subtypes does not convincingly validate their predictions. In Figure 3 the study focuses on the emergence for the TAC3 population and show good evidence for a conserved expression profile (Nkx2.1, Nxp1, Etv1) from macaque specific CRABP1 populations that are either TAC3 or MAF positive, with the former being novel and the latter being conserved. All of these findings while interesting are neither convincing enough in themselves based on the gene expression in populations or across time without further characterization or validation. to extend our understanding of them sufficiently to warrant publication in Nature.

An argument could also be made for combining the two papers into a single concise manuscript as they make extensive use of the same data and as is each paper contains undescribed details relevant to the other paper (e.g. Figure 1a in the OB paper uses the same UMAP as in the initial classes paper, but the gray cells relevant to the classes paper are not described, and some of the same cell populations are included in analysis of both texts).

Comments

- This manuscript includes an extensive dataset from multiple ages and multiple structures, yet the analysis is not vigorous enough to justify the amount of data included. For example, the regions of the cells were only shown in Figure without much discussion about the interpretation and meaning. Also, putting the progenitor region CGE, MGE, LGE label together with other structures such as PFC, Striatum, etc.. in figure 1e is confusing if one wants to compare the relative contribution of each cell type to different regions.
- Regarding the homology between mouse and macaque: 1. what ages are homologous across

species and is it differ by cell type? 2. are there major species differences in relative order or timing of major inhibitory classes by brain region? (line 94 suggests not but no citations or evidence from mouse are given). 3. The datasets may be imperfectly matched across species. The paper includes data from multiple structures in macaque. It isn't clear if there are comparable mouse datasets. Information regarding mouse dataset were only briefly mention in Method, it will be clearer if timepoints and regions for mouse are also included Supplementary Table 1). 4. Is it possible to align mouse and macaque dataset in one umap space to provide a more direct illustration for the conclusion that the authors are trying to make, which is the conserved progenitor population that becomes more distinct between the two species later. I imagine mouse E13.5, E14.5 dataset would be the progenitor timepoint, and 10X mouse E18 cortex data could serve as a more diversified time point.

- Should make sure to clarify throughout the text when a population is given a spatial designation because it was dissected there or inferred to be of that location for other reasons. An example is RMTW_ZIC1/RELN (line 50). Overall, I find the naming conventions a bit confusing. E.g. the mouse Crabp1/Maf class does not seem to express much Crabp1 (Fig 3f)?

- The model for the origin of the TAC3 striatal type is interesting but it is unclear how the correspondence to the adult type is confirmed since macaque adult samples were not included in the analysis. Similarly, how is it determined that the TH and PVALB types emerge from the CRABP1/MAF class, and when do they become distinct? While the authors conclude that the TAC3 initial class does not exist at any point in mouse development, Fig 3a-f suggests that the mouse CRABP1/MAF type maps to both the macaque CRABP1/MAF and the CRABP1/TAC3 initial classes rather than selectively to the CRABP1/MAF class. That seems different than preservation of an ancestral class (CRABP1/MAF) + derivation of a new one.

- I found ED5b difficult to understand because so many structures are plotted in different colors. E.g. the CRABP1/MAF cells in xLGE – are they presumably migrating?

Figure 1: UMAPS should be horizontally inverted, and plane of coronal section should be shown in sagittal views.

Figure 2: macaque Model in b is intriguing. Do they have putative labels for cells at each of these stages for different cell types? I assume the class 1 and class2 is something akin to Jessell's Shh response genes but in time rather than space, but this is neither clear nor well spelled out in the text. There is a good idea here but it is lost on me.

Figure 3: Not sure what I am supposed to get from A: B says it is comparison of 1000 genes but each of these are mixed so more like paralog cells comparisons than genes. This graph is a real mix of Mge, Striatal and

ED2a – As presented, it is difficult to tell when a population is not present because of the developmental stage or because the structure was not sampled in that timepoint.

Referee #2 (Remarks to the Author):

In the article "Atlas of Initial Inhibitory Neuron Classes During Primate Cerebral Development", Schmitz et al. describe the molecular logic of inhibitory neuron specification during macaque early-fetal development. Using their newly generated single-cell RNA-seq dataset and integrating it with available datasets from mouse and more mature macaque samples, they show that almost all the initial classes of early born inhibitory neurons are conserved between macaque and mouse. This indicates that the previously reported increased diversity of inhibitory neuron populations in primates is likely due to extrinsic factors that occur later, from midfetal to early postnatal periods. After presenting their resource, they then analyze whether the recently reported primate-specific population of striatal TAC3 interneurons was transiently present or completely absent during

mouse development. They report that this population cannot be detected throughout development in mouse and that it's actually specified early in primates. Overall, this will definitely be a valued resource for the community. However, there are several points that should be addressed.

Major concerns:

1) The authors have performed single-cell RNA-seq in several cortical areas and other brain regions. However, they focus only on interneurons. Since this is mainly a resource article, the impact of the paper would be tremendously increased if all the other cell types (excitatory neurons and their progenitors, non-neuronal cells, etc.) were reported. Is there any particular reason for the authors to select only inhibitory neurons? Furthermore, the methods section needs to be greatly expanded. The authors need to precisely describe the dissected areas at each developmental stage. I was particularly intrigued by the dissection of adjacent cortical areas, such as motor and somatosensory areas, in specimens from very early developmental stages. Furthermore, ideally there would be at least 2 specimens per developmental stage (only 2 stages have more than 1 specimen). I also didn't find any metadata regarding the specimens (only age is reported), sequencing (how many cells per sample passed qc; how many reads per sample, percentage of exonic, intronic, antisense reads, etc.), overall quality control measures, etc. These are all essential in a resource paper.

2) I think the analysis on the developmental origin and specification of TAC3 striatal interneurons should be expanded. In ED Fig. 4, there are several ISH from Allen's mouse atlas to show that the TAC3 population is likely generated in a boundary region. I think it is important to do a proper histological validation of this in monkey. I appreciate that it will be very hard, or even not possible, to do direct lineage tracing, but it should be easy to do the characterization of this specific region in the developing monkey using a combination of ISH and immunofluorescence. They should also show some validation for the mutually exclusive expression of neurexophilins between CRABP1/MAF and CRABP1/TAC3 neurons. Overall, I was hoping to find more histological validation of these initial classes as sometimes RNA profiles are not reflected at the protein level.

3) In line with my previous point on validation of findings, the authors should cite and incorporate the results of the article from Tong Ma et al. (Nature Neuroscience, 2013). In this article the authors did extensive profiling of the progenitors and early born interneurons in human and macaque, using histology and slice culture.

Minor concerns:

1) I would avoid stating that the initial classes of inhibitory neurons are largely conserved among mammals, though they most probably are. The authors only analyzed 2 species, and therefore should say instead that these classes are likely conserved among mammals.

2) In line 82, it should be Fig. 1e, not 1e.

3) The authors should describe what external LGE is (it's only described in the accompanying paper).

Referee #3 (Remarks to the Author):

- Provides strong evidence for its conclusions. (In part)
- Novel - (Yes). There are no reports that follow macaque interneuron developmental stages at the single cell level.
- Of extreme importance to scientists in the specific field. (Yes). Given that the majority of developmental work is focused in rodents, it is important to determine conservation with primates

and humans. This not only contributes to understanding evolutionary principles, but also functional significance. In the field of developmental biology, cellular heterogeneity has been especially difficult to define in primates, until the discovery of single cell technology. Developmental timing is important for understanding the establishment of brain circuits and defining windows of neuronal susceptibility to intrinsic and extrinsic factors, including diet and environmental toxins.

- Ideally, interesting to researchers in other related disciplines. (Yes) This work advances multiple fields and will be interesting to biologists in the fields of development, evolution, neuroscience and computation.

- Reject, but further work would justify a resubmission

- Key results:

Schmitz et al present a single cell gene expression analysis of primate inhibitory neurons from the beginning of embryonic neurogenesis (PCD40) to late fetal stages (PCD110). Based on computational reconstruction of trajectories, they identify initial classes of inhibitory neurons in macaque that correspond with those identified in mouse, aligning with spatial origin (lateral/ medial/ caudal/ ventromedial). They conclude that primate-specific diversity arises by refinement of conserved classes post mitotically, rather than early born differences. One exception is the primate specific TAC3 striatal interneurons, born during midgestation.

- Validity: The macaque developmental analysis is beautifully done. The number of individual samples, 4 different stages, regional sampling and the sheer number of cells sequenced (109,112) constitute a comprehensive undertaking, and support the validity of the computational analysis. The main concern is with the use of the published mouse datasets that have been combined, form the basis for comparisons, and also for some of the major conclusions. The mouse datasets are made up of

- 1) Dropseq dataset (PRJNA407881, MGE E13.5, CGE/LGE E14.5). In the extended data, these cells do not cluster with other cells. This is expected, since these were not generated using 10X genomics, as the rest of the datasets. Therefore, it would be very difficult to integrate and make valid comparisons. Nothing is mentioned about this difficulty.

- 2) Developmental 10X dataset (PRJNA411878). These are based on purified populations of Lhx6 E13.5 cells, Dlx6a+ E18.5 and P0 cells. Since these are purified, many populations would be removed.

- 3).10X demonstration- this is not explained.

- Originality and significance: Original and significant.

- Data & methodology: The approaches are cutting edge and figures of high quality.

- (1) There is some extended data that could be included in the main Figures (UMAP by timepoint, phase and Leiden, ext data Fig 1a-c). Explain Leiden clustering and numbers labeling the clusters in the methods.

- (2) Explain mathematical concepts/modeling - for every mathematical term- please explain why it was used and a reference. For instance, some may use Jaccard index, but may not know what it is called.

- (3) Provide labeled diagrams for all PCD stages.

- (4) Discuss caveats to using scVelo, and computational predictions of trajectories. (beyond just stating that claims of lineage are not made).

- Appropriate use of statistics and treatment of uncertainties: The statistical tests are referenced, but there could be better explanations for non-experts. The real power is in the number of total cells analyzed in the macaque studies.

- Conclusions: Do you find that the conclusions and data interpretation are robust, valid and reliable? Only partially: Yes- for the macaque dataset, but not for the mouse dataset due to reliance of existing incomplete/inappropriate mouse datasets.

- Suggested improvements: Please list additional experiments or data that could help strengthening the work in a revision.

- 1) Use 10X datasets to sample equivalent time points in neurogenesis in mice.
- 2) Remove mouse comparisons and formal analysis of mouse scSEQ, expand analysis and focus on macaque development.
- 3) One possibility is to combine with accompanying paper, removing the formal analysis of mouse data.
- 3) Discuss differences with human interneuron analysis.
- 4) Fig 3. h/i panels are confusing and need more explanation.
- 5). Line 42: 8, 12?
 - References: Acceptable
 - Clarity and context: Well written. However, explanation of mathematical terms/algorithms could be more clear.
 -
 -
 - ScSeq is a highly evolving field, with new types of analysis reported and larger and larger datasets. I cannot comment on the validity of using SCANPY, Kamada-kawai graphs, Holm-Sidak corrected q values, Scenic module scores.

Author Rebuttals to Initial Comments:

We would like to thank the reviewers for their kind words and constructive feedback. We believe that the suggestions have driven us to substantially improve the study, and that this work will be of broad interest to the developmental biology, evolution, and genomics communities. In particular we present a new approach to constructing a unified taxonomy of mammalian developmental cell types, uncover two contrasting examples of developmental mechanisms shaping cell type evolution, and provide evidence for a novel “reduce and reuse model” of redistribution of late born olfactory bulb-bound neurons to expanded primate white matter locations.

Upon consideration we agree with the reviewers that a combined manuscript allows us to leverage our dataset in both studies and provides more clarity throughout. In addition, presenting the two studies together allows us to more directly compare the contrasting developmental mechanisms of cell type evolution that our atlas reveals: early specification of an entirely novel initial class and redistribution of a conserved initial class to new locations.

In the combined manuscript format, we are also able to expand and draw upon the taxonomy throughout. To this end, we have expanded the analysis of mouse developmental datasets and adult taxonomies as suggested, which enabled us to make data-driven predictions about the relationship between initial and terminal classes of neurons and to anchor the developmental taxonomy in conserved cell types and gene networks.

Following the reviewers' direction, we have performed additional experimental validation and quantification of the primate origins of the novel TAC3+ striatal interneuron population. In addition, we have clarified the model of redistribution of olfactory bulb inhibitory neurons into the neocortex and striatum, and have shown that the TH+ peri-striatal cells, termed striatum laureatum neurons for the wreath they form around the striatum, persist until old age in humans, where they don't exist at all in mice.

Referee #1 (Remarks to the Author):

The paper by Schmitz et al. examines the origins of inhibitory neurons in primates compared to that seen in mice and provides a developmental extension of recent work examining the adult fates of these populations in primate by the McCarroll laboratory. The study confirms recent work in mice suggesting that cell type signatures in mitotic cells are weak. Nonetheless, based on transcriptional signatures resembling those seen postmitotically, they argue that there is some indication of differentiated character in inhibitory projection progenitors, but less so in the similarly derived interneuron populations. Even so the identities of these provisionally committed progenitors reflect weak TF signatures defined by broadly expressed factors such as NR2F1/F2, Mies2, Ebf1 and Isl1. Comparison to that seen in mouse showed strong similarities and interestingly better gene conservation in progenitors than seen in more mature populations, suggesting that developmental programs are more conserved than terminal fates. The study then focuses on the TAC3 striatal population that was recently identified as giving rise to a novel striatal interneuron population in primates and tentatively identifies the progenitor zone origins as a CRABP1+ MGE population, which they speculate diverges from the related CRABP1-MAF class. While the speculation here is intriguing, the over-reliance on maintained gene expression to infer relatedness places their conclusions on a weak footing. The narrative of the manuscript is also somewhat out of synch with the actual figures. For instance, Figure 2 extensively explores genes that are extinguished versus initiated across development as latent time extends, which falls short of making a compelling argument for the purported gene trajectories of specific subtypes. Figure 2d and e while making an argument for the

change in the developmental expression of class1 and 2 genes using regression across time and Jaccard indices present a confusing picture of dynamic genes in cell types whose meaning is poorly explained. In the end these trajectories while arguing for the trajectory of genes within subtypes does not convincingly validate their predictions.

We appreciate the reviewer bringing this shortcoming to light. To address the over reliance on maintained gene expression to infer relatedness, we have now used new analysis methods of dynamic gene expression (nearest neighbor analysis and absorption probabilities based on whole-transcriptome RNA velocity trajectories, Fig. 2) to infer data driven relationships. To address the writing shortcoming, we have moved the majority of the gene-level trajectory analysis into the extended data (ED8) where it can be better explored and explained, as it is not foundational to the key points of the study. In addition, we perform RNAscope and immunohistochemistry validation experiments for two striatal trajectories of interest that support gene-level predictions of sequential expression along putative migratory routes. For the FOXP2/TSHZ1 class, we observe that inactivating genes such as ETV1 and SP8 are expressed in the dLGE (Seen in ED7,11), early activating genes such as TSHZ1 show expression in the progenitor zone and in striatal eSPNs, and late activating genes CASZ1 and particularly OPRM1, show expression mainly in striatal eSPNs but are also seen in dLGE (ED9). Similarly, for TAC3 neurons, we find expression of MKI67 that inactivates during differentiation mainly in progenitor zones (ED7), expression of the early activating combination TAC3/CRABP1 rarely in dividing cells but more widely in the MGE progenitor zone, and expression of late activating STXBP6/CRABP1 mainly in the striatum. Thus, additional validation experiments support inferred trajectories for the populations we discuss, but we have also simplified the main text to focus on key findings - the unified taxonomy, the early specification of TAC3 striatal interneurons, and specialized primate features of dLGE lineages.

In Figure 3 the study focuses on the emergence for the TAC3 population and show good evidence for a conserved expression profile (Nkx2.1, Nxp1, Etv1) from macaque specific CRABP1 populations that are either TAC3 or MAF positive, with the former being novel and the latter being conserved. All of these findings while interesting are neither convincing enough in themselves based on the gene expression in populations or across time without further characterization or validation. to extend our understanding of them sufficiently to warrant publication in Nature.

We agree that the predictions require additional bioinformatic and experimental validation. We have significantly expanded both bioinformatic (Mutual nearest neighbors of extensive developmental and adult datasets Fig. 2, ED5) and tissue validation experiments (RNAscope qualitative and quantitative analysis, Fig 3 ED6,7) here to characterize the birthplace and early differentiation of these sister classes.

An argument could also be made for combining the two papers into a single concise manuscript as they make extensive use of the same data and as is each paper contains undescribed details relevant to the other paper (e.g. Figure 1a in the OB paper uses the same UMAP as in the initial classes paper, but the gray cells relevant to the classes paper are not described, and some of the same cell populations are included in analysis of both texts).

We agree and have attempted to distill the findings into a single manuscript and believe it is indeed higher quality as a single cohesive study building from the taxonomy to describe patterns of cell type evolution.

Comments

- This manuscript includes an extensive dataset from multiple ages and multiple structures, yet the analysis is not vigorous enough to justify the amount of data included. For example, the regions of the cells were only shown in Figure without much discussion about the interpretation and meaning. Also, putting the progenitor region CGE, MGE, LGE label together with other structures such as PFC, Striatum, etc.. in figure 1e is confusing if one wants to compare the relative contribution of each cell type to different regions.

We have attempted to add rich and interpretable bioinformatic analysis to the main text, while expanding our finer-grained gene-level bioinformatics resources during revision. We have added an extended data figure (ED1) displaying the rough dissociated cell composition of classes in each dissected region to address the reviewer's point about 1e (now in Fig 2), and we have substantially expanded our analysis of the developmental taxonomy, providing data driven linkages between initial and terminal classes in the updated Figure 2.

- Regarding the homology between mouse and macaque: 1. what ages are homologous across species and is it differ by cell type?

We have included the comparable human, macaque, and mouse stages across our experimental design as derived from models of neurogenesis using the Translating time resource and labeling these estimated stages below a schematic of macaque brain development (ED1).

2. are there major species differences in relative order or timing of major inhibitory classes by brain region? (line 94 suggests not but no citations or evidence from mouse are given).

We observe conservation in the broad sequence of neurogenesis - Cajal-Retzius neuron production at the earliest stages, spiny projection neuron production preceding granule cells, and an expansion of CGE neurogenesis later in development, which we now cite in ED1. We don't observe any major species difference in order or timing in our informatic analysis, however this is also a complicated question and probably requires intensive birthdating and molecular cell type characterization in macaque to adequately address. Therefore, we have moved this analysis to the Extended Data 1, as it is essentially qualitative and not the main focus of a scRNAseq study.

3. The datasets may be imperfectly matched across species. The paper includes data from multiple structures in macaque. It isn't clear if there are comparable mouse datasets.

We agree with this point and feel that our analysis has substantially improved by incorporating more comparable and comprehensive mouse datasets. In our initial submission mouse was undersampled relative to macaque at certain stages. However, we have now included extensive newly-available mouse data which we believe are both comparable methodologically, and are a superset of our macaque sampling. Given the size of the mouse brain, it is much easier to dissociate entire regions to get full compositional coverage. For example, the new Linnarson lab study dissociated all of ventral telencephalon as a single sample, whereas macaque experiments involved microdissection. Thus, we now put macaque in context of these massive mouse datasets allowing more vigorous analysis of cell type conservation and macaque cellular heterogeneity and updated analysis for the taxonomy of initial classes of inhibitory neurons. For example, we further employed mutual nearest neighbor analysis to identify homologous cells that were undersampled in the macaque dataset, revealing for example,

LGE_FOXP1/ISL1/NPY1R branch of ventral direct spiny projection neurons emerges in both mouse and macaque.

Information regarding mouse dataset were only briefly mention in Method, it will be clearer if timepoints and regions for mouse are also included Supplementary Table 1). 4. Is it possible to align mouse and macaque dataset in one umap space to provide a more direct illustration for the conclusion that the authors are trying to make, which is the conserved progenitor population that becomes more distinct between the two species later. I imagine mouse E13.5, E14.5 dataset would be the progenitor timepoint, and 10X mouse E18 cortex data could serve as a more diversified time point.

We thank the reviewer for this suggestion. We have now exhaustively analyzed available mouse datasets captured on the 10X Chromium platform from development (Fig 1, ED3) and adult (ED5) and more clearly describe these datasets in the methods, including providing predictions of developmental stage homology using the Translating Time resource (ED1). As mentioned above, the mutual nearest neighbor analyses of the macaque and mouse datasets enabled further resolution of several rare populations.

- Should make sure to clarify throughout the text when a population is given a spatial designation because it was dissected there or inferred to be of that location for other reasons. An example is RMTW_ZIC1/RELN (line 50). Overall, I find the naming conventions a bit confusing. E.g. the mouse Crabp1/Maf class does not seem to express much Crabp1 (Fig 3f)?

Recently, there has been a great deal of discussion about the guiding principles for molecular taxonomies of cell types, and one major principle that stands out is to name cell types based on evolutionarily conserved cell intrinsic properties (doi:10.7554/eLife.59928, Miller et al.) which is also consistent with the evolutionary definition of cell types (PMID: 27818507). We have therefore attempted to name initial classes based on conserved markers with high combinatorial specificity for each initial class wherein the combination of 2 markers uniquely identifies the class. For the mouse MGE_CRABP1/MAF class, Crabp1 shows high specificity even if the average level of expression appears lower in the dotplot (Fig 3, also see Extended Data 4 UMAP feature plots and differential expression supplementary table). This specificity is also more apparent in the larger collection of mouse datasets now available. For the spatial designation, we further describe the inference of the source in the methods. In brief, the spatial origin for each class was inferred based on the expression of canonical marker genes for RMTW, MGE, LGE, CGE, and VMF (e.g., FOXP1 plus LHX5, NKX2.1, MEIS2, NR2F2, ZIC1) and supported by immunostaining and by the enrichment of these genes in cells from region-specific dissections. In revision, we have further used RNAscope in macaque to incorporate additional genes in this analysis (ED7,9)

- The model for the origin of the TAC3 striatal type is interesting but it is unclear how the correspondence to the adult type is confirmed since macaque adult samples were not included in the analysis. Similarly, how is it determined that the TH and PVALB types emerge from the CRABP1/MAF class, and when do they become distinct? While the authors conclude that the TAC3 initial class does not exist at any point in mouse development, Fig 3a-f suggests that the mouse CRABP1/MAF type maps to both the macaque CRABP1/MAF and the CRABP1/TAC3 initial classes rather than selectively to the CRABP1/MAF class. That seems different than preservation of an ancestral class (CRABP1/MAF) + derivation of a new one.

Munoz-Manchado et al., 2018 (PMID: 30134177) first showed that a Crabp1-expressing class of mouse newborn neurons transcriptomically corresponds to PV and TH striatal interneurons in more

mature samples and Bengtsson Gonzales et al., 2020 (PMID: 32973206) further characterized the fast-spiking physiology and morphology of the corresponding PV-class. Our updated analysis of the most complete developmental and adult mouse datasets available (Fig 2, ED3-5) now help us to answer this question and further support the linkage of the mouse CRABP1/MAF class to PV, PTHLH and TH striatal interneurons by nearest neighbor and RNA velocity analysis. Comparable adult primate datasets including TAC3 cells in striatum captured by the 10X Chromium controller were unavailable as the Krienen dataset includes marmoset and human striatum data captured by Drop-seq (and only cortical data but not striatal data from macaque). However, we now more clearly show the preservation of marker genes from the developing initial class to the adult mouse population (Fig. 2) and we further confirm the existence of a rare LHX8/TAC3 subpopulation predicted transcriptomically in adult marmoset by using RNAscope in prenatal macaque striatum (Fig. 3). In addition, the macaque CRABP1/TAC3 class shows a lower correlation than the macaque CRABP1/MAF class to the mouse CRABP1/MAF class (Fig. 3a) and showed more marker gene divergence (ED4b), supporting increased divergence of the primate-specific TAC3 class.

- I found ED5b difficult to understand because so many structures are plotted in different colors. E.g. the CRABP1/MAF cells in xLGE – are they presumably migrating?

The CRABP1/MAF cells in this anatomical location may or may not be migrating, but we now confirm their presence throughout the striatum. This sample was derived from the anterior end of the dLGE (xLGE), and was likely dissected along with some amount of the anterior MGE and the caudate head. RNAscope now confirms the presence of the CRABP1/MAF and TAC3 population in anterior caudate. This figure is now ED1 and includes a new schematic of the overall design, complementing Fig 1a and further visualizations of the cell type clusters across ages and regions

Figure 1: UMAPS should be horizontally inverted, and plane of coronal section should be shown in sagittal views.

We agree that these suggestions improve readability and now show the independently-generated mouse and macaque UMAPs in a similar orientation in figure 1, and we mark the plane of sectioning.

Figure 2: macaque Model in b is intriguing. Do they have putative labels for cells at each of these stages for different cell types? I assume the class 1 and class2 is something akin to Jessell's Shh response genes but in time rather than space, but this is neither clear nor well spelled out in the text. There is a good idea here but it is lost on me.

This cartoon of the model was indeed confusing. The Class1 and Class2 schematized scatter plots and Venn diagrams were meant to illustrate possible patterns of covariation in gene expression dynamics between any pair of initial classes. This model is now simplified and better explained in Extended Data 8 to address the extent of shared versus cell type-specific trajectories during post-mitotic differentiation of initial classes.

Figure 3: Not sure what I am supposed to get from A: B says it is comparison of 1000 genes but each of these are mixed so more like paralog cells comparisons than genes. This graph is a real mix of Mge, Striatum and

The reviewer is correct – the heatmap shows the correlation of vectors for each cell class, with 6,227 (previously 1000) 1-to-1 ortholog gene expression values per class in each vector supporting paralogous class predictions between species. This is now better explained in the methods, improved

with more exhaustive mouse datasets, and complemented by an independent mutual nearest neighbor analysis at the level of individual cells (Figure 2).

ED2a – As presented, it is difficult to tell when a population is not present because of the developmental stage or because the structure was not sampled in that timepoint.

We agree and have added "region" vs "time point" and "class" vs "region" bar graph to this figure in order to help alleviate this issue. (see ED 1).

Referee #2 (Remarks to the Author):

In the article “Atlas of Initial Inhibitory Neuron Classes During Primate Cerebral Development”, Schmitz et al. describe the molecular logic of inhibitory neuron specification during macaque early-fetal development. Using their newly generated single-cell RNA-seq dataset and integrating it with available datasets from mouse and more mature macaque samples, they show that almost all the initial classes of early born inhibitory neurons are conserved between macaque and mouse. This indicates that the previously reported increased diversity of inhibitory neuron populations in primates is likely due to extrinsic factors that occur later, from midfetal to early postnatal periods. After presenting their resource, they then analyze whether the recently reported primate-specific population of striatal TAC3 interneurons was transiently present or completely absent during mouse development. They report that this population cannot be detected throughout development in mouse and that it's actually specified early in primates. Overall, this will definitely be a valued resource for the community.

We thank the reviewer for these kind words.

However, there are several points that should be addressed.

Major concerns:

1) The authors have performed single-cell RNA-seq in several cortical areas and other brain regions. However, they focus only on interneurons. Since this is mainly a resource article, the impact of the paper would be tremendously increased if all the other cell types (excitatory neurons and their progenitors, non-neuronal cells, etc.) were reported. Is there any particular reason for the authors to select only inhibitory neurons?

We have focused on inhibitory neurons as a tractable system to study in detail, because including all other cell types would have led to a higher-level study with less depth on particular lineages of evolutionary interest, because recent studies have focused on the role of cortical area in influencing excitatory development, and because we predicted that inhibitory neurons that migrate long distances might be particularly influenced by brain reorganization and unequal scaling of regions during primate evolution. Our focus on inhibitory neurons allowed us to describe two contrasting developmental mechanisms contributing to cell type evolution in the brain that, particularly after combining the two manuscripts, adds biological themes beyond the resource aspect. We have further added Supplemental Tables 1 and 2 which address quality control metrics and updated the browsable online portal from which all processed data can be downloaded to buttress the resource aspect of the study.

Importantly, a further study including the whole brain with all cells is in progress as part of a BRAIN Initiative collaboration and will be released as the complete and unified dataset.

Furthermore, the methods section needs to be greatly expanded. The authors need to precisely describe the dissected areas at each developmental stage. I was particularly intrigued by the dissection of adjacent cortical areas, such as motor and somatosensory areas, in specimens from very early developmental stages. Furthermore, ideally there would be at least 2 specimens per developmental stage (only 2 stages have more than 1 specimen). I also didn't find any metadata regarding the specimens (only age is reported), sequencing (how many cells per sample passed qc; how many reads per sample, percentage of exonic, intronic, antisense reads, etc.), overall quality control measures, etc. These are all essential in a resource paper.

We agree that a more extensive description of the metadata and methods is required. We have compiled supplementary tables with these metadata at the level of both samples and individual cells and have gone into more depth in the methods. We agree about the importance of replication and designed our study around the tradeoff of including replicates (3 PCD65 replicates and 2 PCD80 replicates) while also spanning stages of neurogenesis and minimizing use of non-human primate specimens. We also note that populations from early stages (PCD40 and PCD50) and late stages (PCD90 – PCD110) also show strong overlap between nearby timepoints (and between labs). To the dissection point, the central sulcus was not clearly visible at the earliest timepoints and presumptive regions were dissected, addressed in the text as: "A number of regions were difficult to distinguish at earlier timepoints as key anatomical landmarks are still forming, and so presumptive regions were dissected (e.g. motor vs somatosensory cortex prior to appearance of the central sulcus or the anterior end of the MGE and LGE)".

2) I think the analysis on the developmental origin and specification of TAC3 striatal interneurons should be expanded. In ED Fig. 4, there are several ISH from Allen's mouse atlas to show that the TAC3 population is likely generated in a boundary region. I think it is important to do a proper histological validation of this in monkey. I appreciate that it will be very hard, or even not possible, to do direct lineage tracing, but it should be easy to do the characterization of this specific region in the developing monkey using a combination of ISH and immunofluorescence. They should also show some validation for the mutually exclusive expression of neurexophilins between CRABP1/MAF and CRABP1/TAC3 neurons. Overall, I was hoping to find more histological validation of these initial classes as sometimes RNA profiles are not reflected at the protein level.

We agree and have substantially expanded both the bioinformatics using mutual nearest neighbors analysis (Fig. 2) and tissue validation experiments using RNAscope (we were unable to find working TAC3 antibodies in the macaque) and manual cell counting (Fig. 3, ED 6-8). In absence of lineage tracing, we have attempted to determine where newborn striatal interneurons are migrating from and performed detailed quantification of the MGE and striatum separately across the rostrocaudal expanse. We have further performed validation of additional markers of these classes by RNAscope including STXBP6, ANGPT2, and RBP4 (see Extended Data 7). We agree that protein-level validation would be optimal in these classes for many of the interesting molecular characteristics and note that we were able to perform protein-level validation on the LGE initial classes and their derivatives due to the availability of relevant and working antibodies.

3) In line with my previous point on validation of findings, the authors should cite and incorporate the results of the article from Tong Ma et al. (Nature Neuroscience, 2013). In this article the authors did extensive profiling of the progenitors and early born interneurons in human and macaque, using histology and slice culture.

We appreciate the reviewer's suggestion and now cite this important paper on primate interneuron domains. We had referenced study for determining the anatomy initially and we apologize for overlooking including it in the reference list.

Minor concerns:

1) I would avoid stating that the initial classes of inhibitory neurons are largely conserved among mammals, though they most probably are. The authors only analyzed 2 species, and therefore should say instead that these classes are likely conserved among mammals.

The reviewer is correct, and we have fixed our language accordingly to describe our result of conservation between macaque and mouse. It will be extremely interesting for future studies to examine whether these initial classes are conserved in additional and more distant mammalian orders.

2) In line 82, it should be Fig. 1e, not 1e.

Thank you!

3) The authors should describe what external LGE is (it's only described in the accompanying paper).

This has been resolved by merging the two manuscripts.

Referee #3 (Remarks to the Author):

- Provides strong evidence for its conclusions. (In part)
- Novel - (Yes). There are no reports that follow macaque interneuron developmental stages at the single cell level.
- Of extreme importance to scientists in the specific field. (Yes). Given that the majority of developmental work is focused in rodents, it is important to determine conservation with primates and humans. This not only contributes to understanding evolutionary principles, but also functional significance. In the field of developmental biology, cellular heterogeneity has been especially difficult to define in primates, until the discovery of single cell technology. Developmental timing is important for understanding the establishment of brain circuits and defining windows of neuronal susceptibility to intrinsic and extrinsic factors, including diet and environmental toxins.
- Ideally, interesting to researchers in other related disciplines. (Yes) This work advances multiple fields and will be interesting to biologists in the fields of development, evolution, neuroscience and computation.

We thank the reviewer for the kind words.

- Reject, but further work would justify a resubmission
- Key results:

Schmitz et al present a single cell gene expression analysis of primate inhibitory neurons from the beginning of embryonic neurogenesis (PCD40) to late fetal stages (PCD110). Based on computational reconstruction of trajectories, they identify initial classes of inhibitory neurons in macaque that correspond with those identified in mouse, aligning with spatial origin (lateral/ medial/ caudal/ ventromedial). They conclude that primate-specific diversity arises by refinement of conserved classes post mitotically, rather than early born differences. One exception is the primate specific TAC3 striatal interneurons, born during midgestation.

- Validity: The macaque developmental analysis is beautifully done. The number of individual samples, 4 different stages, regional sampling and the sheer number of cells sequenced (109,112) constitute a comprehensive undertaking, and support the validity of the computational analysis. The main concern is with the use of the published mouse datasets that have been combined, form the basis for comparisons, and also for some of the major conclusions. The mouse datasets are made up of

1) Dropseq dataset (PRJNA407881, MGE E13.5, CGE/LGE E14.5). In the extended data, these cells do not cluster with other cells. This is expected, since these were not generated using 10X genomics, as the rest of the datasets. Therefore, it would be very difficult to integrate and make valid comparisons. Nothing is mentioned about this difficulty.

We thank the reviewer for these kind words about the macaque developmental analysis and for the encouragement to improve our analysis of comparable mouse datasets. We agree that platform differences could influence co-clustering and differential expression. We have now elected to focus on mouse datasets generated using 10X genomics to reduce batch effects. Therefore, we no longer include the PRJNA407881 drop-seq dataset. During revision, additional large-scale mouse 10X datasets became available that allowed us to more extensively analyze mouse developmental stages and regions while maintaining uniformity of experimental platforms. In addition, we have uniformly reprocessed all datasets and used Cellbender+BBKNN for overcoming batch effects in coclustering. By incorporating these additional mouse datasets the overall level of our analysis and resulting taxonomy has improved substantially.

2) Developmental 10X dataset (PRJNA411878). These are based on purified populations of Lhx6 E13.5 cells, Dlx6a+ E18.5 and P0 cells. Since these are purified, many populations would be removed.

We agree and we now note the origin of this dataset more clearly in the methods. We cast a wide net across mouse datasets including this dataset that involved enrichment strategies as well as datasets that captured cells without enrichment to exhaustively sample rare populations and to test for the presence of a rare transient developmental TAC3 population, which we were unable to detect. It is true that the PRJNA411878 dataset may have compositional differences due to cell selection. However as batch composition biases are commonplace in whole cell dissociations for droplet-based single cell RNA-seq, we refrain from all but the coarsest compositional conclusions based upon the single cell RNA-seq data, and we believe this single dataset composition difference does not have detrimental effects on the analysis in the context of the multiple larger datasets we have now compiled.

3).10X demonstration- this is not explained.

This is the E18 mouse cortical neuron data made available from 10X's website as a demonstration dataset and SRA, and is now further explained with its proper accession (GSE93421) in methods.

- Originality and significance: Original and significant.
- Data & methodology: The approaches are cutting edge and figures of high quality.

We thank the reviewer for these kind words.

(1) There is some extended data that could be included in the main Figures (UMAP by timepoint, phase and Leiden, ext data Fig 1a-c). Explain Leiden clustering and numbers labeling the clusters in the methods.

(2) Explain mathematical concepts/modeling - for every mathematical term- please explain why it was used and a reference. For instance, some may use Jaccard index, but may not know what it is called.

We agree and we have moved the UMAP colored by timepoint to Figure 1. We also agree that the initial Leiden clustering would have fit well in the main text with the two manuscript format. However we have elected to leave this plot in the extended data because of the space limitations of the single condensed manuscript, and because subsequent analysis is based upon the distilled initial classes rather than direct Leiden clusters that mainly reflect stages of differentiation within an initial class. We have made the methods section more extensive, and have included a more complete description of these methods.

(3) Provide labeled diagrams for all PCD stages.

We have provided this in Extended Data Fig. 1.

(4) Discuss caveats to using scVelo, and computational predictions of trajectories. (beyond just stating that claims of lineage are not made).

We agree that computational predictions of lineage relationships based on transcriptional trajectories are hypothesis-generating and that claims of lineage will ultimately require lineage-tracing. We have updated the unified taxonomy figure panel to explicitly highlight predictions that need to be tested and those that have support from prior lineage tracing studies. We have also added the caveat "that transcriptional similarities affect the predictions, while chromatin or lineage relationships are not observed" We have additionally added a new extended data figure co-embedding developmental and adult datasets and linking these populations by a combination of scVelo and nearest neighbor relationships and by more clearly explaining these updated methods.

- Appropriate use of statistics and treatment of uncertainties: The statistical tests are referenced, but there could be better explanations for non-experts. The real power is in the number of total cells analyzed in the macaque studies.

Thank you for this suggestion, we have tried to rewrite these sections to make them more accessible! By moving the inferred gene expression trajectories and their correlations between cell types to the supplement, we have created additional space to explain these analyses while simplifying the main text to the key points from both initial manuscripts.

- Conclusions: Do you find that the conclusions and data interpretation are robust, valid and reliable? Only partially: Yes- for the macaque dataset, but not for the mouse dataset due to reliance of existing incomplete/inappropriate mouse datasets.

- Suggested improvements: Please list additional experiments or data that could help strengthening the work in a revision.

1) Use 10X datasets to sample equivalent time points in neurogenesis in mice.

We agree that the mouse datasets initially available undersampled certain timepoints and required mixing data from 10X Chromium and drop-seq approaches to cast the widest net.

The newly included Linnarson whole brain developmental (fig. 1) and adult datasets, along with the Konopka mouse striatum (ED5) and Camp organoid data (ED6) have strengthened the analysis of conserved initial classes and of linkages to adult populations.

2) Remove mouse comparisons and formal analysis of mouse scSEQ, expand analysis and focus on macaque development.

3) One possibility is to combine with accompanying paper, removing the formal analysis of mouse data.

We have chosen both to combine with the accompanying paper and to strengthen the mouse scRNA-seq.

3) Discuss differences with human interneuron analysis.

We appreciate this suggestion, but we don't believe there is a comparable human dataset available at this time that we can access.

4) Fig 3. h/i panels are confusing and need more explanation.

5). Line 42: 8, 12?

- References: Acceptable
- Clarity and context: Well written. However, explanation of mathematical terms/algorithms could be more clear.
-
-
- ScSeq is a highly evolving field, with new types of analysis reported and larger and larger datasets. I cannot comment on the validity of using SCANPY, Kamada-kawai graphs, Holm-Sidak corrected q values, Scenic module scores.

We have added additional language in the methods to describe each bioinformatics analysis more clearly.

Your manuscript entitled "Re-use of Olfactory Bulb-Bound Neurons in Expanded Primate Cortex and Striatum" has now been seen by 3 referees, whose comments are attached below. While they find your work of potential interest, as do we, they have raised important concerns that in our view need to be addressed before we can consider publication in Nature.

Referee #1 (Remarks to the Author):

The analysis of olfactory-directed neurons that are shown to populate in cortex and striatum extends the previous work of the Alvarez-Buylla laboratory investigating how complications in the rostral migratory stream of primates shunts cell types with distinct expression patterns to various locations within the mature white matter, claustrum of the frontal cortical and striatal structures in primates. The study identifies how four subcortical populations are positioned within LGE, which consists of two that are well-conserved classes and make the direct and indirect striatal populations in what they call the iLGE. Two additional groups arise from a novel extension of the LGE that is apposed to the external capsule (which they name the xLGE) appear to be unique to primate and in addition to the expected interneuron populations populate the frontal lobe. These represent neuronal types that through complex migratory streams utilize conduits to ultimately settle within the forebrain (rather than in the olfactory bulb, as dLGE neurons do in mouse). The description of cell types makes extensive use of previous mouse work that described gene expression of progenitors in the LGE and links their findings to weakly TH expressing populations comprised of FoxP2/TSHZ1 and MEIS2/PAX6 cells. These populations they posit are related to the so-called eccentric SPN populations discovered in adult mouse striatum (Saunders et al. 2018) and to the periglomerular population in mouse. The authors examine the distribution of the xLGE population to the cortex, which is characterized by

LGE_FOXP/TSHZ1 which they deem to be similar to the striatal eccentric cells but expressing opiate receptors. They also report a MEIS2/PAX6 population that contributes to the ARC structure previously described in humans that arise from the A-xLGE which they term TH+ PGCs. Hence, they propose the identification of three populations within the frontal lobe arising from either FoxP2/TSHZ1 or MEIS2/PAX6 and diversifying based on markers into a number of subpopulations.

This is a solidly done anatomical study that expands our understanding of frontal lobe cells associated with rostral migration and the ARC. While the progression of markers of each are plausible, their existence relies on unique molecular combinations to identify them as functional classes. However, no real characterization of these classes, their connectivity or their functional utilization is identified, making it difficult to infer whether these are truly primate specific populations or less heterogeneous neuronal subtypes with variance in expression of a few key markers (something that is well known to occur across species). Hence, while they may ultimately be shown to have functional significance, the present study is entirely descriptive and makes at best an incomplete argument for the diversity they propose. The authors conclude that these may represent Crick's speculated binding neurons. While this may be right, their neuro-molecular characterization falls short of making a compelling case for their importance.

We thank the reviewer for their kind words about how the study expands our understanding of frontal lobe cells associated with rostral migration and the Arc. We agree that further neuro-molecular characterization of the TH+ SLNs and Arc-derived deep white matter interstitial neurons will be important for the field and deserves follow-up studies given the functional significance of white matter neurons and their evolving distribution. However, we believe that the characterization of these cells presented here at the level of molecularly-defined populations and migratory routes is novel and will enable these future studies, and that the further characterization of their connectivity and functional utilization in macaque is beyond the scope of the merged and condensed single manuscript. We agree that marker genes may turn over between species. Therefore, we have now conducted extensive analysis of additional developmental (Fig 1-2, ED3,4,5) and adult mouse single cell transcriptional datasets and additional staining in mouse (Fig. 5, ED13). Our results support the interpretation that the LGE_MEIS2/PAX6 initial class is conserved between species but differs in primates by 1) undergoing chain migration in a histologically distinct structure into cortical deep white matter and 2) that the co-expression pattern observed in primate striatum and claustrum is only observed in mouse olfactory structures suggesting this initial class migrates to a distinct location in primates.

The unexpected, common origin of the eccentric striatal population and OB PGC (and amygdala ITCs) should be more fully characterized. First, are adult/postnatal data used to established that the LGE-FOXP2/TSHZ1 class definitively becomes the eccentric population? Second, the hypothesized shared origin between the eccentric striatal type and the amygdala Foxp2+ ITCs is intriguing should be explored more comprehensively. Fate mapping (in mouse) could confirm that ITCs and eccentric cells emerge from a shared origin.

We have now incorporated postnatal and adult data as suggested using data from the whole mouse forebrain and applied RNA velocity to attempt to address the question of the FOXP2/TSHZ1-eSPN-ITC connection more systematically and explicitly. We find that ITCs and eSPNs in fact co-cluster in the adult Linnarson + Konopka data with a single cluster containing cells from amygdala and striatum samples called BN-eSPN_FOXP2/TSHZ1 (ED5). We further performed RNAscope analysis in macaque of key markers that correspond to the inferred eSPN trajectory, including ETV1, TSHZ1, CASZ1, and OPRM1 and we confirm ETV1 and TSHZ1-expressing cells at the dorsolateral portion of the dLGE and identify cells expressing TSHZ1, CASZ1, and OPRM1 at both the dorsolateral edge of dLGE and in the striatum (ED8). We also agree that Cre fate mapping in mouse is a worthwhile direction. However, our biological findings in the combined manuscript focus on contrasting developmental mechanisms for inhibitory neuron diversification between primate and

rodent. Therefore, lineage-tracing of the evolutionarily conserved eSPN population, though of general interest for understanding the recently-discovered striatal population, may be beyond the scope of the current study, however we note that the Campbell lab has published developmental transcriptomic and lineage tracing data supporting the migration of a dLGE population with many of the same markers along the LMS to amygdala. Our updated analyses linking initial and terminal classes further generate many hypotheses which warrant separate focused studies in mouse building on the updated informatic and in situ hybridization analyses now presented here.

The prenatal anatomic distribution of the LGE_MEIS2/PAX6 is extensively described but I am left with an incomplete understanding of this population. First, are they homologous to the PAX6+ interneurons described in various Allen Institute taxonomies? Do they persist in the adult primate in large numbers? In general the manuscript would benefit from more concise description and presentation of the anatomy, quantification of the populations over time, and clearer analyses demonstrating how they become the previously characterized adult populations.

We have now more systematically explored the relationship between our atlas of initial classes and with existing adult ontologies (Fig. 2, and the initial and terminal class dictionary in Supplemental table 4), noting that early postnatal intermediate stages have not yet been explored. The MEIS2/PAX6 class or LGE_FOXP2/TSHZ1 appears to be homologous to the MEIS2+ "Interstitial" GABAergic cells identified by the Allen taxonomy rather than to the PAX6+ (Ctx_NR2F2/PAX6) (presumed CGE or POA/H derived) cortical cells. This MEIS2+ population forms a single outgroup cluster to other inhibitory neurons in the Allen taxonomy of cortical populations, and wasn't found in the available MOp 10X data, but our analysis of developmental structures including telencephalon identifies two initial classes MEIS2/PAX6 and FOXP2/TSHZ1 migrating through the cortex, and demonstrates that each population reaches diverse destinations including deep and superficial layers of primate white matter, respectively in juvenile primate. These white matter populations may have limited sampling in other atlases and analyses that focus on cortical grey matter.

We also agree that the question of whether these rare MEIS2 neurons persist in adult primate is important. We were able to obtain a frontal lobe post-mortem brain sample from an 88 year old human to assess whether these cells persist throughout adult life in primate and we confirmed the presence and persistence of TH+SLNs in human through old age (Fig. 5g-h).

We agree that our initial text used too much space to describe the anatomy. In this merged format, we believe that the presentation of the anatomy is clearer and more concise allowing us to focus on key results and making the overall study accessible to a broader readership (Fig 4-5, ED10-14). We have also performed new quantification for the CRABP1+ striatal interneurons across the rostrocaudal extent of the MGE (Fig. 3) in order to identify birth domains and the striatum destination of these populations.

In reducing the space dedicated to anatomy, we have also elected to use standard anatomical term dorsal LGE or dLGE, rather than to define the related term external LGE or xLGE as this will avoid confusion about terminology and allows more focus on the cell type findings.

Fig 1d: Is the correlation across species conducted including all ages? And are cells dissected from the LGE or from other structures (e.g. is the mouse LGE-OB_MEIS2/PAX6 cluster sampled from the OB or from the LGE?)?

The correlations across species includes all of the developmental cells, which includes all non-adult ages, plus newborn adult granule cells (LGE-OB_MEIS2/PAX6).

Fig 2: Some text is too small

We have increased the size of text here.

Fig 2b: Compares adult mouse striatal cell types to developing macaque striatal cell types. Maybe need adult primate striatal or mouse developmental striatal? Were the cross-species populations matched using expression of all genes or just the depicted markers?

We agree with the reviewer that matching homologous cell types across both species and developmental stage simultaneously is challenging and we thank the reviewer for the suggestion to link by stage and species separately using the expression of all genes. Therefore, we have expanded this analysis by incorporating mouse developmental and adult striatal data. Incorporating this data allowed us to utilize current cutting-edge methods for matching cell types between species at the same developmental stage (both transcriptional correlations at the cluster level across highly variable genes and mutual nearest neighbors at the individual cell level), and separately between species of the same developmental stage (RNA velocity combined with nearest neighbor relationships) (Fig 2). And per a previous reviewer comment we have required all datasets to be collected on the same 10X Chromium platform. This allowed us to identify data-driven relationships between the developmental LGE initial classes and the adult spiny projection neuron subtypes, whereas the previous heatmap highlighted the conspicuous intersection of marker genes for each correspondence but without a statistical comparison.

Fig 4a – where is remnant RMS?

This location is now clarified in the Figure 5 (ED box) and ED14.

Fig 4h – what is SWMIN?

Good question, this acronym was left undefined but referred to rare superficial white matter inhibitory neurons (SWMINs) we observed in human and macaque expressing markers of the FOXP2/TSHZ1 initial class. These acronyms are clarified in the new manuscript (Abbreviation section) and figures, and we defined a terminal class SWMIN_FOXP2/TSHZ1.

Referee #2 (Remarks to the Author):

Provides strong evidence for its conclusions. (Yes)

- Novel - (Yes). Cortical expansion and reduced olfactory bulb size are evolutionary features of primate brains. While the consequences of this evolution (more thinking, less olfaction) are clear, how this may have happened is not understood. In this work, the authors combine high-tech computational methods in the field of single-cell sequencing and anatomical reconstructions to identify primate-specific migrations of olfactory bulb-like LGE-born neurons into expanded striatum and white matter. They propose that during evolution of the primate brain, such diverted migrations may have also contributed to the early postnatal cortex.
- Of extreme importance to scientists in the specific field. (Yes). Given that the majority of neurodevelopmental work is focused in rodents, it is important to determine conservation with primates. This not only contributes to understanding evolutionary principles, but also functional significance. With advancing technologies in single cell sequencing, it has become possible to link cellular origins in different species, providing the opportunity to define commonalities and differences. While causality is difficult to determine in primates, the authors use neuronal classification to form a

hypothesis for brain evolution. Ultimately, these studies are critical for understanding the establishment of brain circuits and defining windows of neuronal susceptibility to intrinsic and extrinsic factors, including diet and environmental toxins.

We are grateful for these kind words.

- Ideally, interesting to researchers in other related disciplines. (Yes) This work advances multiple fields and will be interesting to biologists in the fields of stem cells and development, evolution, neuroscience and computation.
- Accept after addressing points
- Key results:

The authors combine high-tech computational methods in the field of single-cell sequencing and anatomical reconstructions to identify primate-specific migratory streams of interneurons. While primate and rodent inhibitory interneurons are born in embryonic lateral ganglionic eminences (LGE) and share molecular signatures, destinations are shifted. In the accompanying manuscript, Schmitz et al present a single cell gene expression analysis of primate inhibitory neurons from the beginning of embryonic neurogenesis (PCD40) to late fetal stages (PCD110). Here, they analyze 4 initial classes of LGE-born macaque inhibitory neurons, and find that two classes (LGE_FOXP2/TSHZ1 and LGE_MEIS2/PAX6) are enriched in the cortex compared to subcortex. (Fig 1c). Furthermore, they find that LGE_FOXP2/TSHZ1 and LGE_MEIS2/PAX6 are similar to mouse LGE-OB_MEIS2/PAX6, while the other two LGE-born classes (LGE_FOXP1/ISL1 and LGE_FOXP1/PENK) are not. Experiments to determine the location of

these rodent olfactory bulb-like interneurons (LGE_FOXP2/TSHZ1 and LGE_MEIS2/PAX6) in primate brains leads to identification of three distinct primate populations (rare superficial white matter neurons, [homologous to eccentric spiny neurons], medial frontal lobe deep white matter neurons, and a primate-specific population resembling olfactory dopaminergic neurons adjacent to the striatum.

- Validity: The macaque scSeq analysis is based on a number of individual samples, 4 different stages, regional sampling. The sheer number of cells sequenced (109,112) constitutes a comprehensive undertaking, and support the validity of the computational analysis. Further validation of cellular identities are based on high quality immunohistochemical co-labelling of subclasses of olfactory bulb-like interneurons in primate sections.

- Originality and significance: Original and significant.
- Data & methodology: The approaches are cutting edge and figures of high quality.

- Appropriate use of statistics and treatment of uncertainties: The statistical tests are referenced, but there could be better explanations for non-experts in the methods. The real power is in the number of total cells analyzed in the macaque studies.

- Conclusions: Do you find that the conclusions and data interpretation are robust, valid and reliable?
Yes

We thank the reviewer for these kind words.

- Suggested improvements: Please list additional experiments or data that could help strengthening the work in a revision.

Minor points:

1) line 49: TH expressing in Fig 1c not Fig 1b?

Thank you for catching that!

2) fig 1c: subcortex not subortex

Also thank you!

3) line 53 – confusing ref to Fig 1c? dLGE and pLGE-1 are not in the Fig

4) All 4 LGE initial classes are Meis2+ - why is one class called Meis2/Pax6?

We struggled to come up with the simplest names for the initial classes because many cell types are defined by combinatorial gene expression. The reasoning for this one is that the combination of the two named markers (along with DLX genes) are among the top markers of the class and are able to distinguish this class from PAX6+ CGE interneurons, and these markers are shared across mouse and macaque. We also picked markers for which there were high quality antibodies, where possible. The marker heatmap of Figure 2 (and ED4) and the gene expression signatures included in the taxonomy schematic of Fig 2 are intended to help understand the naming, and hopefully are helpful for readers to choose minimal sets of markers to identify the initial classes. We have preserved these plots in the merged manuscript. Ultimately two genes out of many markers is somewhat arbitrary and so we have now also included a supplementary "dictionary of classes" table and tables of differentially expressed markers, which will hopefully help others match to the cell classes we describe either qualitatively or quantitatively.

5) line 31: Fig 4a-d

Also thank you!

6) Explain mathematical terms and concepts in greater depth in the methods section.

We have taken care to expand this section and improve the clarity here.

- References: Acceptable
- Clarity and context: Check for accuracy with respect to references to main figures. Some of the figure labels are too small to read (ie Fig 2b, needed to zoom in 300X).

Referee #3 (Remarks to the Author):

In the article "Re-use of Olfactory Bulb-Bound Neurons in Expanded Primate Cortex and Striatum", Schmitz et al. profile the molecular identity of the lateral ganglionic eminence (using the data from the accompanying resource paper) and the interneuron populations that are generated in this structure. They also investigate, using immunohistochemistry, the migratory streams that are used by LGE interneurons, especially the ones that are born in a demarcated region they call external LGE. Whereas the accompanying article is focused mostly on the initial classes of inhibitory neurons, this

manuscript highlights how the xLGE is an important source of cortical, striatal, and white matter interneurons in macaque. This is opposed to the dorsal LGE in mouse (which mostly corresponds to the structure the authors call xLGE), which generates interneurons mostly to the olfactory bulb, though some are also bound to the cortex and the striatum. They report how different subpopulations of interneurons (superficial white matter neurons, intercalated cells of the amygdala, eccentric spiny neurons, and certain olfactory bulb neurons) are likely derived from the same population of FOXP2/TSHZ1 interneurons. The subpopulation of eccentric spiny neurons that phenotypically converges to the two classical medium spiny neurons is particularly intriguing. They also show that MEIS2/PAX6 interneurons also migrate from the anterior xLGE preferentially to the prefrontal cortex and the border of the striatum. Interestingly, these interneurons express tyrosine hydroxylase and may, therefore, be another dopamine source in primates.

Together, the two studies highlight the complexity of inhibitory neurons specification and migration in macaque, when compared to mouse. Whereas the other manuscript shows mostly cell intrinsic programs that are important for the specification of inhibitory neurons, this manuscript shows that later, likely cell extrinsic programs also play a role in diversifying the populations of interneurons in several brain structures.

We thank the reviewer for these kind words and for appreciating these contrasting developmental mechanisms contributing to inhibitory neuron diversification. By merging the two manuscripts we are now able to compare and contrast these intrinsic and likely extrinsic programs.

Overall, I found some of the figure panels very small and very hard to properly observe.

We appreciate this feedback and we have done our best to maximize the size and clarity of images in this version, especially in Figure 4 and 5. Further distillation of the anatomy in the main text allowed us to expand the key main text figure panels.

One intriguing thing I could observe was that it seems that the peri-striatal TH+ cells look different from the other TH populations the authors report. Whereas in the claustrum and the ventral olfactory nucleus, TH staining can be seen both in the cell body and the neuronal processes, the peri-striatal cells seem to have TH restricted to the cell body. This should be discussed, because if TH is restricted to the cell body, these cells will probably not be dopaminergic.

This is an interesting point, it did appear that some of the cells had TH+ processes while others it was very difficult to tell. We now show the unmerged images in the extended data (ED14) to highlight examples of both TH+ and TH- fibers in these locations, including in the same cell (ED14c).

Minor comments:

1) As in the other paper, the authors should refrain from saying that what they are reporting is primate-specific. They are only analyzing macaque and human (two primates) and mouse (non-primate). We don't know whether these populations and migratory streams exist in other mammalian species with large brains.

We agree with the reviewer - the analysis of human, macaque, and mouse differences does not enable determination of whether a trait is a derived primate gain or derived mouse loss. We have adjusted our language accordingly to acknowledge that we have only analyzed species within Euarchontoglires in this study. We note that for the TAC3 population, the absence of comparable cells

in adult mouse and ferret (detailed by Krienen study) supports a derived gain in primates as most parsimonious.

2) The authors should report how they classified cells as FOXP2 high vs FOXP2 low.

We observed in the gene expression data that *FOXP2* expression is robust in the FOXP2/TSHZ1 population while only detectable at low levels in some cells of the MEIS2/PAX6 population. The immunofluorescence images supported strong FOXP2 expression at the protein level in the FOXP2/TSHZ1 cells (FOXP2-High), while FOXP2 immunoreactivity was only visible in MEIS2/PAX6 cells with over-exposure of the FOXP2/TSHZ1 cells (FOXP2 low). We now realize that this distinction was confusing and that it is difficult to confirm low FOXP2 protein levels in the MEIS2/PAX6 class. Therefore, in the updated text, we have avoided making this high vs low distinction and we consider only cells robustly immunoreactive for FOXP2 as FOXP2-positive.

3) In line 49, the figure should be 1c, not 1b.

Thank you for catching this!

4) Figures could be better organized.

We agree that the figures could be better organized. We further considered the presentation order as the same immunohistochemistry panel often contains information about multiple LGE initial classes. We distilled the figures further in the merged manuscript to allow for a more topical organization and logical flow.

Reviewer Reports on the First Revision:

Referee #1 (Remarks to the Author):

In their revised paper that consolidates their previous submission into a single manuscript, the authors have resolved the issues that previously detracted from the wonderful dataset presented in their prior submissions. Not only have they greatly reduced the over-reliance on complex nomenclature, but they have also clarified their message with regards to how common progenitor states diversify to give species specific specializations in primate. In addition, the improved and expanded use of analytical tools (nearest neighbor analysis and RNA velocity) coupled with validation using selective RNAscope and immunocytochemistry (and the use of human organoids while perhaps a bit anecdotal was a nice additional touch) makes for a combined study that is dramatically more compelling than the previous version. In addition, I concur with their sentiment that a lineage analysis is mice, especially given that a comparable one in primates is not possible, would not help the paper's message. In sum, I now find the take home of this paper of considerable interest and feel with some minor revisions it is now suitable for publication in Nature. The focus of the paper centers on three distinct populations 1) those of the RMS 2) the novel primate TAC3 population in the striatum and 3) the claustral-associated cell types (dubbed TH+ striatum laureatum). Adding a summary diagram to figure 2 (which I admit is already rather complex) or space permitting as a summary diagram in a Figure 6 would in my estimation help the reader consolidate the findings of this work more completely. Their overarching hypothesis of the different mechanisms by which a common set of developmental genetic motifs diversify into distinct neuronal types in adult mouse versus primate is particularly appealing. I commend the authors for seriousness in which they rewrote the paper and am confident that in this new combined form it will find considerable favor with the research community.

Referee #2 (Remarks to the Author):

In this revised manuscript, Schmitz et al. addressed most of my concerns. By combining the two original manuscripts they have now a much better integrated study, composed by a useful resource and an analysis of some of the most interesting findings that stem from it. I believe that it will be much more informative for readers to contrast the species differences in early specification versus late specification of inhibitory neuronal identities. Furthermore, they made an effort to validate more of their findings, using RNAscope when antibodies against the proteins of interest were not available. This is particularly important for the developmental origin of the TAC3 interneurons.

Nevertheless, I think the authors could, and should, improve a couple of things. The figures that are related to the resource part are of much higher quality than the figures with immunofluorescence and/or in situ hybridization. Some of these are still confusing and not well organized. For example:

1) Figure 3g – where is this image from? As can be observed in panels e and f, the distribution of these markers is not homogeneous throughout the striatum. The authors should at least say in the figure legend at which level the image was taken.

2) Figure 5h – it is not clear to me that what the authors are showing is a cell. There is no doubt that the highlighted process is TH+/SCGN+. However, TH+ processes always have varicosities that can be sometimes confused with cell bodies. In this case, the SCGN+ signal, which is in fact smaller than some nuclei close to it, seems to be TH-. Is it possible to show the individual channels and/or a projection of the z planes?

3) As a minor point, ED figures 7-12 can also be improved. Some of the panels are out of focus (ED fig. 9c, for example) or have too many annotations (in ED fig.7 it is hard to see anything with

so many arrows, which are also too big).

Referee #3 (Remarks to the Author):

The authors have addressed previous concerns. The combined manuscript is substantially improved. I would recommend publication.

Jhumku Kohtz

Author Rebuttals to the First Revision:

Referees' comments:

Abstract: A subsequent paper by authors of the original “rosehip” paper (Boldog et al) found that the molecular type is not actually primate specific; rather, the molecular type is conserved but gives rise to striking morphological and physiological innovations in primates compared with mice. The subsequent analysis of this is in <https://www.nature.com/articles/s41586-019-1506-7> (Fig 5f)

We agree and have removed this reference from the abstract.

Pg 2: “We applied stringent quality control, batch correction, dimensionality reduction, Leiden clustering, and RNA velocity trajectory analysis to identify transcriptionally similar classes of progenitors and post-mitotic inhibitory neurons among 109,112 macaque and 141,065 mouse cells, which were separated by expression of DLX and GAD genes (Methods).”

As written sounds like DLX and GAD genes distinguish macaque from mouse cells (when intent is presumably to distinguish inhibitory cells from other cells in the datasets?).

We agree and have clarified this sentence.

Pg 3: Do we know that the NKX2.1/LAMP5 type has chandelier morphology? Did the studies cited there (Tasic 2017, Paul 2017) show the morphology of this cell type? Authors could also cite the recent Buzsaki paper, which characterizes at least some of these cells (superficial cortical layer resident ones) as having chandelier-like morphology: <https://www.nature.com/articles/s41593-021-00797-6?proof=t%2Btarget%3D>

Thank you for pointing this out - we have added this reference.

The shared origin of amygdala intercalated and striatal eccentric spiny neurons is very intriguing. I could not ascertain from the various stainings whether you have determined the route by which the cells enter amygdala (e.g. the summary panel in Fig 4j schematically shows the route of the eSPNs down from DL-dLGE but does not show the ITCs). Perhaps this is already known from the literature in mouse, or is impossible to tell because they migrate through developing striatum? Do you have sections from primate amygdala showing when or from which direction ITCs enter amygdala?

We believe the cells enter the amygdala along the lateral migratory stream. Extended Data Fig. 10a shows the TSHZ1 cells along the primate LMS, now labeled, in addition to TSHZ1/CASZ1 cells in the dorsal LGE highlighted in the box. This LMS position corresponds to migratory streams of Tshz1 cells from dLGE to amygdala in mouse described by Kuerbitz et al., 2018; 2021 (below).

Fig 3g. Colors of arrows are not explained, and panels have no borders so at first it looks like 1 FOV instead of separate channels. No indication of where in striatum 3g is taken. No scale bars in g, h.

Thank you - we now clarify the meaning of the arrow colors and that the magnification is from the putamen from section 66 of a PCD65 rhesus sample (Extended Data Figure 8c).

Fig 4. For consistency with previous figures, could label stains directly in figure instead of just in legend. 4i – summary figure is useful but what is the Y axis in the cartoon barplots? Scale = proportional volume of brain structure (relative to whole brain)? Proportions = ?

We agree - this adds clarity and consistency. We now label all stains directly in the figure.

Referee #1 (Remarks to the Author):

In their revised paper that consolidates their previous submission into a single manuscript, the authors have resolved the issues that previously detracted from the wonderful dataset presented in their prior submissions. Not only have they greatly reduced the over-reliance on complex nomenclature, but they have also clarified their message with regards to how common progenitor states diversify to give species specific specializations in primate. In addition, the improved and expanded use of analytical tools (nearest neighbor analysis and RNA velocity) coupled with validation using selective RNAscope and immunocytochemistry (and the use of human organoids while perhaps a bit anecdotal was a nice additional touch) makes for a combined study that is dramatically more compelling than the previous version. In addition, I concur with their sentiment that a lineage analysis in mice, especially given that a comparable one in primates is not possible, would not help the paper's message. In sum, I now find the take home of this paper of considerable interest and feel with some minor revisions it is now suitable for publication in Nature. The focus of the paper centers on three distinct populations 1) those of the RMS 2) the novel primate TAC3 population in the striatum and 3) the claustral-associated cell types (dubbed TH+ striatum laureatum). Adding a summary diagram to figure 2 (which I admit is already rather complex) or space permitting as a summary diagram in a Figure 6 would in my estimation help the reader consolidate the findings of this work more completely. Their overarching hypothesis of the different mechanisms by which a common set of developmental genetic motifs diversify into distinct neuronal types in adult mouse versus primate is particularly appealing. I commend the authors for seriousness in which they rewrote the paper and am confident that in this new combined form it will find considerable favor with the research community.

We appreciate the kind words and constructive feedback during the process. We agree that these summary figures could be expanded, however with space constraints we have tried to consolidate as much as possible with the figures we have.

Referee #2 (Remarks to the Author):

In this revised manuscript, Schmitz et al. addressed most of my concerns. By combining the two original manuscripts they have now a much better integrated study, composed by a useful resource and an analysis of some of the most interesting findings that stem from it. I believe that it will be much more informative for readers to contrast the species differences in early specification versus late specification of inhibitory neuronal identities. Furthermore, they made an effort to validate more of their findings, using RNAscope when antibodies against the proteins of interest were not available. This is particularly important for the developmental origin of the TAC3 interneurons.

Nevertheless, I think the authors could, and should, improve a couple of things. The figures that are related to the resource part are of much higher quality than the figures with immunofluorescence and/or in situ hybridization. Some of these are still confusing and not well organized. For example:

1) Figure 3g – where is this image from? As can be observed in panels e and f, the distribution of these markers is not homogeneous throughout the striatum. The authors should at least say in the figure legend at which level the image was taken.

Thank you for highlighting this - as noted above, we now clarify the meaning of the arrow colors and that the magnification is from the putamen from section 66 of a PCD65 rhesus sample (Extended Data Figure 8c).

2) Figure 5h – it is not clear to me that what the authors are showing is a cell. There is no doubt that the highlighted process is TH+/SCGN+. However, TH+ processes always have varicosities that can be sometimes confused with cell bodies. In this case, the SCGN+ signal, which is in fact smaller than some nuclei close to it, seems to be TH-. Is it possible to show the individual channels and/or a projection of the z planes?

We agree on the importance of showing the unmerged images and now include these in Extended Data Fig. 15.

3) As a minor point, ED figures 7-12 can also be improved. Some of the panels are out of focus (ED fig. 9c, for example) or have too many annotations (in ED fig.7 it is hard to see anything with so many arrows, which are also too big).

We agree and have attempted to improve, simplify, and more clearly explain the extended data figures.

Referee #3 (Remarks to the Author):

The authors have addressed previous concerns. The combined manuscript is substantially improved. I would recommend publication.

Jhumku Kohtz

We thank you for all your kind words!